# Share Your Representation Only: Guaranteed Improvement of the Privacy-Utility Tradeoff in Federated Learning

**Zebang Shen**[*]
ETH Zürich
zebang.shen@inf.ethz.ch

**Jiayuan Ye, Anmin Kang**
National University of Singapore
{jiayuanye,anmin.kang}@u.nus.edu

**Hamed Hassani**
University of Pennsylvania
hassani@seas.upenn.edu

**Reza Shokri**
National University of Singapore
reza@comp.nus.edu.sg

## Abstract

Repeated parameter sharing in federated learning causes significant information leakage about private data, thus defeating its main purpose: data privacy. Mitigating the risk of this information leakage, using state of the art differentially private algorithms, also does not come for free. Randomized mechanisms can prevent convergence of models on learning even the useful representation functions, especially if there is more disagreement between local models on the classification functions (due to data heterogeneity). In this paper, we consider a representation federated learning objective that encourages various parties to collaboratively refine the consensus part of the model, with differential privacy guarantees, while separately allowing sufficient freedom for local personalization (without releasing it). We prove that in the linear representation setting, while the objective is non-convex, our proposed new algorithm CENTAUR converges to a ball centered around the *global optimal* solution at a linear rate, and the radius of the ball is proportional to the reciprocal of the privacy budget. With this novel utility analysis, we improve the SOTA utility-privacy trade-off for this problem by a factor of $\sqrt{d}$, where $d$ is the input dimension. We empirically evaluate our method with the image classification task on CIFAR10, CIFAR100, and EMNIST, and observe a significant performance improvement over the prior work under the same small privacy budget. The code can be found in this link.

## 1 Introduction

In federated learning (FL), multiple parties cooperate to learn a model under the orchestration of a central server while keeping the data local. However, this paradigm alone is insufficient to provide rigorous privacy guarantees, even when local parties only share partial information (e.g. gradients) about their data. An adversary (e.g. one of the parties) can infer whether a particular record is in the training data set of other parties (Nasr et al., 2019), or even precisely reconstruct their training data (Zhu et al., 2019). To formally mitigate these privacy risks, we need to guarantee that any information *shared between the parties* during the training phase has *bounded information leakage* about the local data. This can be achieved using FL under differential privacy (DP) guarantees.

FL and DP are relatively well-studied separately. However, their challenges multiply when conducting FL under a DP constraint, in real-world settings where the data distributions can vary substantially across the clients (Li et al., 2020b; Acar et al., 2020; Shen et al., 2022). A direct consequence of such data heterogeneity is that the optimal local models might vary significantly across clients and differ drastically from the global solution. This results in large local gradients (Jiang et al., 2019). However, these large signals leak information about the local training data, and cannot be

---

[*]The work is done when Zebang Shen was a post-doctoral researcher at University of Pennsylvania.

communicated as such when we need to guarantee DP. We require clipping gradient values (usually by a small threshold (De et al., 2022)) before sending them to the server, to bound the sensitivity of the gradient function with respect to changes in training data Abadi et al. (2016). As the local per-sample gradients (due to data heterogeneity) tend to be large even at the global optimum, clipping per-example gradient by a small threshold and then randomizing it, will result in a high error in the overall gradient computation, and thus degrading the accuracy of the model learned via FL.

**Contributions.** In this work, we identify an important bottleneck for achieving high utility in FL under a tight privacy budget: There exists a magnified conflict between learning the representation function and classification head, when we clip gradients to bound their sensitivity (which is required for achieving DP). This conflict causes slow convergence of the representation function and disproportional scaling of the local gradients, and consequently leads to the inevitable utility drop in DP FL. To address this issue, we observe that in many FL classification scenarios, participants have minimal disagreement on data representations (Bengio et al., 2013; Chen et al., 2020; Collins et al., 2021), but possibly have very different classifier heads (e.g., the last layer of the neural network). Therefore, instead of solving the standard classification problem, we borrow ideas from the literature of model personalization and view the neural network model as a composition of a representation extractor and a small classifier head, and optimize these two components in different manners. In the proposed scheme, CENTAUR, we train a single differentially private global representation extractor while allowing each participant to have a different personalized classifier head. Such a decomposition has been considered in previous arts like (Collins et al., 2021) and (Singhal et al., 2021), but only in a non-DP setting, and also in (Jain et al., 2021), but only for a linear embedding case.

Due to low heterogeneity in data representation (compared to the whole model), the DP learned representation in our new scheme outperforms prior schemes that perform DP optimization over the entire model. In the setting where both the representation function and the classifier heads are linear w.r.t. their parameters, we prove a novel utility-privacy trade-off for an instance of CENTAUR, yielding a significant $O(\sqrt{d})$ improvement over previous art, where $d$ is the input dimension (Corollary 5.1). A major algorithmic novelty of our proposed approach is a cross-validation scheme for boosting the success probability of the classic noisy power method for privacy-preserving spectral analysis.

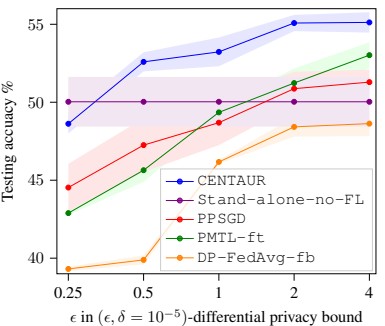

Figure 1: Privacy utility trade-off for models trained under CENTAUR and other algorithms on CIFAR10 (500 clients, 5 shards per user). Error bar denotes the std. across 3 runs.

We present strong empirical evidence for the superior performance of CENTAUR over the prior work, under the small DP budget of $(1, 10^{-5})$ in a variety of data-heterogeneity settings on benchmark datasets CIFAR10, CIFAR100, and EMNIST. Our method outperforms the prior work in all settings. Moreover, we showcase that CENTAUR *uniformly* enjoys a better utility-privacy trade-off over its competitors on the CIFAR10 dataset across different privacy budget $\epsilon$ (Figure 1). Importantly, CENTAUR outperforms the local stand-alone training even with, $\epsilon = 0.5$, thus justifying the benefit of collaborative learning compared to stand-alone training for a larger range of privacy budget.

## 1.1 RELATED WORK

Federated learning with differential privacy has been extensively studied since its emergence (Shokri & Shmatikov, 2015; McMahan et al., 2017a). Without any trusted central party, the local DP model requires each client to randomize its messages before sending them to other (malicious) parties. Consequently, the trade-off between local DP and accuracy is significantly worse than that for centralized setting and requires huge amount of data for learning even simple statistics (Duchi et al., 2014; Erlingsson et al., 2014; Ding et al., 2017). By using secure aggregation protocol, recent works (McMahan et al., 2017b; Agarwal et al., 2018; Levy et al., 2021; Kairouz et al., 2021) study user-level DP under Billboard model to enable utility. We also focus on such user-level DP setting. Model personalization approaches (Smith et al., 2017; Fallah et al., 2020; Li et al., 2020b; Arivazhagan et al., 2019; Collins et al., 2021; Pillutla et al., 2022) enable each client to learn a different (while related) model, thus alleviating the model drifting issue due to data heterogeneity. Recent works further investigate whether model personalization approaches enable improved privacy accuracy trade-off for federated learning. Hu et al. (2021) propose a private federated multi-task learning algorithm by adding task-specific regularization to each client's optimization objective. However,

the regularization has limited ability to deal with data heterogeneity. Bietti et al. (2022) propose the `PPSGD` algorithm that enables training additive personalized models with user-level DP. However, their generalization guarantees crucially rely on the convexity of loss functions. On the contrary, we study the convergence of `CENTAUR` algorithm under more general non-convex objectives.

The closest work to our approach in the literature is Jain et al. (2021), who also propose differentially privately learning shared low-dimensional linear representation with individualized classification head. However, their algorithm relies on solving the least square problem exactly at server side, by performing SVD on perturbed noisy representation matrix. Hence, the generalization guarantee of their algorithm intrinsically has an expensive $d^{1.5}$ dependency on the data dimension $d$. By contrast, we perform noisy gradient descent at server side and improve upon this error dependency by a factor of $\sqrt{d}$. Their algorithm is also limited to the linear representation learning problem, unlike our `CENTAUR` algorithm which enables training multiple layers of shared representations.

Our work builds on the `FedRep` algorithm (Collins et al., 2021), which also relies on learning shared representations between clients but does not consider privacy. In contrast, our work provides a novel private and federated representation learning framework. Moreover, a major different ingredient of our algorithm is the initialization procedure, which requires performing *differentially private* SVD to the data matrix. We use the noisy power method (Hardt & Price, 2014) as a crucial tool to enable a constant probability for utility guarantee. We then perform cross-validation to further boost success probability to arbitrarily large (inspired by Liang et al. (2014)).

## 2 NOTATIONS AND BACKGROUND ON PRIVACY

**Notations.** We denote the clipping operation by $\mathrm{clip}(\boldsymbol{x}; \zeta) \doteq \boldsymbol{x} \cdot \min\{1, \zeta/\|\boldsymbol{x}\|\}$, and denote the Gaussian mechanism as $\mathrm{GM}_{\zeta,\sigma}(\{\boldsymbol{x}_i\}_{i=1}^s) \doteq \frac{1}{s}(\sum_{i=1}^s \mathrm{clip}(\boldsymbol{x}_i; \zeta) + \sigma\zeta W)$ where $W \sim \mathcal{N}(0, \mathbb{I})$.

Define Rényi Differential Privacy (RDP) on a dataset space $\mathcal{D}$ equipped with a distance d as follows.

**Definition 2.1** (RDP). *For measures $\nu, \nu'$ over the same space with $\nu \ll \nu'$, their Rényi divergence $R_\alpha(\nu, \nu') = \frac{1}{\alpha-1} \log E_\alpha(\nu, \nu')$, where $E_\alpha(\nu, \nu') = \int \left(\frac{\mathrm{d}\nu}{\mathrm{d}\nu'}\right)^\alpha \mathrm{d}\nu'$. A randomized algorithm $\mathcal{M} : \mathcal{D} \to \Theta$ satisfies $(\alpha, \epsilon)$-RDP, if $\forall \mathbb{D}, \mathbb{D}' \in \mathcal{D}$ with $\mathrm{d}(\mathbb{D}, \mathbb{D}') \le 1$, we have $R_\alpha(\mathcal{M}(\mathbb{D}), \mathcal{M}(\mathbb{D}')) \le \epsilon$.*

**User-level-RDP.** Let $\mathcal{D}$ be the space of all of $n$ tuples of local datasets $\{\mathbb{S}_i\}_{i=1}^n$, where each local dataset consists of $m$ data points, i.e. $\mathcal{D} = \{\{\mathbb{S}_i\}_{i=1}^n \mid \mathbb{S}_i = \{\boldsymbol{z}_{ij}\}_{j=1}^m\}$. The distance d is the Hamming distance in the dataset level, i.e. $\mathrm{d}(\{\mathbb{S}_i\}_{i=1}^n, \{\mathbb{S}_i'\}_{i=1}^n) = \sum_{i=1}^n \mathbf{1}_{\mathbb{S}_i \ne \mathbb{S}_i'}$. We refer to the privacy guarantee recovered by this choice of dataset space as user-level-RDP.

In Appendix A.3, we further describe the Gaussian Mechanism and the composition of RDP, the standard notion of Differential Privacy (DP), and the conversion Lemma from RDP to DP.

**Threat Models.** We aim to protect the privacy of each user against potential adversarial other clients, i.e., any eavesdropper will not be able to tell whether one users has participated in the collaborative learning procedure, given the information released during training phase. We establish user-level RDP guarantees under the billboard model, which is a communication protocol that is particularly compatible with algorithms in the federated setting and has been adopted in many previous works (Jain et al., 2021; Hu et al., 2021; Bietti et al., 2022). In this model, a trusted server (either a trusted party or one that uses cryptographic techniques like multiparty computation) aggregates information subject to a DP constraint, which is then shared as public messages with all the $n$ users. Then, each user computes its own personalized model based on the public messages and its own private data (Hsu et al., 2014). Our RDP guarantees only hold for releasing the shared representations $b_{T_g}$ in Algorithm 1, and the guarantees are equivalent to joint-(R)DP guarantees Kearns et al. (2014) if all the personalized models $w_i^{T_l}$ of individual users were additionally released as outputs of Algorithm 2.

## 3 PROBLEM FORMULATION

Consider a Federated Learning (FL) setting where there are $n$ agents interacting with a central server and the $i$th agent holds $m_i$ local data points $\mathbb{S}_i = \{(\boldsymbol{x}_{ij}, y_{ij})\}_{j=1}^{m_i}$. Here $(\boldsymbol{x}_{ij}, y_{ij}) \in \mathbb{R}^d \times \mathbb{R}$ denotes a pair of an input vector and the corresponding label. Suppose that each local dataset $\mathbb{S}_i$ is sampled from an underlying joint distribution $p_i(\boldsymbol{x}, y)$ of the input and response pair. We consider the *data-heterogenous* setting where potentially $p_i \ne p_j$ for $i \ne j$. We further consider a hypothesis model

$f(\cdot; \boldsymbol{\theta}) : \mathbb{R}^d \to \mathbb{R}$ which maps an input $\boldsymbol{x}$ to the predicted label $y$. Here $\boldsymbol{\theta} \in \mathbb{R}^q$ is the trainable parameter of the model $f$. Let $\ell : \mathbb{R} \times \mathbb{R} \to \mathbb{R}$ be a loss function that penalizes the error of the prediction $f(\boldsymbol{x}; \boldsymbol{\theta})$ given the true label $y$. The local objective on client $i$ is defined as

$$l(\boldsymbol{\theta}; \mathbb{S}_i) \doteq \frac{1}{m_i} \sum_{j=1}^{m_i} \ell(f(\boldsymbol{x}_{ij}; \boldsymbol{\theta}), y_{ij}). \tag{1}$$

The standard FL formulation seeks a global consensus solution, whose objective is defined as

$$\arg\min_{\theta} \mathcal{L}(\boldsymbol{\theta}) \doteq \frac{1}{n} \sum_{i=1}^{n} l(\boldsymbol{\theta}; \mathbb{S}_i). \tag{2}$$

While this formulation is reasonable when the local data distributions are similar, the obtained global solution may be far from the local optima $\arg\min l(\boldsymbol{\theta}; \mathbb{S}_i)$ under diverse local data distributions, a phenomenon known as *statistical heterogeneity* in the FL literature (Li et al., 2020a; Wang et al., 2019; Malinovskiy et al., 2020; Mitra et al., 2021; Charles & Konečnỳ, 2021; Acar et al., 2020; Karimireddy et al., 2020). Such a (potentially significant) mismatch between local and global optimal solutions limits the incentive for collaboration, and cause extra difficulties when DP constraints are imposed (Remark 3.1). These considerations motivate us to search for personalized solutions that can be learned in a federated fashion, with less utility loss due to the DP constraint.

**Federated representation learning with differential privacy.** It is now well-documented that in some common and real-world FL tasks, such as image classification and word prediction, clients have minimal disagreement on data representations (Chen et al., 2020; Collins et al., 2021). Based on this observation, a more reasonable alternative to the FL objective in equation 2 should focus on learning the data representation, which is the information that is agreed on among most parties, while also allowing each client to personalize its learning on information that the other clients disagree on. To formalize this, suppose that the variable $\boldsymbol{\theta} \in \mathbb{R}^q$ can be partitioned into a pair $[\boldsymbol{w}, \boldsymbol{b}] \in \mathbb{R}^{q_1} \times \mathbb{R}^{q_2}$ with $q = q_1 + q_2$ and the parameterized model admits the composition $f = h \circ \phi$ where $\phi : \mathbb{R}^d \to \mathbb{R}^k$ is the *representation extractor* that maps $d$-dimensional data points to a lower dimensional space of size $k$ and $h : \mathbb{R}^k \to \mathbb{R}$ is a *classifier head* that maps from the lower dimensional subspace to the space of labels. An important example is bottom and top layers of the neural network model. We use $\boldsymbol{w}$ and $\boldsymbol{b}$ to denote the parameters that determine $h$ and $\phi$, respectively. With the above notation, we consider the following federated representation learning (FRL) problem

$$\min_{\boldsymbol{b} \in \mathbb{B}} \frac{1}{n} \sum_{i=1}^{n} \min_{\boldsymbol{w}_i} \left\{ l([\boldsymbol{w}_i, \boldsymbol{b}]; \mathbb{S}_i) := \frac{1}{m_i} \sum_{j=1}^{m_i} \ell(h(\phi(\boldsymbol{x}_{ij}; \boldsymbol{b}); \boldsymbol{w}_i), y_{ij}) \right\}, \tag{3}$$

where we maintain a single global representation extraction function $\phi(\cdot; \boldsymbol{b})$ subject to the constraint $\boldsymbol{b} \in \mathbb{B} \subseteq \mathbb{R}^{q_2}$ while allowing each client to use its personalized classification head $h(\cdot; \boldsymbol{w}_i)$ locally. The constraint $\mathbb{B}$ is included so that equation 3 also covers the linear case studied in section 5.

The choice of the FRL formulation in equation 3 entails considerations from both DP and optimization perspectives: From the DP standpoint, the phenomenon of statistical heterogeneity introduces additional difficulties for federated learning under DP constraint (see Remark 3.1 below). If the clients collaborate to train only a shared representation function, then the aforementioned disadvantages can be alleviated; From the optimization standpoint, we typically have $k \ll d$, i.e. the dimension of the extracted features is much smaller than that of the original input. Hence, for a fixed representation function $\phi(\cdot; \boldsymbol{b})$, the client specific heads $h(\cdot; \boldsymbol{w}_i)$ are in general easy to optimize locally as the number of parameters, $k$, is typically small.

**Remark 3.1** (Statistical heterogeneity makes DP guarantee harder to establish.). *To establish DP guarantees for gradient based methods, e.g. DP-SGD, a common choice is the Gaussian mechanism, which is comprised of the gradient clipping step and the noise injection step. It is empirically observed that to achieve a better privacy-utility trade-off, a small clipping threshold is preferred, since it limits the large variance due to the injected noise (De et al., 2022). Moreover, the effect of the bias (due to clipping) subsides as the per-sample gradient norm diminishes during the centralized training, a phenomenon known as benign overfitting in deep learning (Bartlett et al., 2020; Li et al., 2021; Bartlett et al., 2021). However, due to the phenomenon of distribution shift, the local (per-sample) gradients in the standard FL setting (described in equation 2) remain large even at the global optimal solution, and hence setting a small (per-sample) gradient clipping threshold will*

---

**Algorithm 1** SERVER procedure of CENTAUR

1: **procedure** SERVER($\boldsymbol{b}^0, p_g, T_g, \eta_g, \sigma_g, \zeta_g$)
2:     // $\boldsymbol{b}^0$ is obtained from the INITIALIZATION procedure.
3:     **for** $t \leftarrow 0$ to $T_g - 1$ **do**
4:         Sample set $\mathbb{C}^t$ of active clients using Poisson sampling with parameter $p_g$.
5:         Broadcast the current global representation parameter $\boldsymbol{b}^t$ to active clients.
6:         Receive the local update directions $\{\boldsymbol{g}_i^t\}_{i \in \mathbb{C}^t}$ from the CLIENT procedures.
7:         Compute the update direction $\boldsymbol{g}^t = \text{GM}_{\zeta_g, \sigma_g}(\{\boldsymbol{g}_i^t\}_{i \in \mathbb{C}^t})$
8:         Update the global representation function $\boldsymbol{b}^{t+1} := \text{AGGREGATION}(\boldsymbol{b}^t, \boldsymbol{g}^t, \eta_g)$.
9:         // The AGGREGATION procedure depends on the feasible set $\mathbb{B}$ in equation 3.
10:     **return** $\boldsymbol{b}^{T_g}$.

---

**Algorithm 2** CLIENT procedure of CENTAUR in the general case (for client $i$)

1: **procedure** CLIENT($\boldsymbol{b}^t, \bar{m}, T_l, \eta_l$)
2:     [Phase 1: Local classifier update.] $\boldsymbol{w}_i^{t+1} = \arg\min_{\boldsymbol{w}} l([\boldsymbol{b}^t, \boldsymbol{w}]; \mathbb{S}_i)$.
3:     [Phase 2: Local representation function update.] Set $\boldsymbol{b}_i^{t,0} = \boldsymbol{b}^t$;
4:     **for** $s \leftarrow 0$ to $T_l - 1$ **do**
5:         Sample a subset $\mathbb{S}_i^s$ of size $\bar{m}$ from the local dataset $\mathbb{S}_i$ without replacement
6:         Update the local representation function $\boldsymbol{b}_i^{t,s+1} := \boldsymbol{b}_i^{t,s} - \eta_l \cdot \partial_{\boldsymbol{b}} l([\boldsymbol{b}_i^{t,s}, \boldsymbol{w}_i^{t+1}]; \mathbb{S}_i^s)$.
7:     [Phase 3: Summarize the local update direction.] **return** $\boldsymbol{g}_i^t := \boldsymbol{b}_i^{t,T_l} - \boldsymbol{b}^t$.

---

*result in a large and non-diminishing bias in the overall gradient computation.*
*In contrast, for tasks where the representation extracting functions are approximately homogeneous, the local and global optimal of the FRL formulation 3 are close and hence the gradients w.r.t. the representation function vanishes at the optimal, which is amiable to small clipping threshold.*

## 4 DIFFERENTIAL PRIVATE FEDERATED REPRESENTATION LEARNING

In this section we present the proposed CENTAUR method to solve the FRL problem in equation 3.

**SERVER procedure (Algorithm 1)** takes the following quantities as inputs: $\boldsymbol{b}^0$ denotes the initializer for the parameter of the global representation function, obtained from a procedure INITIALIZATION; $p_g$ denotes the portion of the clients that will participate in training per global communication round; $T_g$ denotes the total number of global communication rounds; $\eta_g$ denotes the global update step size; $(\sigma_g, \zeta_g)$ stand for the noise multiplier and the clipping threshold of the Gaussian mechanism (that ensures user-level RDP). Note that in this section, we consider random INITIALIZATION over unconstrained space $\mathbb{B} = \mathbb{R}^{q_2}$, and the procedure AGGREGATION($\boldsymbol{b}^t, \boldsymbol{g}^t, \eta_g) = \boldsymbol{b}^t + \eta_g \cdot \boldsymbol{g}^t$. Under these configurations, SERVER follows the standard FL protocol: After broadcasting the current global representation function to the activate clients, it aggregates the information returned from the CLIENT procedure to update the global representation function.

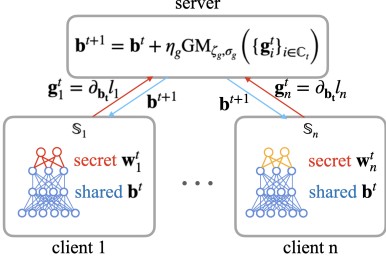

Figure 2: The $t$-th global round of the CENTAUR algorithm, where clients keep their classification head $\boldsymbol{w}_i^t$ secret while updating shared representation $\boldsymbol{b}^t \rightarrow \boldsymbol{b}^{t+1}$ based on perturbed gradients $\boldsymbol{g}_i^t$ from sampled clients $i \in \mathbb{C}_t$.

**CLIENT procedure (Algorithm 2)** takes the following quantities as inputs: $\boldsymbol{b}^t$ denotes the parameter of the global representation function received from the server; $\bar{m}$ denotes the number of local data points used as mini-batch to update the local representation function; $T_l$ denotes the number of local update iterations; $\eta_l$ denotes the local update step size. CLIENT can be divided into three phases: 1. After receiving the current global parameter $\boldsymbol{b}^t$ of the representation function from the server, the client update the local classifier head to the minimizer of the local objective $\boldsymbol{w}_i^{t+1} = \arg\min_{\boldsymbol{w}} l([\boldsymbol{b}^t, \boldsymbol{w}]; \mathbb{S}_i)$. This is possible since the local objective $l$ usually admits very simple structure, e.g. it is convex w.r.t. $\boldsymbol{w}$, once the representation function is fixed. In prac-

tice, we would run multiple SGD epochs on $\boldsymbol{w}$ to approximately minimize $l$ with $\boldsymbol{b}^t$ fixed. This is computationally cheap since the dimension of $\boldsymbol{w}$ is much smaller compared to the whole variable $\boldsymbol{\theta} = [\boldsymbol{w}, \boldsymbol{b}]$. 2. Once the local classifier head is updated to $\boldsymbol{w}_i^{t+1}$, the client optimizes its local representation function with multiple SGD steps, starting from the current global consensus $\boldsymbol{b}^t$. 3. Finally, each client calculate its local update direction for representation via the difference between its latest local representation function and the previous global consensus, $\boldsymbol{b}_i^{t,T_l} - \boldsymbol{b}^t$.

**Remark 4.1** (Privacy guarantee). *By the composition theorem for subsampled Gaussian Mechanism Mironov et al. (2019), we prove that Algorithm 1 and Algorithm 2 satisfies user-level $(\alpha, \epsilon)$-Rényi differential privacy for $\epsilon = T_g \cdot S_\alpha(p_g, \sigma_g)$, where $S_\alpha(p_g, \sigma_g) = R_\alpha(\mathcal{N}(0, \sigma_g^2) \| \mathcal{N}(0, \sigma_g^2) + p_g \cdot \mathcal{N}(1, \sigma_g^2))$. In the special case of full gradient descent with $p_g = 1$, we have that $\epsilon = \alpha \cdot T_g / (2\sigma_g^2)$.*

## 5 GUARANTEED IMPROVEMENT OF THE UTILITY-PRIVACY TRADE-OFF

In the previous section, we present CENTAUR for the general FRL problems (3). Due to the lack of structure information, for a general non-convex optimization problem we cannot expect any utility guarantee beyond the convergence to a stationary point. In this section, we consider a specific instance of the FRL problems where both the representation function $\phi$ and the local classifiers $h$ are *linear* w.r.t. their parameters $\boldsymbol{b}$ and $\boldsymbol{w}_i$. This model has been commonly used in the analysis of representation learning (Collins et al., 2021; 2022), meta learning (Tripuraneni et al., 2021; Du et al., 2020; Thekumparampil et al., 2021; Sun et al., 2021), model personalization (Jain et al., 2021), multi-task learning (Maurer et al., 2016). For this (still nonconvex) instance, we prove that CENTAUR converges to a ball centered around the *global minimizer* in a linear rate where the size of the ball depends on the required RDP parameters $(\alpha, \epsilon)$. Let $\epsilon_a$ be the utility and let $\epsilon_{dp}$ be the DP-parameter (CENTAUR is an $(\epsilon_{dp}, \delta)$-DP mechanism). We obtain the improved privacy-utility tradeoff $\epsilon_a \cdot \epsilon_{dp} \geq O(d/n)$, which is $O(\sqrt{d})$ better than the current SOTA result, $\epsilon_a \cdot \epsilon_{dp} \geq O(d^{1.5}/n)$, by (Jain et al., 2021). In the following, we will first review objective (3) in the linear representation setting, and analyse CENTAUR to show the improved utility-privacy tradeoff.

**Federated Linear Representation Learning (LRL)** Recall the problem formulation in Section 3. For simplicity, we assume $m_i = m$ for all $i \in \{1, \ldots, n\}$ for some constant $m$. Consider the LRL setting, where given the input $\boldsymbol{x}_{ij}$ the response $y_{ij} \in \mathbb{R}$ satisfies $y_{ij} = \boldsymbol{w}_i^{*\top} \boldsymbol{B}^{*\top} \boldsymbol{x}_{ij}$. [1] Here $\boldsymbol{B}^* \in \mathbb{R}^{d \times k}$ is a *column orthonormal* global representation matrix and $\boldsymbol{w}_i^* \in \mathbb{R}^k$ is an agent-specific optimal local linear classifiers. In terms of the notations, $\boldsymbol{B}$ corresponds to the parameter $\boldsymbol{b}$ in the general formulation (3), but is capitalized since it is now regarded as a matrix. The feasible domain $\mathbb{B}$ is the set of column orthonormal matrices $\mathbb{O}_{d,k} = \{\boldsymbol{B} \in \mathbb{R}^{d \times k} | \boldsymbol{B}^\top \boldsymbol{B} = \boldsymbol{I}_k\}$.
Given a local dataset $\mathbb{S}$ (could be $\mathbb{S}_i$ or its subset), define

$$l([\boldsymbol{w}, \boldsymbol{B}]; \mathbb{S}) = \frac{1}{|\mathbb{S}|} \sum_{(\boldsymbol{x}, y) \in \mathbb{S}} \frac{1}{2} (\boldsymbol{w}^\top \boldsymbol{B}^\top \boldsymbol{x} - y)^2. \tag{4}$$

Our goal is to recover the ground truth representation matrix $\boldsymbol{B}^*$ using the collection of the local datasets $\{\mathbb{S}_i\}_{i=1}^n$ via solving the following linear instance of the FRL problem equation 3

$$\min_{\boldsymbol{B}^\top \boldsymbol{B} = \boldsymbol{I}_k} \frac{1}{n} \sum_{i=1}^n \min_{\boldsymbol{w}_i} l([\boldsymbol{w}_i, \boldsymbol{B}]; \mathbb{S}_i) \tag{L-FRL}$$

in a federated and DP manner. Here, equation L-FRL is an instance of the general FRL problem (3) with $h$ and $\phi$ set to linear functions and $\ell$ set to the least square loss. Note that, despite the (relatively) simple structure of equation L-FRL, it is still *non-convex* w.r.t. the variable $\boldsymbol{B}$.

**Methodology for the LRL case.** To establish our novel privacy and utility guarantees, we need to specify INITIALIZATION and AGGREGATION in the procedures SERVER and we also need to slightly modify the CLIENT procedure, which are elaborated as follows.
1. To obtain the novel guarantee of converging to the global minimizer for the LRL case, the initializer $\boldsymbol{B}^0$ needs to be within a constant distance to the global optimal $\boldsymbol{B}^*$ which requires a more

---

[1] A similar problem is considered in (Jain et al., 2021), but the measurement $y_{ij}$ suffers from an extra white noise with variance $\sigma_F$. In our paper, we consider the noiseless case and hence when comparing with (Jain et al., 2021), we treat the $\sigma_F = 0$ in their results for a fair comparison.

---

**Algorithm 3** INITIALIZATION procedure for CENTAUR in the LRL case.

1: **procedure** INITIALIZATION($T_0, \epsilon_i, L_0, \sigma_0, \zeta_0$)
2:     Run $T_0$ independent copies of PPM($L, \sigma_0, \zeta_0$) to obtain $T_0$ candidates $\{\boldsymbol{B}_c\}_{c=1}^{T_0}$;
3:     // PPM stands for private power method and is presented in Algorithm 5 in the appendix.
4:     // Boost this probability to $1 - n^{-k}$ via cross validation.
5:     Find $\hat{c}$ in $[T_0]$ such that for at least half of $c \in [T_0]$, $s_i(\boldsymbol{B}_c^\top \boldsymbol{B}_{\hat{c}}) \geq 1 - 2\epsilon_i^2, \forall i \in [k]$.
6:     // $s_i(\cdot)$ denotes the $i^{th}$ singular value of a matrix.
7:     **return** $\boldsymbol{B}_{\hat{c}}$.

---

**Algorithm 4** CLIENT procedure for CENTAUR in the LRL case

1: **procedure** CLIENT($\boldsymbol{B}^t, \bar{m}$)
2:     Sample without replacement two subsets $\mathbb{S}_i^{t,1}$ and $\mathbb{S}_i^{t,2}$ from $\mathbb{S}_i$ both with cardinality $\bar{m}$.
3:     [Phase 1:] Update the local head $\boldsymbol{w}_i^{t+1} = \arg\min_{\boldsymbol{w} \in \mathbb{R}^k} l(\boldsymbol{w}, \boldsymbol{B}^t; \mathbb{S}_i^{t,1})$.
4:     [Phase 2:] Compute the local gradient of the representation $\boldsymbol{G}_i^t = \partial_{\boldsymbol{B}} l([\boldsymbol{w}_i^{t+1}, \boldsymbol{B}^t]; \mathbb{S}_i^{t,2})$;
5:     [Phase 3:] **return** $\boldsymbol{G}_i^t$

---

sophisticated procedure than the simple random initialization. We show that this requirement can be ensured by running a modified instance of the private power method (PPM) by (Hardt & Price, 2014), but the utility guarantee only holds *with a constant probability* (Lemma F.1). A key contribution of our work is to propose a novel *cross-validation* scheme to boost the success probability of the PPM with a small extra cost. Our scheme only takes as input the outputs of independent PPM trials, and hence can be treated as post-processing, which is free of privacy risk (Lemma F.3). The proposed INITIALIZATION procedure is presented in Algorithm 3. The analyses are deferred to Appendix F.

2. As discussed earlier, the feasible domain $\mathbb{B}$ is the set of column orthonormal matrices. In order to ensure the feasibility of $\boldsymbol{B}^{t+1}$, we set AGGREGATION($\boldsymbol{B}^t, \boldsymbol{G}^t, \eta_g$) = $\mathcal{QR}(\boldsymbol{B}^t - \eta_g \cdot \boldsymbol{G}^t)$, where $\mathcal{QR}(\cdot)$ denotes the QR decomposition and only returns the $Q$ matrix.

3. We make a small modification in line 2 where two subsets $\mathbb{S}_i^{t,1}$ and $\mathbb{S}_i^{t,2}$ of the local dataset are sampled *without replacement* from $\mathbb{S}_i$ and are used to replace $\mathbb{S}_i$ in Phases 1 and 2. This change is required to establish our novel utility result, which ensures that clipping threshold of the Gaussian mechanism (line 7) in the SERVER procedure is never reached with a high probability (Lemma C.1).

## 5.1 ANALYSIS OF CENTAUR IN THE LRL SETTING

Use $s_i(\boldsymbol{A})$ to denote the $i$th largest singular value of a matrix $\boldsymbol{A}$. Let $\boldsymbol{W}^* \in \mathbb{R}^{n \times k}$ be the collection of the local optimal classifier heads with $\boldsymbol{W}_{i,:}^* = \boldsymbol{w}_i^*$. We use $s_i$ as a shorthand for $s_i(\boldsymbol{W}^*/\sqrt{n})$ and use $\kappa = s_1/s_k$ to denote the condition number of the problem. We choose the scaling $1/\sqrt{n}$ such that $s_i$ remains meaningful as $n \to \infty$. We make the following assumptions.

**Assumption 5.1.** $\boldsymbol{x}_{ij}$ *is zero mean,* $\boldsymbol{I}_d$*-covariance, 1-subgaussian (defined in Appendix A.1).*

**Assumption 5.2.** *There exists a constant $\mu > 0$ such that $\max_{i \in [n]} \|\boldsymbol{w}_i^*\|_2 \leq \mu \sqrt{k} s_k$.*

**Assumption 5.3.** *The number of local data points is sufficiently large:* $m \geq \tilde{c}_{ld} \max(k^2, k^4 d/n)$. *Here $\tilde{c}_{ld}$ hides the dependence on $\kappa$, $\mu$, and the log factors.*

These assumptions are standard in literature (Collins et al., 2021; Jain et al., 2021). An elaborated discussion is provided in Appendix K. While Problem (L-FRL) is non-convex, we show that the consensus representation $\boldsymbol{B}^t$ converges to a ball centered around the ground truth solution $\boldsymbol{B}^*$ in a linear rate. The size of the ball is controlled by the noise multiplier $\sigma_g$, which will be the free parameter that controls the utility-privacy tradeoff.

**High level idea.** For Problem (L-FRL), we find an initial point close to the ground truth solution via the method of moments. Given this strong initialization, CENTAUR converges linearly to the vicinity of the ground truth since it can be interpreted as an inexact gradient descent method with a fast decreasing bias term. One caveat that requires particular attention is the clipping step in the Gaussian mechanism (line 5 in Algorithm 4) will destroy the above interpretation if the threshold parameter $\zeta_g$ is set too small. To resolve this, we set $\zeta_g$ to be a high probability upper bound of $\|\boldsymbol{G}_i^t\|_F$ so that the clipping step only takes effect with a negligible probability.

In the utility analysis of the LRL case, we use the principal angle distance to measure the convergence of the column-orthonormal variable $\boldsymbol{B}$ towards the ground truth $\boldsymbol{B}^*$. We also refer to this quantity as the *utility* of the algorithm. Let $\boldsymbol{B}_\perp$ be the orthogonal complement of $\boldsymbol{B}$. We define

$$\text{dist}(\boldsymbol{B}, \boldsymbol{B}^*) := \|(\boldsymbol{I}_d - \boldsymbol{B}\boldsymbol{B}^\top)\boldsymbol{B}^*\|_2 = \|\boldsymbol{B}_\perp^\top \boldsymbol{B}^*\|_2 \leq 1.$$

To simplify the presentation of the result, we make the following assumptions: The dimension of the input is sufficiently large $d \geq \kappa^8 k^3 \log n$ and the number of clients is sufficiently large $n \geq m$. The proof of the following theorem can be find in Appendix C.

**Theorem 5.1** (Utility Analysis). *Consider the instance of* CENTAUR *for the LRL setting with its* CLIENT *and* INITIALIZATION *procedures defined in Algorithms 4 and 3 respectively. Suppose that the matrix $\boldsymbol{B}^0$ returned by* INITIALIZATION *satisfies* $\text{dist}(\boldsymbol{B}^0, \boldsymbol{B}^*) \leq \epsilon_0 = 0.2$, *and suppose that the mini-batch size parameter $\bar{m}$ satisfies $\bar{m} \geq c_m \max\{\kappa^2 k^2 \log n, k^2 d/n\}$. Set the clipping threshold of the Gaussian mechanism $\zeta_g = c_\zeta \mu^2 k s_k^2 \sqrt{dk \log n}$, the global step size $\eta_g = 1/4s_1^2$, the number of global rounds $T_g = c_T \kappa^2 \log(\kappa \eta_g \zeta_g \sigma_g d/n)$. Assuming that the noise multiplier in the Gaussian mechanism is sufficiently small[2]: $\sigma_g \leq c_\sigma n \kappa^4/(\mu^2 d \sqrt{k \log n})$. Let $c_m$, $c_\zeta$, $c_T$, $c_\sigma$, $c_p$ and $c_d$ be universal constants. We have with probability at least $1 - c_p \bar{m} T_g \cdot n^{-k}$,* $\text{dist}(\boldsymbol{B}^{T_g}, \boldsymbol{B}^*) \leq c_d \kappa^2 \eta_g \sigma_g \zeta_g \sqrt{d}/n = \tilde{c}_d \sigma_g \mu^2 k^{1.5} d/n$. [3]

Since the SERVER procedure remains exactly the same as Algorithm 1 in the LRL case, the main body (anything after INITIALIZATION) of the resulting CENTAUR instance has the same privacy guarantee as described in remark 4.1. However, we still need to account for the privacy leakage of the INITIALIZATION procedure in Algorithm 3 as it is data-dependent. This will be deferred to Appendix F, where we show that Algorithm 3 is an $(\alpha, \epsilon_{init})$-RDP mechanism, with $\epsilon_{init}$ defined in Corollary F.1. Combining this fact with the RDP analysis for the main body leads to the following privacy guarantee for CENTAUR in the LRL case (see Appendix A.4).

**Theorem 5.2** (Privacy Bound). *Consider the instance of* CENTAUR *with its* CLIENT *procedure defined in Algorithm 4 and its* INITIALIZATION *procedure defined in Algorithm 3. Suppose that the* INITIALIZATION *procedure is an $(\alpha, \epsilon_{init})$-RDP mechanism (proved in Corollary F.1). Let $\sigma_g$, the noise multiple in the* CLIENT *procedure, be a free parameter that controls the privacy-utility trade-off. By setting the inputs to* SERVER *as $p_g = 1$, $T_g = c_T \kappa^2 \log(\kappa \eta_g \zeta_g \sigma_g d/n)$, the instance of* CENTAUR *under consideration is an $(\alpha, \epsilon_{init} + \epsilon_{rdp}/2)$-RDP mechanism, where $\epsilon_{rdp} = 4\alpha T_g/\sigma_g^2$. Moreover, when $\sigma_g = \tilde{O}(n/(k^4 \mu^2 d))$, we have $\epsilon_{init} \leq \epsilon_{rdp}/2$, in which case* CENTAUR *is an $(\alpha, \epsilon_{rdp})$-RDP mechanism and is also an $(\epsilon_{dp}, \delta)$-DP mechanism with $\epsilon_{dp} = 2\sqrt{T_g \log(1/\delta)}/\sigma_g$.*

**Overall Utility-Privacy Trade-off** We now combine the utility and privacy analyses of CENTAUR in the LRL setting to obtain the overall utility-privacy trade-off in the following sense: According to Theorem 5.1, to achieve a high utility, i.e. a small $\epsilon_a$, we need to choose a small noise multiplier $\sigma_g$ while Theorem 5.2 states that the smaller $\sigma_g$ is, the larger the privacy cost.

**Corollary 5.1.** *Use $\epsilon_a$ to denote a target utility, i.e. $\text{dist}(\boldsymbol{B}^T, \boldsymbol{B}^*) \leq \epsilon_a$ where $\boldsymbol{B}^T$ is the output of* CENTAUR *and use $\epsilon_{dp}$ to denote a privacy budget, i.e.* CENTAUR *is an $(\epsilon_{dp}, \delta)$-DP mechanism. Suppose that $\epsilon_{dp} \geq \tilde{c}_t' \mu^2 d\sqrt{k}/(\kappa^3 n)$, which is a restriction due to the requirement on $\sigma_g$ in Theorem 5.1. Under Assumptions 5.1 to 5.3,* CENTUAR *outputs a solution that provably achieves the $\epsilon_a$ utility within the $\epsilon_{dp}$ budget, under the condition that the tuple $(\epsilon_a, \epsilon_{dp})$ satisfies $\tilde{c}_t \kappa k^{1.5} \mu^2 d/n \leq \epsilon_a \cdot \epsilon_{dp}$, where $\tilde{c}_t$ and $\tilde{c}_t'$ hide the constants and $\log$ terms.*

When focusing on the input dimension $d$ and the number of clients $n$ and treating other factors as constants, the restriction on $\epsilon_{dp}$ and the trade-off of the tuple $(\epsilon_a, \epsilon_{dp})$ can be simplified to $\epsilon_{dp} \geq \Theta(d/n)$ and $\Theta(d/n) \leq \epsilon_a \cdot \epsilon_{dp}$. Recall that in the previous SOTA result of the LRL setting (Jain et al., 2021), the restriction on the DP budget is $\epsilon_{dp} \geq \Theta(d^{1.5}/n)$ (point iii in Assumption 4.1 therein) and the utility-privacy tradeoff can be interpreted as $\Theta(d^{1.5}/n) \leq \epsilon_a \cdot \epsilon_{dp}$ (Lemma 4.4

---

[2]The intuition behind this requirement is that our convergence analysis requires the iterates to stay within a ball centered around the ground truth, with a constant radius (measured in terms of the principal angle distance). Adding a large noise will break this argument. Similar requirements are made in Tripuraneni et al. (2021).

[3]Note that Jain et al. (2021) use the Frobenius norm (instead of the spectral norm) of $\boldsymbol{B}_\perp^\top \boldsymbol{B}^* \in \mathbb{R}^{d \times k}$ as the optimality metric. However, since $rank(\boldsymbol{B}_\perp^\top \boldsymbol{B}^*) \leq k$, we can always bound $\|\boldsymbol{B}_\perp^\top \boldsymbol{B}^*\|_F \leq \sqrt{k}\text{dist}(\boldsymbol{B}, \boldsymbol{B}^*)$. With this extra factor, Theorem 5.1 quadratically depends on $k$, same as Lemma 4.4 in (Jain et al., 2021), while the dependency on $d$ is substantially reduced, from $d^{1.5}$ to $d$.

Table 1: Testing accuracy (%) on CIFAR10/CIFAR100/EMNIST with various data allocation settings. No data augmentation is used for training the representations. $n$ stands for the number clients and $S$ stands for the number classes per client. The $\delta$ parameter of DP is fixed to $10^{-5}$ as a common choice in the literature. The budget parameter of DP is fixed to a small value of 1 for results in this table, i.e. $\epsilon_{dp} = 1$.

| Methods | Stand-alone-no-FL | DP-FedAvg-fb | PMTL-ft | PPSGD | CENTAUR |
|---|---|---|---|---|---|
| CIFAR10, n=500, S=5 | 50.03 (1.59) | 46.17 (0.12) | 49.35 (0.44) | 48.69 (1.44) | **53.23 (0.92)** |
| CIFAR10, n=1000,S=2 | 74.06 (0.45) | 57.02 (1.36) | 67.79 (1.77) | 71.52 (2.30) | **76.23 (0.48)** |
| CIFAR10, n=1000,S=5 | 44.60 (1.30) | 37.15 (1.22) | 35.99 (1.79) | 44.61 (2.68) | **49.92 (0.71)** |
| CIFAR100, n=1000,S=5 | 39.20 (1.12) | 22.17 (2.12) | 24.17 (1.43) | 32.97 (1.48) | **44.54 (1.05)** |
| EMNIST, n=1000,S=5 | 93.47 (0.14) | 91.32 (0.22) | 92.32 (0.22) | 93.44 (0.20) | **94.17 (0.19)** |
| EMNIST, n=2000,S=5 | 90.67 (0.46) | 86.85 (1.11) | 88.81 (2.08) | 88.96 (1.93) | **92.79 (0.25)** |

therein). Hence, we obtain a $\Theta(\sqrt{d})$ improvement in both regards, which means that CENTAUR delivers the utility-privacy guarantees for a much wider range of combinations of $(\epsilon_a, \epsilon_{dp})$. Please see an elaborated discussion of this result in Appendix J.

## 6 EXPERIMENTS

In this section, we present the empirical results that show the significant advantage of the proposed CENTAUR over previous arts. Four baselines are included: Stand-alone-no-FL which stands for local stand-alone training; DP-FedAvg-fb which stands for DP-FedAvg with local fine tuning (Yu et al., 2020); PPSGD proposed by Bietti et al. (2022); and PMTL-ft which stands for PMTL proposed by Hu et al. (2021) with local fine tuning. Note that Stand-alone-no-FL does not involve any global communications, therefore no privacy mechanism is added to its implementation. This makes Stand-alone-no-FL a strong baseline as the utility of all included differentially private competing methods are affected by gradient clipping and noise injection, especially when the DP budget is small, e.g. $\epsilon_{dp} = 1$. Another advantage of Stand-alone-no-FL setting is that, the local stand-alone models are highly flexible, i.e. the model on one client and be completely different from the one on others. On the contrary, while the models of all other non-local methods share a common representation part, which takes up the major portion of the whole model.

We focus on the task of image classification and conduct experiments on three representative datasets, namely CIFAR10, CIFAR100, and EMNIST. In terms of architecture of the neural network, we use LeNet for CIFAR10/CIFAR100 and use MLP for EMNIST, as commonly used in the federated learning literature, the details of which are discussed in Appendix. In terms of data augmentation, we do not perform any data augmentation for training the representation, as we observe that classic data augmentation for DP training leads to worse accuracy, as also reported in De et al. (2022). We also tried a new type of data augmentation suggested by De et al. (2022), which does not consistently improve the classification (validation) accuracy in our experiments neither.

**CENTAUR Has the Best Privacy Utility Trade-off.** We first present the utility (testing accuracy) of models trained with CENTAUR and other baselines algorithms under a fixed small privacy budget $\epsilon_{dp} = 1$, for a variety of heterogeneous FL settings. To simulate the data-heterogeneity phenomenon ubiquitous in the research of federated learning, we follow the data allocation scheme of (Collins et al., 2021): Specifically, we first split the original dataset into a training part (90%) and a validation part (10%) and we then allocate the training part equally to $n$ clients while ensuring that each client has at most data from $S$ classes. In Table 1, we observe that, under this small privacy budget, our proposed CENTAUR enjoy better performance than all the included baseline algorithms. Importantly, CENTAUR is **the only method** that consistently **outperforms the strong local-only baseline**, and therefore justifies the choice of collaborative learning as opposed of local stand-alone training. Finally, we further demonstrate that CENTAUR enables superior privacy utility trade-off uniformly across different privacy budget $\epsilon$, for the setting of EMNIST dataset ($n = 2000, S = 5$) in Figure 1.

**Conclusion.** In this work, we point out that the phenomenon of statistical heterogeneity, one of the major challenges of federated learning, introduces extra difficulty when DP constraints are imposed. To alleviate this difficulty, we consider the federated representation learning where only the representation function is to be globally shared and trained. We provide a rigorous guarantee for the utility-privacy trade-off of the proposed CENTAUR method in the linear representation setting, which is $O(\sqrt{d})$ better than the SOTA result. We also empirically show that CENTAUR provides better utility uniformly on several vision datasets under various data heterogeneous settings.

ACKNOWLEDGEMENT

The work of Zebang Shen was supported by NSF-CPS-1837253. Hamed Hassani acknowledges the support from the NSF Institute for CORE Emerging Methods in Data Science (EnCORE), under award CCF-2217058. The research of Reza Shokri is supported by Google PDPO faculty research award, Intel within the www.private-ai.org center, Meta faculty research award, the NUS Early Career Research Award (NUS ECRA award number NUS ECRA FY19 P16), and the National Research Foundation, Singapore under its Strategic Capability Research Centres Funding Initiative. Any opinions, findings and conclusions or recommendations expressed in this material are those of the author(s) and do not reflect the views of National Research Foundation, Singapore. In addition, Zebang Shen thanks Prof. Hui Qian from Zhejiang University for providing the computational resource, who is supported by National Key Research and Development Program of China under Grant 2020AAA0107400. The authors would like to thank Hongyan Chang for helpful discussions on earlier stages of this paper.

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

## A  More on Preliminaries

### A.1  Subgaussian Random Variable

Let $x \in \mathbb{R}$ be a subgaussian random variable. We use $\|x\|_{\psi_2} = \inf\{s > 0 | \mathbb{E}[\exp(x^2/s^2)] \leq 2\}$ to denote subgaussian norm and say $x$ is $\|x\|_{\psi_2}$-subgaussian. For a random vector $\mathbf{x} \in \mathbb{R}^d$, we say $\mathbf{x}$ is subgaussian if $\langle \boldsymbol{x}, \mathbf{x} \rangle$ is subgaussian for any weight vector $\boldsymbol{x} \in \mathbb{R}^d$ and the subgaussian norm of the random vector $\mathbf{x}$ is defined as $\|\mathbf{x}\|_{\psi_2} = \sup_{\boldsymbol{x} \in \mathbb{R}^d, \|\boldsymbol{x}\|_2 = 1} \|\langle \boldsymbol{x}, \mathbf{x} \rangle\|_{\psi_2}$. We refer to $x$ as $\|x\|_{\psi_2}$-subgaussian.

### A.2  A Useful Identity

Note that $\boldsymbol{I}_k = \boldsymbol{B}^\top [\boldsymbol{B}^* \ \boldsymbol{B}_\perp^*][\boldsymbol{B}^* \ \boldsymbol{B}_\perp^*]^\top \boldsymbol{B} = \boldsymbol{B}^\top \boldsymbol{B}^* \boldsymbol{B}^{*\top} \boldsymbol{B} + \boldsymbol{B}^\top \boldsymbol{B}_\perp^* \boldsymbol{B}_\perp^{*\top} \boldsymbol{B}$ and hence

$$
\begin{aligned}
s_{\min}^2(\boldsymbol{B}^\top \boldsymbol{B}^*) = s_{\min}(\boldsymbol{B}^\top \boldsymbol{B}^* \boldsymbol{B}^{*\top} \boldsymbol{B}) &= \min_{\|\boldsymbol{a}\|=1} \boldsymbol{a}^\top \boldsymbol{B}^\top \boldsymbol{B}^* \boldsymbol{B}^{*\top} \boldsymbol{B} \boldsymbol{a} \\
&= \min_{\|\boldsymbol{a}\|=1} \boldsymbol{a}^\top \left( \boldsymbol{I}_k - \boldsymbol{B}^\top \boldsymbol{B}_\perp^* \boldsymbol{B}_\perp^{*\top} \boldsymbol{B} \right) \boldsymbol{a} \\
&= 1 - s_{\max}(\boldsymbol{B}^\top \boldsymbol{B}_\perp^* \boldsymbol{B}_\perp^{*\top} \boldsymbol{B}) = 1 - (\operatorname{dist}(\boldsymbol{B}^*, \boldsymbol{B}))^2. \quad (5)
\end{aligned}
$$

### A.3  More on Privacy Preliminaries

**Lemma A.1** (Gaussian Mechanism of RDP). *Let $R_\alpha$ be the Rényi divergence defined in Definition 2.1, we have $R_\alpha(\mathcal{N}(0, \sigma^2), \mathcal{N}(\mu, \sigma^2)) = \alpha \mu^2/(2\sigma^2)$. Here $\mathcal{N}$ stands for the standard Gaussian distribution.*

**Lemma A.2** (Composition of RDP). *Recall the definition of $\mathcal{D}$ in Definition 2.1 and let $R_1$ and $R_2$ be some abstract space. Let $\mathcal{M}_1 : \mathcal{D} \to R_1$ and $\mathcal{M}_1 : \mathcal{D} \times R_1 \to R_2$ be $(\alpha, \epsilon_1)$-RDP and $(\alpha, \epsilon_2)$-RDP respectively, then the mechanism defined as $(X, Y)$, where $X \sim \mathcal{M}_1(\mathcal{S})$ and $Y \sim \mathcal{M}_2(\mathcal{S}, X)$, satisfies $(\alpha, \epsilon_1 + \epsilon_2)$-RDP.*

**Definition A.1** (DP). *Let $\Theta$ be an abstract output space. A randomized algorithm $\mathcal{M} : \mathcal{D} \to \Theta$ is $(\epsilon, \delta)$-differential private if for all $\mathbb{D}, \mathbb{D}' \in \mathcal{D}$ with $d(\mathbb{D}, \mathbb{D}') \leq 1$, we have that for all subset of the range, $S \subseteq \Theta$, the algorithm $\mathcal{M}$ satisfies: $\Pr\{\mathcal{M}(\mathbb{D}) \in S\} \leq \exp(\epsilon) \Pr\{\mathcal{M}(\mathbb{D}') \in S\} + \delta$.*

**Theorem A.1** (Conversion from RDP to DP). *If $\mathcal{M}$ is an $(\alpha, \epsilon)$-RDP mechanism, it also satisfies $(\epsilon + \frac{\log 1/\delta}{\alpha - 1}, \delta)$-differential privacy for any $0 < \delta < 1$.*

### A.4  Proof of Theorem 5.2

*Proof.* Recall the definition of the Gaussian mechanism

$$
\mathrm{GM}_{\zeta, \sigma}(\{\boldsymbol{x}_i\}_{i=1}^s) \doteq \frac{1}{s}\left(\sum_{i=1}^s \operatorname{clip}(\boldsymbol{x}_i; \zeta) + \sigma \zeta W\right)
$$

| $\epsilon_{dp}$ | Stand-alone-no-FL | CENTAUR | DP-FedAvg-fb | PPSGD | PMTL-ft |
|---|---|---|---|---|---|
| 0.25 | no clip | 0.01 | 0.01 | 0.01 | 0.01 |
| 0.5 | no clip | 0.01 | 0.01 | 0.01 | 0.04 |
| 1 | no clip | 0.01 | 0.04 | 0.01 | 0.04 |
| 2 | no clip | 0.04 | 0.06 | 0.04 | 0.04 |
| 4 | no clip | 0.04 | 0.06 | 0.04 | 0.06 |

Table 2: Clipping threshold $\zeta_g$ to reproduce the results in Figure 1.

where $W \sim \mathcal{N}(0, \boldsymbol{I})$. Lemma A.1 states that $\text{GM}_{\zeta_g, \sigma_g}$ is a $(\alpha, 2\alpha/\sigma_g^2)$-RDP mechanism for $\alpha \geq 1$ (the sensitivity is $2\zeta_g/n$ while the variance of the noise is $(\sigma_g \zeta_g/n)^2$). Using the composition of RDP again over all the iterates $t \in [T]$, we obtain that Algorithm 1 is an $(\alpha, \epsilon_{init} + \frac{2\alpha T_g}{\sigma_g^2})$-RDP mechanism.

$\square$

## B   DETAILS ON EXPERIMENTS SETUP AND MORE RESULTS

**Models.**   We use LeNet-5 for the datasets CIFAR10 and CIFAR100. LeNet-5 consists of two convolution layers with (64, 64) channels and two hidden fully-connected layers. For CIFAR10, the number of hidden neurons are (384, 32) while for CIFAR100, the number of hidden neurons are (128, 32). We use ReLU for activation. No batch normalization or dropout layer is used.
We use MLP for experiments on EMNIST. It consists of three hidden layers with size (256, 128, 16). We use ReLU for activation. No batch normalization or dropout layer is used.

**Hyperparameters.**   All of our experiments are conducted in the fully participating setting, i.e. $p_c = 1$. According to our experiments, the following hyperparameters are most important to the performance of CENTAUR: the clipping threshold of the Gaussian mechanism $\zeta_g$, the global step size $\eta_g$, the local step size $\eta_l$, the number of global rounds $T_g$.
For CIFAR10, to reproduce the utility-privacy tradeoff presented in Figure 1, we grid search the clipping threshold $\zeta_g$ in the set $\{0.01, 0.02, 0.04, 0.06\}$ for every combination of privacy budget and baseline. The resulting optimal clipping threshold is listed in the Table 2. For other parameters, we set $\eta_l = 0.01, T_g = 200$ uniformly.

To reproduce the utility results in Table 1, for CIFAR10, we uniformly set $\zeta_g = 0.01, \eta_l = 0.01, \eta_g = 1, T_g = 200$; for CIFAR100, we uniformly set $\zeta_g = 0.02, \eta_l = 0.01, T_g = 100, \eta_g = 1$. Note that once the privacy budget $\epsilon_{dp}$ is given, we use the privacy engine from the package of Opacus to determine the noise multiplier $\sigma_g$, given $T_g$.
For EMNIST, we uniformly set $\zeta_g = 0.25, \eta_l = 0.01, T_g = 40, \eta_g = 1$.

There are also some algorithm-specific parameters: For PPSGD, we set the step size for the local correction to $\eta = 0.1$ and set ratio between the global and the local step size to $\alpha = 0.1$. For PMTL-ft, we set $\lambda$, the regularization parameter to 1.

For baselines that require local fine tuning, we perform 15 local epochs to fine tune the local head with a fixed step size of $0.01$.

**About data augmentation.**   In the Non-DP setting, the technique of data augmentation usually significantly improves the testing accuracy in CV tasks. However, in the DP setting, as reported in the previous work De et al. (2022), directly utilizing data augmentation leads to inferior performance. In the same work, the authors proposed an alternative version of the data augmentation technique which would improve the testing accuracy on various CV tasks in the centralized DP training setting. We tried their strategy in the federated representation learning setting under consideration, which however does not improve the utility in our case.

On the other hand, since the fine tuning of the local classification head does not require DP protection (recall that in CENTAUR, the head is kept private), we employed the standard data augmentation in this phase (optimizing over the local classification head), which improves the testing accuracy for

Table 3: Testing accuracy (%) on CIFAR10 with various data allocation settings given larger communication budget $T_g = 400$. No data augmentation is used for training the representations. $n$ stands for the number clients and $S$ stands for the number classes per client. The $\delta$ parameter of DP is fixed to $10^{-5}$ as a common choice in the literature. The budget parameter of DP is fixed to a small value of $\epsilon_{dp} = 1$ for results in this table.

| Methods | Stand-alone-no-FL | DP-FedAvg-fb | PMTL-ft | PPSGD | CENTAUR |
|---|---|---|---|---|---|
| CIFAR10, n=1000,S=2 | 74.06 (0.45) | 63.97 (0.98) | 67.71 (0.78) | 74.63 (0.76) | **77.80 (0.52)** |
| CIFAR10, n=1000,S=5 | 44.60 (1.30) | 41.12 (0.40) | 45.75 (0.81) | 48.29 (1.79) | **51.05 (0.35)** |

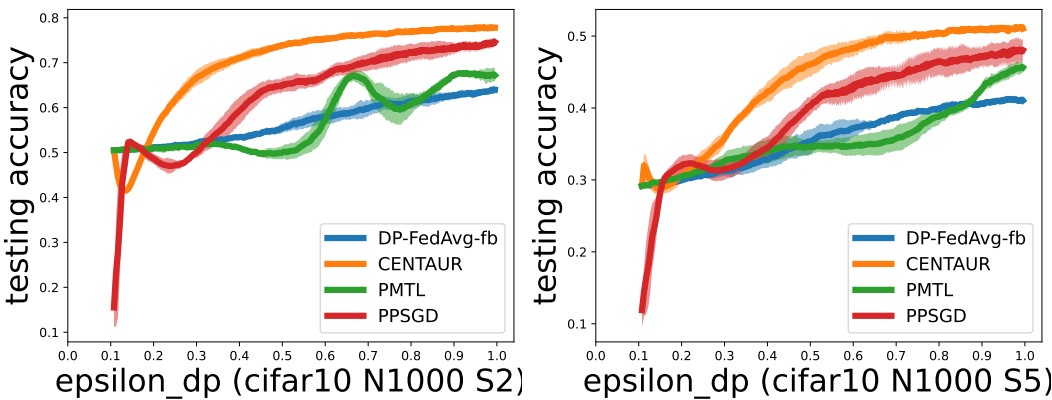

Figure 3: Testing accuracy vs $\epsilon_{dp}$ during training.

CENTAUR. We also tried the same technique for the fine tuning part of other baselines, but it actually leads to worse performance. Hence in the reported results, data augmentation is only used for the fine tuning of the local classification head in CENTAUR and is not used in any other cases.

### B.1 MORE EMPIRICAL RESULTS

To study the performance of different baselines given a larger communication budget, i.e. a larger $T_g$, we conduct additional experiments on CIFAR10 and report the results in Table 3. We can observe that CENTAUR has the best performance among all the included methods, uniformly in all configurations. In Figure 3, we further show the testing accuracy (utility) vs the privacy cost $\epsilon_{dp}$ during the training. We observe that CENTAUR quickly converges to a high utility can consistently outperforms the included baselines.

## C  UTILITY ANALYSIS OF THE CENTAUR INSTANCE FOR THE LRL CASE

To present the utility analysis of the CENTAUR instance for the LRL case, Problem (L-FRL) is equivalently formulated as a standard matrix sensing problem. By setting the clipping threshold $\zeta_g$ to a high probability upper bound of the norm of the local gradient $\boldsymbol{G}_i^t$ (see Lemma C.1), we show that CENTAUR can be regarded as an inexact gradient descent method. Given that $\bar{m}$, the mini-batch size parameter, is sufficiently large and that the initializer $\boldsymbol{B}^0$ is sufficiently close the ground truth $\boldsymbol{B}^*$, we establish a high probability one-step contraction lemma that controls the utility $\mathrm{dist}(\boldsymbol{B}^t, \boldsymbol{B}^*)$, which directly leads to the main utility theorem 5.1.

**Matrix Sensing Formulation**  Consider a reparameterization[4] of the local classifier $\boldsymbol{w}_i = \sqrt{n}\boldsymbol{v}_i$. Problem (L-FRL) can be written as

$$\min_{B^\top B = \boldsymbol{I}_k} \frac{1}{n}\sum_{i=1}^n \min_{\boldsymbol{v}_i \in \mathbb{R}^k} \left( \frac{1}{m}\sum_{j=1}^m \frac{1}{2}(\sqrt{n}\langle \boldsymbol{v}_i, \boldsymbol{B}^\top \boldsymbol{x}_{ij}\rangle - y_{ij})^2 \right). \tag{6}$$

---

[4]We consider this rescaling of $\boldsymbol{w}_i$ so that the corresponding linear operator in equation 8 is an isometric operator in expectation (see more discussion below).

Further, denote $\boldsymbol{V} \in \mathbb{R}^{n \times k}$ the collection of local classifiers, $\boldsymbol{V}_{i,:} = \boldsymbol{v}_i$. The collection of optimal local classifier heads $\boldsymbol{W}^*$ can also be rescaled as $\boldsymbol{W}^* = \sqrt{n} \boldsymbol{V}^*$ and the responses $y_{ij} \in \mathbb{R}$ satisfy

$$y_{ij} = \sqrt{n} \langle \boldsymbol{B}^{*\top} \boldsymbol{x}_{ij}, \boldsymbol{V}_{i,:}^* \rangle. \tag{7}$$

Define the rank-1 matrices $\boldsymbol{A}_{ij} = \boldsymbol{x}_{ij} \boldsymbol{e}^{(i)\top} \in \mathbb{R}^{d \times n}$ ($\boldsymbol{e}^{(i)} \in \mathbb{R}^n$) and define the operators $\mathcal{A}_i :$ $\mathbb{R}^{d \times n} \to \mathbb{R}^m$ and $\mathcal{A} : \mathbb{R}^{d \times n} \to \mathbb{R}^N$

$$\mathcal{A}_i(\boldsymbol{X}) = \{\sqrt{n} \langle \boldsymbol{A}_{ij}, \boldsymbol{X} \rangle\}_{j \in [m]} \in \mathbb{R}^m, \mathcal{A}(\boldsymbol{X}) = \{\mathcal{A}_i(\boldsymbol{X})\}_{i \in [n]} \in \mathbb{R}^N, \tag{8}$$

where we use $N = nm$ to denote the total number of data points globally. Note that $\frac{1}{\sqrt{N}} \mathcal{A}$ is an isometric operator in expectation w.r.t. the randomness of $\{\boldsymbol{x}_{ij}\}$, i.e. for any $\boldsymbol{X} \in \mathbb{R}^{d \times n}$

$$\mathbb{E}_{\{\boldsymbol{x}_{ij}\}}[\langle \frac{1}{\sqrt{N}} \mathcal{A}(\boldsymbol{X}), \frac{1}{\sqrt{N}} \mathcal{A}(\boldsymbol{X}) \rangle] = \frac{1}{N} \sum_{i=1}^n \sum_{j=1}^m n \boldsymbol{e}^{(i)\top} \boldsymbol{X}^\top \mathbb{E}_{\boldsymbol{x}_{ij}}[\boldsymbol{x}_{ij} \boldsymbol{x}_{ij}^\top] \boldsymbol{X} \boldsymbol{e}^{(i)}$$

$$= \frac{1}{N} \sum_{i=1}^n \sum_{j=1}^m n \boldsymbol{X}_{:,i}^\top \boldsymbol{X}_{:,i} = \frac{1}{N} \sum_{i=1}^n \sum_{j=1}^m n \|\boldsymbol{X}_{:,i}\|^2 = \|\boldsymbol{X}\|_F^2,$$

where we use Assumption 5.1 in the second equality. With the notations defined above, we can rewrite Problem (L-FRL) as a standard matrix sensing problem with the operator $\mathcal{A}$

$$\min_{\boldsymbol{B}^\top \boldsymbol{B} = \boldsymbol{I}_k} \min_{\boldsymbol{V} \in \mathbb{R}^{n \times k}} \mathcal{F}(\boldsymbol{B}, \boldsymbol{V}; \mathcal{A}) = \frac{1}{2N} \|\mathcal{A}(\boldsymbol{B} \boldsymbol{V}^\top) - \mathcal{A}(\boldsymbol{B}^* \boldsymbol{V}^{*\top})\|^2. \tag{MSP}$$

Since CENTAUR only uses a portion of the data points from $\mathbb{S}_i$ to compute the local gradient $\boldsymbol{G}_i^t$ (see line 2 in Algorithm 4), it is useful to define the operators $\mathcal{A}_i^{t,1}$ and $\mathcal{A}_i^{t,2}$ that corresponds to $\mathbb{S}_i^{t,1}$ and $\mathbb{S}_i^{t,2}$ respectively, and their globally aggregated versions $\mathcal{A}^{t,1}$ and $\mathcal{A}^{t,2}$:

$$\mathcal{A}_i^{t,l}(\boldsymbol{X}) = \{\sqrt{n} \langle \boldsymbol{x}_{ij} \boldsymbol{e}^{(i)\top}, \boldsymbol{X} \rangle\}_{j \in \mathbb{S}_i^{t,l}} \in \mathbb{R}^{\bar{m}}, \mathcal{A}^{t,l}(\boldsymbol{X}) = \{\mathcal{A}_i^{t,l}(\boldsymbol{X})\}_{i \in [n]} \in \mathbb{R}^{\bar{N}}, l = 1, 2, \tag{9}$$

where we denote $\bar{N} \doteq \bar{m} n$.

**Clippings are inactive with a high probability**  The following lemma shows that by properly setting the clipping thresholds $\zeta_g$, the clipping step of the Gaussian mechanism in Algorithm 4 takes no effect with a high probability.

**Lemma C.1.** *Consider the LRL setting. Under the assumptions 5.2 and 5.1, we have with a probability at least $1 - \bar{m} n^{-k}$,*
$$\|\boldsymbol{G}_i^t\|_F \leq \zeta_g \doteq c_\zeta \mu^2 k s_k^2 \sqrt{dk \log n},$$
*where $\boldsymbol{G}_i^t$ is computed in line 4 of Algorithm 4 and $\zeta_g$ is some universal constant.*

*Proof.* The detailed expression of $\boldsymbol{G}_i^t$ in line 4 of Algorithm 4 can be calculated as follows:

$$\boldsymbol{G}_i^t = \partial_{\boldsymbol{B}} l([\boldsymbol{w}_i^{t+1}, \boldsymbol{B}^t]; \mathbb{S}_i^{t,2}) = \frac{1}{\bar{m}} \sum_{(\boldsymbol{x}_{ij}, y_{ij}) \in \mathbb{S}_i^{t,2}} \left( \langle \boldsymbol{B}^{t\top} \boldsymbol{x}_{ij}, \boldsymbol{w}_i^{t+1} \rangle - y_{ij} \right) \boldsymbol{x}_{ij} \boldsymbol{w}_i^{t+1\top}. \tag{10}$$

Using the triangle inequality of the matrix norm, $\zeta$ is a high probability upper bound of $\|\boldsymbol{G}_i^t\|_F$ if the inequality

$$\| \left( \langle \boldsymbol{B}^{t\top} \boldsymbol{x}_{ij}, \boldsymbol{w} \rangle - y_{ij} \right) \boldsymbol{x}_{ij} \boldsymbol{w}_i^{t+1\top} \|_F \leq \zeta \tag{11}$$

holds with a high probability. In the following, we show that the inequalities $|\langle \boldsymbol{B}^{t\top} \boldsymbol{x}_{ij}, \boldsymbol{w} \rangle| \leq \zeta_1$, $|y_{ij}| \leq \zeta_y$, $\|\boldsymbol{x}_{ij}\|_2 \leq \zeta_x$ and $\|\boldsymbol{w}_i^{t+1}\| \leq \zeta_w$ hold jointly with probability at least $1 - 5n^{-k}$, which together with $(\zeta_y + \zeta_1) \zeta_x \zeta_w \leq \zeta_g$ and the union bound leads to the result of the lemma.

**Choice of $\zeta_y$**  Recall that in Assumption 5.1, we assume that $\boldsymbol{x}_{ij}$ is a sub-Gaussian random vector with $\|\boldsymbol{x}_{ij}\|_{\psi_2} = 1$. Using the definition of a sub-Gaussian random vector, we have

$$\mathbb{P}\{|y_{ij}| \geq \zeta_y\} \leq 2 \exp(-c_s \zeta_y^2 / \|\boldsymbol{w}_i^*\|^2) \leq \exp(-k \log n), \tag{12}$$

with the choice $\zeta_y = \mu \sqrt{k} s_k \cdot \sqrt{(k \log n + \log 2)/c_s} = O(\mu s_k k \sqrt{\log n})$ since $\|\boldsymbol{w}_i^*\|_2 \leq \mu \sqrt{k} s_k$. Here $c_s$ is some constant and we recall that $s_k$ is a shorthand for $s_k(\boldsymbol{W}^*/\sqrt{n})$.

**Choice of $\zeta_x$**  Recall that $x_{ij}$ is a sub-Gaussian random vector with $\|x_{ij}\|_{\psi_2} = 1$ and therefore with probability at least $1 - \delta$,

$$\|x_{ij}\|_2 \leq 4\sqrt{d} + 2\sqrt{\log \frac{1}{\delta}}. \tag{13}$$

Therefore by taking $\delta = \exp(-k \log n)$, we have that $\zeta_x = 4\sqrt{d} + 2\sqrt{\log \frac{1}{\delta}} = O(\sqrt{d})$.

**Choice of $\zeta_w$**  We can show that $\zeta_w = 2\mu\sqrt{k}s_k$ is a high probability upper bound of $\|w_i^{t+1}\|$. Proving this bound requires detailed analysis of FedRep and is discussed later in equation 63.

**Choice of $\zeta_1$**  The following event is conditioned on the event $\|w_i^{t+1}\|_2 \leq 2\mu\sqrt{k}s_k$. We will then bound the probability of both events happen simultaneously using the union bound. Since $x_{ij}$ is a sub-Gaussian random vector with $\|x_{ij}\|_{\psi_2} = 1$, using the definition of a sub-Gaussian random vector, we have

$$\mathbb{P}\{|\langle B^t w_i^{t+1}, x_{ij}\rangle| \geq \zeta_1\} \leq 2\exp(-c_s \zeta_1^2 / \|w_i^{t+1}\|^2) \leq \exp(-k \log n), \tag{14}$$

with the choice $\zeta_1 = 2\mu\sqrt{k}s_k \cdot \sqrt{(k \log n + \log 2)/c_s} = O(\mu s_k k\sqrt{\log n})$ since $\|w_i^{t+1}\|_2 \leq 2\mu\sqrt{k}s_k$. Here $c_s$ is some constant and we recall that $s_k$ is a shorthand for $s_k(W^*/\sqrt{n})$. Using the union bound, we have that w.p. at least $1 - 2\exp(-k \log n)$, the upper bound $\zeta_1$ is valid. $\qquad\square$

**The idea behind the proof**  To present the intuition of the utility analysis of CENTAUR, define

$$V(B; \mathcal{A}) = \arg\min_{V \in \mathbb{R}^{n \times k}} \mathcal{F}(B, V; \mathcal{A}), \tag{15}$$

where $\mathcal{A}$ is some matrix sensing operator and $\mathcal{F}$ is defined in Problem (MSP). Under the event that the clipping steps in the Gaussian mechanism in Algorithm 4 takes no effect, the average gradient $G^t \doteq \frac{1}{n}\sum_{i=1}^n G_i^t$ admits the following compact form

$$G^t = \partial_B \mathcal{F}(B^t, V(B^t; \mathcal{A}^{t,1}); \mathcal{A}^{t,2}). \tag{16}$$

Suppose that $\mathcal{A}^{t,1} \simeq \mathcal{A}^{t,2} \simeq \mathcal{A}$ (recall that all these linear operators are comprised of i.i.d. data points $x_{ij}$ and are hence similar when $m$ is large). Further define the objective

$$\mathcal{G}(B; \mathcal{A}) = \min_{V \in \mathbb{R}^{n \times k}} \mathcal{F}(B, V; \mathcal{A}). \tag{17}$$

We have $G^t \simeq \nabla\mathcal{G}(B^t; \mathcal{A})$ since

$$\nabla\mathcal{G}(B^t; \mathcal{A}) = \partial_B \mathcal{F}(B^t, V(B^t); \mathcal{A}) + \mathcal{J}_V(B^t)^\top \partial_V \mathcal{F}(B^t, V(B^t); \mathcal{A}) = \partial_B \mathcal{F}(B^t, V(B^t; \mathcal{A}); \mathcal{A}),$$

where $\mathcal{J}_V(B)$ denotes the Jacobian matrix of $V(B; \mathcal{A})$ with respect to $B$ and the second equality holds due to the optimality of $V(B^t; \mathcal{A})$. Consequently, conditioned on the event that all the clipping operation are inactive, CENTAUR behaves similar to the noisy gradient descent on the objective $\mathcal{G}(B; \mathcal{A})$ (up to the difference between $\mathcal{A}^{t,1}$, $\mathcal{A}^{t,1}$, and $\mathcal{A}$). In the following, we show that while the objective $\mathcal{G}(B; \mathcal{A})$ is non-convex globally, but we can show that CENTAUR converges locally within a region around the underlying ground truth.

**One-step contraction**  To present our theory, we first establish the following properties that the operators $\mathcal{A}^{t,1}$ and $\mathcal{A}^{t,2}$ (defined in equation 9) satisfy. Recall that $\bar{N} = \bar{m}n = \sum_{i=1}^n |\mathcal{S}_i^{t,1}|$. The proofs are deferred to Appendix E.

**Lemma C.2.**  *Under Assumption 5.1, the linear operator $\mathcal{A}^{t,1}$ satisfies the following property with probability at least $1 - \exp(-c_1 k \log n)$:*

$$\sup_{V \in \mathbb{R}^{n \times k}, \|V\|_F = 1} |\frac{1}{\bar{N}}\langle \mathcal{A}^{t,1}(B^t V^\top), \mathcal{A}^{t,1}(B^t V^\top)\rangle - 1| \leq \delta^{(1)}.$$

*Here, the factor $\delta^{(1)} = \sqrt{k \log n}/\sqrt{\bar{m}}$.*

**Lemma C.3.** *Under Assumption 5.1, the linear operator $\mathcal{A}^{t,1}$ satisfies the following property with probability at least $1 - \exp(-c_2 k \log n)$: For $\boldsymbol{V}_1, \boldsymbol{V}_2 \in \mathbb{R}^{n \times k}$,*

$$\sup_{\|\boldsymbol{V}_1\|_F = \|\boldsymbol{V}_2\|_F = 1} |\frac{1}{N}\langle \mathcal{A}^{t,1}((\boldsymbol{B}^t \boldsymbol{B}^{t\top} - \boldsymbol{I}_d)\boldsymbol{B}^* \boldsymbol{V}_1^\top), \mathcal{A}^{t,1}(\boldsymbol{B}^t \boldsymbol{V}_2^\top)\rangle| \leq \delta^{(2)}\mathrm{dist}(\boldsymbol{B}^t, \boldsymbol{B}^*).$$

*Here, the factor $\delta^{(2)} = \sqrt{k \log n}/\sqrt{\bar{m}}$.*

**Lemma C.4.** *Under Assumptions 5.1 and 5.2, the linear operator $\mathcal{A}^{t,2}$ satisfies the following property with probability at least $1 - \exp(-c_3 k \log n)$: For $\boldsymbol{a} \in \mathbb{R}^d, \boldsymbol{b} \in \mathbb{R}^k$*

$$\sup_{\|\boldsymbol{a}\| = \|\boldsymbol{b}\| = 1} |\frac{1}{N}\langle \mathcal{A}^{t,2}(\boldsymbol{B}^t \boldsymbol{V}^{t+1\top} - \boldsymbol{B}^* \boldsymbol{V}^{*\top}), \mathcal{A}^{t,2}(\boldsymbol{a}\boldsymbol{b}^\top \boldsymbol{V}^{t+1\top})\rangle| \leq \delta^{(3)} s_1^2 k\, \mathrm{dist}(\boldsymbol{B}^t, \boldsymbol{B}^*).$$

*Here, the factor $\delta^{(3)} = 4(\sqrt{d} + \sqrt{k \log n})/\left(\sqrt{\bar{m}n}\kappa^2\right)$.*

We now present the one-step contraction lemma of `CENTAUR` in the LRL setting. The proof can be found in Appendix D.1.

**Lemma C.5** (One-step contraction). *Consider the instance of `CENTAUR` for the LRL setting with its* CLIENT *and* INITIALIZATION *procedures defined in Algorithms 4 and 3 respectively. Suppose that the matrix $\boldsymbol{B}^0$ returned by* INITIALIZATION *satisfies $\mathrm{dist}(\boldsymbol{B}^0, \boldsymbol{B}^*) \leq \epsilon_0 = 0.2$ and suppose that the mini-batch size parameter satisfies*

$$\bar{m} \geq c_m \max\{\kappa^2 k^2 \log n, \frac{k^2 d + k^3 \log n}{n}\},$$

*for some universal constant $c_m$. Set the clipping threshold of the Gaussian mechanism $\zeta_g$ according to Lemma C.1 and set the global step size $\eta_g = 1/4s_1^2$. Suppose that the level of manually injected noise is sufficiently small: For some universal constant $c_\sigma$, it satisfies*

$$\sigma_g \leq \frac{c_\sigma n}{\mu^2(\sqrt{d} + \sqrt{k \log n})} \min\left(\frac{1}{k^2 \log n}, \frac{\kappa^4}{\sqrt{dk \log n}}\right). \tag{18}$$

*We have the following one-step contraction from a single iteration of `CENTAUR`*

$$\mathrm{dist}(\boldsymbol{B}^*, \boldsymbol{B}^{t+1}) \leq \mathrm{dist}(\boldsymbol{B}^*, \boldsymbol{B}^t)\sqrt{1 - E_0/8\kappa^2} + 3C_\mathcal{N}\frac{\eta_g \sigma_g \zeta_g}{n}\sqrt{d}, \tag{19}$$

*holds with probability at least $1 - c_p \bar{m}n^{-k}$, where $c_p$ is some universal constant. Moreover, with the same probability, we also have*

$$\mathrm{dist}(\boldsymbol{B}^*, \boldsymbol{B}^t) \leq \mathrm{dist}(\boldsymbol{B}^*, \boldsymbol{B}^0). \tag{20}$$

**Remark C.1.** *The lower bound of the mini-batch size parameter $\bar{m}$ is derived to satisfy the following inequalities:*

$$\max\{\frac{\delta^{(2)}\sqrt{k}}{1 - \delta^{(1)}}, \frac{(\delta^{(2)})^2 k}{(1 - \delta^{(1)})^2}, \delta^{(3)} k\} \leq \frac{s_k^2(1 - \epsilon_0^2)}{36s_1^2},$$

*which is required to establish the above one-step contraction lemma. The upper bound of the noise multiplier $\sigma_g$ is derived to satisfy the following inequalities:*

$$\frac{\eta_g \sigma_g \zeta_g}{n} \leq \min\left(\frac{\sqrt{1 - \epsilon_0^2}}{4C_\mathcal{N}\kappa^2\sqrt{k \log n}}, \frac{8\kappa^2\epsilon_0}{3C_\mathcal{N}\sqrt{dE_0}}\right). \tag{21}$$

*Proof of Theorem 5.1.* Denote $E_0 = \sqrt{1 - \epsilon_0^2}$. Using the recursion (19), we have

$$\mathrm{dist}(\boldsymbol{B}^*, \boldsymbol{B}^t) \leq (1 - E_0/8\kappa^2)^{t/2}\mathrm{dist}(\boldsymbol{B}^*, \boldsymbol{B}^0) + 3C_\mathcal{N}\frac{\eta_g \sigma_g \zeta_g}{n}\sqrt{d}/(1 - \sqrt{1 - E_0/8\kappa^2})$$

$$\leq (1 - E_0/8\kappa^2)^{t/2}\mathrm{dist}(\boldsymbol{B}^*, \boldsymbol{B}^0) + 48C_\mathcal{N}\kappa^2\frac{\eta_g \sigma_g \zeta_g}{n}\sqrt{d}/E_0. \tag{22}$$

With the choice of $T$ specified in the theorem, we have that

$$\mathrm{dist}(\boldsymbol{B}^*, \boldsymbol{B}^t) \leq c_d \kappa^2 \frac{\eta_g \sigma_g \zeta_g}{n}\sqrt{d}, \tag{23}$$

for some universal constant $c_d$. By plugging the choices of $\eta_g$, $\zeta_g$, we obtain the simplified bound

$$\mathrm{dist}(\boldsymbol{B}^*, \boldsymbol{B}^t) \leq \tilde{c}_d \sigma_g \mu^2 k^{1.5} d/n,$$

where $\tilde{c}_d$ hides the constant and log terms. $\qquad\square$

# D PROOF OF THE ONE-STEP CONTRACTION LEMMA

## D.1 PROOF OF LEMMA C.5

*Proof.* The following discussion will be conditioned on the event that all the clipping operations are inactive, whose probability is proved to be at least $1 - 5n^{-k}$ in Lemma C.1. Using union bound to combine the following result and Lemma C.1 leads to the result.

Recall that the average gradient $\boldsymbol{G}^t = \frac{1}{n}\sum_{i=1}^{n}\boldsymbol{G}_i^t$ and denote $\boldsymbol{Q}^t = \boldsymbol{B}^t\boldsymbol{V}^{t+1^\top} - \boldsymbol{B}^*\boldsymbol{V}^{*\top}$. Denote

$$\bar{\boldsymbol{B}}^{t+1} = \boldsymbol{B}^t + \eta_g \cdot \boldsymbol{G}^t. \tag{24}$$

Recall that $\mathbb{C}^t = \{1, \ldots, n\}$ given $p_g = 1$. Since the clipping operations are inactive, we have

$$\bar{\boldsymbol{B}}^{t+1} = \boldsymbol{B}^t - \eta_g \boldsymbol{Q}^t \boldsymbol{V}^{t+1} + \eta_g\left(\boldsymbol{Q}^t\boldsymbol{V}^{t+1} - \boldsymbol{G}^t\right) + \frac{\eta_g\sigma_g\zeta_g}{n}\boldsymbol{W}^t, \tag{25}$$

where $\boldsymbol{W}^t$ denotes the noise added by the Gaussian mechanism in the $t^{th}$ global round. Note that $\boldsymbol{B}_\perp^{*\top}\boldsymbol{Q}^t = \boldsymbol{B}_\perp^{*\top}\boldsymbol{B}^t\boldsymbol{V}^{t+1^\top}$, and denote the QR decomposition $\bar{\boldsymbol{B}}^{t+1} = \boldsymbol{B}^{t+1}\boldsymbol{R}^{t+1}$. We have

$$\boldsymbol{B}_\perp^{*\top}\boldsymbol{B}^{t+1}$$

$$= \boldsymbol{B}_\perp^{*\top}\left(\boldsymbol{B}^t - \eta_g\boldsymbol{Q}^t\boldsymbol{V}^{t+1} + \eta_g\left(\boldsymbol{Q}^t\boldsymbol{V}^{t+1} - \boldsymbol{G}^t\right) + \frac{\eta_g\sigma_g\zeta_g}{n}\boldsymbol{W}^t\right)(\boldsymbol{R}^{t+1})^{-1}$$

$$= \boldsymbol{B}_\perp^{*\top}\boldsymbol{B}^t\left(\boldsymbol{I}_k - \eta_g\boldsymbol{V}^{t+1^\top}\boldsymbol{V}^{t+1}\right)(\boldsymbol{R}^{t+1})^{-1}$$

$$+ \eta_g\boldsymbol{B}_\perp^{*\top}\left(\boldsymbol{Q}^t\boldsymbol{V}^{t+1} - \boldsymbol{G}^t\right)(\boldsymbol{R}^{t+1})^{-1} + \frac{\eta_g\sigma_g\zeta_g}{n}\boldsymbol{B}_\perp^{*\top}\boldsymbol{W}^t(\boldsymbol{R}^{t+1})^{-1}.$$

Recall the definition $\text{dist}(\boldsymbol{B}^*, \boldsymbol{B}^t) = \|\boldsymbol{B}_\perp^{*\top}\boldsymbol{B}^t\|$. We bound

$$\text{dist}(\boldsymbol{B}^*, \boldsymbol{B}^{t+1}) \leq \text{dist}(\boldsymbol{B}^*, \boldsymbol{B}^t)\|\boldsymbol{I}_k - \eta_g\boldsymbol{V}^{t+1^\top}\boldsymbol{V}^{t+1}\|\|(\boldsymbol{R}^{t+1})^{-1}\|$$

$$+ \eta\|\boldsymbol{Q}^t\boldsymbol{V}^{t+1} - \boldsymbol{G}^t\|\|(\boldsymbol{R}^{t+1})^{-1}\| + \frac{\eta_g\sigma_g\zeta_g}{n}\|\boldsymbol{B}_\perp^{*\top}\boldsymbol{W}^t\|\|(\boldsymbol{R}^{t+1})^{-1}\|. \tag{26}$$

In the following, we show that the factor $\|\boldsymbol{I}_k - \eta_g\boldsymbol{V}^{t+1^\top}\boldsymbol{V}^{t+1}\|\|(\boldsymbol{R}^{t+1})^{-1}\| < 1$ which leads to a contraction in the principal angle distance and treat the rest two terms as controllable noise for sufficiently small constants $(\delta^{(1)}, \delta^{(2)}, \delta^{(3)})$ (see Lemma C.1) and a sufficiently smaller noise multiplier $\sigma_g$ (see equation 18).

**Bound $\|\boldsymbol{I}_k - \eta_g\boldsymbol{V}^{t+1^\top}\boldsymbol{V}^{t+1}\|$.** Recall that $\eta_g = 1/4s_1^2$. Using Lemma D.5, we have

$$\|\boldsymbol{I}_k - \eta_g\boldsymbol{V}^{t+1^\top}\boldsymbol{V}^{t+1}\| \leq 1 - \eta_g\left(E_0 s_k^2 - \frac{2\delta^{(2)}s_1^2\sqrt{k}}{1-\delta^{(1)}}\text{dist}(\boldsymbol{B}^*, \boldsymbol{B}^t)\right) \leq 1 - \eta_g E_0 s_k^2 \cdot 0.75, \tag{27}$$

where we use the following inequality in Remark C.1

$$\frac{2\delta^{(2)}s_1^2\sqrt{k}}{1-\delta^{(1)}}\text{dist}(\boldsymbol{B}^*, \boldsymbol{B}^t) \leq \frac{2\delta^{(2)}s_1^2\sqrt{k}}{1-\delta^{(1)}} \leq E_0 s_k^2/4. \tag{28}$$

**Bound $\|(\boldsymbol{R}^{t+1})^{-1}\|$.** With the choice of $\bar{m}$ stated in the lemma, the tuple $(\delta^{(1)}, \delta^{(2)}, \delta^{(3)})$ satisfies the requirements in Remark C.1. Using Lemma D.7, we have with probability at least $1 - 4n^{-k}$

$$\|(\boldsymbol{R}^{t+1})^{-1}\| \leq 1/\sqrt{1 - \eta_g s_k^2 E_0/2}. \tag{29}$$

Combining equation 27 and equation 29, we have the contraction

$$\|\boldsymbol{I}_k - \eta_g\boldsymbol{V}^{t+1^\top}\boldsymbol{V}^{t+1}\|\|(\boldsymbol{R}^{t+1})^{-1}\| \leq \frac{1 - \eta_g s_k^2 E_0 \cdot 0.75}{\sqrt{1 - \eta_g s_k^2 E_0 \cdot 0.5}} < 1.$$

We now bound the last two terms of equation 26.

**Bound $\|\boldsymbol{Q}^t\boldsymbol{V}^{t+1} - \boldsymbol{G}^t\|$.**  Using Lemma D.6, we have

$$\|\boldsymbol{Q}^t\boldsymbol{V}^{t+1} - \boldsymbol{G}^t\| \le \delta^{(3)}s_1^2 k\,\mathrm{dist}(\boldsymbol{B}^*, \boldsymbol{B}^t) \le E_0 s_k^2\,\mathrm{dist}(\boldsymbol{B}^*, \boldsymbol{B}^t)\cdot 0.25, \tag{30}$$

where we use condition in Remark C.1 $\delta^{(3)}s_1^2 k \le E_0 s_k^2/4$.

**Bound $\boldsymbol{U_\perp^*}^\top \boldsymbol{W}^t$.**  Due to the rotational invariance of independent Gaussian random variables, every entry in $\boldsymbol{B_\perp^*}^\top \boldsymbol{W}^t \in \mathbb{R}^{(n-k)\times k}$ is distributed as $\mathcal{N}(0,1)$.  Using Lemma I.1, we have with probability at least $1 - n^{-k}$

$$\|\boldsymbol{B_\perp^*}^\top \boldsymbol{W}^t\| \le C_\mathcal{N}(\sqrt{d-k} + \sqrt{k} + \sqrt{\ln(k\log n)}) \le 3C_\mathcal{N}\sqrt{d}, \tag{31}$$

where we assume $d = \Omega(\log\log n)$ to simplify the above bound.

**Final Result.**  Combining the above bounds, we conclude that the following one-step contraction holds under the assumptions stated in the theorem.

$$\mathrm{dist}(\boldsymbol{B}^*, \boldsymbol{B}^{t+1}) \le \mathrm{dist}(\boldsymbol{B}^*, \boldsymbol{B}^t)\frac{1 - \eta_g E_0 s_k^2\cdot(0.75 - 0.25)}{\sqrt{1 - \eta_g E_0 s_k^2\cdot 0.5}} + 3C_\mathcal{N}\frac{\eta_g\sigma_g\zeta_g}{n}\sqrt{d}$$

$$\le \mathrm{dist}(\boldsymbol{B}^*, \boldsymbol{B}^t)\sqrt{1 - \eta_g E_0 s_k^2\cdot 0.5} + 3C_\mathcal{N}\frac{\eta_g\sigma_g\zeta_g}{n}\sqrt{d}$$

$\square$

## D.2   LEMMAS FOR THE UTILITY ANALYSIS

**Lemma D.1.**  *Use $vec(\cdot)$ and $\otimes$ to denote the standard vectorization operation and Kronecker product respectively. Recall that $\boldsymbol{V}^{t+1} := \arg\min_{\boldsymbol{V}\in\mathbb{R}^{n\times k}} \mathcal{F}(\boldsymbol{B}^t, \boldsymbol{V}; \mathcal{A}^{t,1})$. We have*

$$vec(\boldsymbol{V}^{t+1}) = vec((\boldsymbol{B}^*\boldsymbol{V}^{*\top})^\top \boldsymbol{B}^t) - \boldsymbol{f}^t \tag{32}$$

*where*

$$\boldsymbol{f}^t = \left(\boldsymbol{B}^{t\top}\boldsymbol{B}^* \otimes \boldsymbol{I}_d - \boldsymbol{H}^{tt-1}\boldsymbol{H}^{t*}\right)vec(\boldsymbol{V}^*) = \boldsymbol{H}^{tt-1}\left(\boldsymbol{H}^{tt}(\boldsymbol{B}^{t\top}\boldsymbol{B}^* \otimes \boldsymbol{I}_d) - \boldsymbol{H}^{t*}\right)vec(\boldsymbol{V}^*). \tag{33}$$

*Here we denote*

$$\boldsymbol{H}^{t*} = \frac{1}{n}\sum_{i=1}^n \frac{1}{\bar{m}}\sum_{j\in\mathbb{S}_i^{t,1}} vec(\boldsymbol{A}_{ij}^\top\boldsymbol{B}^t)\,vec(\boldsymbol{A}_{ij}^\top\boldsymbol{B}^*)^\top \in \mathbb{R}^{nk\times nk}, \tag{34}$$

$$\boldsymbol{H}^{tt} = \frac{1}{n}\sum_{i=1}^n \frac{1}{\bar{m}}\sum_{j\in\mathbb{S}_i^{t,1}} vec(\boldsymbol{A}_{ij}^\top\boldsymbol{B}^t)\,vec(\boldsymbol{A}_{ij}^\top\boldsymbol{B}^t)^\top \in \mathbb{R}^{nk\times nk}. \tag{35}$$

*We use $\boldsymbol{F}^t \in \mathbb{R}^{n\times k}$ to denote the matrix satisfying $\boldsymbol{f}^t = vec(\boldsymbol{F}^t)$.*

*Proof.* Recall that $\bar{N} = \bar{m}n$. For the simplicity of notations, we use the collection $\{\boldsymbol{A}_l\}$, $l = 1,\dots,\bar{N}$ to denote $\{\boldsymbol{A}_{ij}\}$, $i\in[n]$, $j\in\mathcal{S}_i^{t,1}$ (there exists a one-to-one mapping between the indices $l$ and $(i,j)$). Compute that for any $\boldsymbol{B}\in\mathbb{R}^{d\times k}$ and $\boldsymbol{V}\in\mathbb{R}^{n\times k}$

$$vec(\partial_{\boldsymbol{V}}\mathcal{F}(\boldsymbol{B}, \boldsymbol{V}))$$

$$= \frac{1}{\bar{N}}\sum_{l=1}^{\bar{N}}(vec(\boldsymbol{A}_l^\top\boldsymbol{B})^\top vec(\boldsymbol{V}) - vec(\boldsymbol{A}_l^\top\boldsymbol{B}^*)^\top vec(\boldsymbol{V}^*))\,vec(\boldsymbol{A}_l^\top\boldsymbol{B})$$

$$= \left(\frac{1}{\bar{N}}\sum_{l=1}^{\bar{N}} vec(\boldsymbol{A}_l^\top\boldsymbol{B})\,vec(\boldsymbol{A}_l^\top\boldsymbol{B})^\top\right)vec(\boldsymbol{V}) - \left(\frac{1}{\bar{N}}\sum_{l=1}^{\bar{N}} vec(\boldsymbol{A}_l^\top\boldsymbol{B})\,vec(\boldsymbol{A}_l^\top\boldsymbol{B}^*)^\top\right)vec(\boldsymbol{V}^*).$$

Since $\boldsymbol{V}^{t+1} = \arg\min_{\boldsymbol{V} \in \mathbb{R}^{n \times k}} \mathcal{F}(\boldsymbol{B}^t, \boldsymbol{V}; \mathcal{A}^{t,1})$, we have $\partial_{\boldsymbol{V}} \mathcal{F}(\boldsymbol{B}^t, \boldsymbol{V}^{t+1}; \mathcal{A}^{t,1}) = 0$ and hence

$$
vec(\boldsymbol{V}^{t+1}) = \left( \frac{1}{\bar{N}} \sum_{l=1}^{\bar{N}} vec(\boldsymbol{A}_l^\top \boldsymbol{B}^t) \, vec(\boldsymbol{A}_l^\top \boldsymbol{B}^t)^\top \right)^{-1} \left( \frac{1}{\bar{N}} \sum_{l=1}^{\bar{N}} vec(\boldsymbol{A}_l^\top \boldsymbol{B}^t) \, vec(\boldsymbol{A}_l^\top \boldsymbol{B}^*)^\top \right) vec(\boldsymbol{V}^*)
$$

$$
= \boldsymbol{H}^{tt-1} \boldsymbol{H}^{t*} vec(\boldsymbol{V}^*),
$$

where we denote

$$
\boldsymbol{H}^{t*} = \frac{1}{\bar{N}} \sum_{l=1}^{\bar{N}} vec(\boldsymbol{A}_l^\top \boldsymbol{B}^t) \, vec(\boldsymbol{A}_l^\top \boldsymbol{B}^*)^\top \in \mathbb{R}^{nk \times nk},
$$

$$
\boldsymbol{H}^{tt} = \frac{1}{\bar{N}} \sum_{l=1}^{\bar{N}} vec(\boldsymbol{A}_l^\top \boldsymbol{B}^t) \, vec(\boldsymbol{A}_l^\top \boldsymbol{B}^t)^\top \in \mathbb{R}^{nk \times nk}.
$$

Use $\otimes$ to denote the Kronecker product of matrices. Recall that

$$
(\boldsymbol{B}^\top \otimes \boldsymbol{A}) \, vec(\boldsymbol{X}) = vec(\boldsymbol{A}\boldsymbol{X}\boldsymbol{B}). \tag{36}
$$

We have

$$
vec\left( (\boldsymbol{B}^* \boldsymbol{V}^{*\top})^\top \boldsymbol{B}^t \right) = vec\left( \boldsymbol{V}^* \boldsymbol{B}^{*\top} \boldsymbol{B}^t \right) = (\boldsymbol{B}^{t\top} \boldsymbol{B}^* \otimes I_d) \, vec(\boldsymbol{V}^*). \tag{37}
$$

$\square$

**Lemma D.2.** *Recall the definition of $\boldsymbol{H}^{tt}$ in equation 35 and that $s_{\min}$ denotes the minimum singular value of a matrix. Suppose that the matrix sensing operator $\mathcal{A}^{t,1}$ satisfies Condition C.2 with constant $\delta^{(1)}$. We can bound*

$$
s_{\min}(\boldsymbol{H}^{tt}) \geq 1 - \delta^{(1)}.
$$

*Proof.* From the definition of $s_{\min}$, we have

$$
s_{\min}(\boldsymbol{H}^{tt}) = \min_{\substack{\boldsymbol{P} \in \mathbb{R}^{n \times k} \\ \|\boldsymbol{P}\|_F = 1}} vec(\boldsymbol{P})^\top \boldsymbol{H}^{tt} vec(\boldsymbol{P}) = \frac{1}{\bar{N}} \langle \mathcal{A}^{t,1}(\boldsymbol{B}^t \boldsymbol{P}^\top), \mathcal{A}^{t,1}(\boldsymbol{B}^t \boldsymbol{P}^\top) \rangle \geq 1 - \delta^{(1)}.
$$

$\square$

**Lemma D.3.** *Recall the definitions of $\boldsymbol{H}^{tt}$ and $\boldsymbol{H}^{t*}$ in equation 35 and equation 34 and recall that $\mathrm{dist}(\boldsymbol{B}^t, \boldsymbol{B}^*)$ is the principal angle distance between the current variable $\boldsymbol{B}^t$ and the ground truth $\boldsymbol{B}^*$. Suppose that the matrix sensing operator $\mathcal{A}^{t,1}$ satisfies Condition C.3 with constant $\delta^{(2)}$. We can bound*

$$
\|\boldsymbol{H}^{tt}(\boldsymbol{B}^{t\top} \boldsymbol{B}^* \otimes I_d) - \boldsymbol{H}^{t*}\|_2 \leq \delta^{(2)} \mathrm{dist}(\boldsymbol{B}^t, \boldsymbol{B}^*). \tag{38}
$$

*Proof.* Recall that $\bar{N} = \bar{m}n$. For the simplicity of notations, we use the collection $\{\boldsymbol{A}_l\}$, $l = 1, \ldots, \bar{N}$ to denote $\{\boldsymbol{A}_{ij}\}$, $i \in [n]$, $j \in \mathcal{S}_i^{t,1}$ (there can be a one-to-one mapping between the indices $l$ and $(i, j)$). For arbitrary $\boldsymbol{W} \in \mathbb{R}^{n \times k}$, $\boldsymbol{P} \in \mathbb{R}^{n \times k}$ with $\|\boldsymbol{W}\|_F = \|\boldsymbol{P}\|_F = 1$, we have

$$
vec(\boldsymbol{W})^\top \boldsymbol{H}^{tt}(\boldsymbol{B}^{t\top} \boldsymbol{B}^* \otimes I_d) \, vec(\boldsymbol{P}) = \frac{1}{\bar{N}} \sum_{i=1}^{\bar{N}} vec(\boldsymbol{W})^\top vec(\boldsymbol{A}_i^\top \boldsymbol{B}^t) \, vec(\boldsymbol{A}_i^\top \boldsymbol{B}^t)^\top vec(\boldsymbol{P} \boldsymbol{B}^{*\top} \boldsymbol{B}^t)
$$

$$
= \frac{1}{\bar{N}} \sum_{i=1}^{\bar{N}} \langle \boldsymbol{A}_i, \boldsymbol{B}^t \boldsymbol{W}^\top \rangle \langle \boldsymbol{A}_i, \boldsymbol{B}^t \boldsymbol{B}^{t\top} \boldsymbol{B}^* \boldsymbol{P}^\top \rangle
$$

$$
= \frac{1}{\bar{N}} \langle \mathcal{A}^{t,1}(\boldsymbol{B}^t \boldsymbol{W}^\top), \mathcal{A}^{t,1}(\boldsymbol{B}^t \boldsymbol{B}^{t\top} \boldsymbol{B}^* \boldsymbol{P}^\top) \rangle,
$$

where we use $(\boldsymbol{B}^\top \otimes \boldsymbol{A}) \, vec(\boldsymbol{X}) = vec(\boldsymbol{A}\boldsymbol{X}\boldsymbol{B})$ in the first equality. Similarly, we can compute that

$$
vec(\boldsymbol{W})^\top \boldsymbol{H}^{t*} vec(\boldsymbol{P}) = \frac{1}{\bar{N}} \langle \mathcal{A}^{t,1}(\boldsymbol{B}^t \boldsymbol{W}^\top), \mathcal{A}^{t,1}(\boldsymbol{B}^* \boldsymbol{P}^\top) \rangle, \tag{39}
$$

and hence

$$vec(\boldsymbol{W})^\top \left( \boldsymbol{H}^{tt}(\boldsymbol{B}^{t\top}\boldsymbol{B}^* \otimes \boldsymbol{I}_d) - \boldsymbol{H}^{t*} \right) vec(\boldsymbol{P}) = \frac{1}{N}\langle \mathcal{A}^{t,1}(\boldsymbol{B}^t\boldsymbol{W}^\top), \mathcal{A}^{t,1}((\boldsymbol{B}^t\boldsymbol{B}^{t\top} - \boldsymbol{I}_k)\boldsymbol{B}^*\boldsymbol{P}^\top)\rangle.$$
(40)

Using Condition C.3 and the definition of $s_{\max}$, we have the result. $\qquad\square$

**Lemma D.4.** *Recall the definitions of $\boldsymbol{F}^t$ and $f^t$ in equation 33 and recall that $\mathrm{dist}(\boldsymbol{B}^t, \boldsymbol{B}^*)$ is the principal angle distance between the current variable $\boldsymbol{B}^t$ and the ground truth $\boldsymbol{B}^*$. Suppose that the matrix sensing operator $\mathcal{A}$ satisfies Conditions C.2 and C.3 with constants $\delta^{(1)}$ and $\delta^{(2)}$ respectively. We can bound*

$$\|\boldsymbol{F}^t\|_F = \|\boldsymbol{f}^t\|_2 \le \frac{\delta^{(2)}s_1\sqrt{k}}{1-\delta^{(1)}}\mathrm{dist}(\boldsymbol{B}^*, \boldsymbol{B}^t),$$
(41)

*and*

$$\|\boldsymbol{V}^{t+1}\|_F \le \sqrt{k}s_1 + \frac{\delta^{(2)}s_1\sqrt{k}}{1-\delta^{(1)}}\mathrm{dist}(\boldsymbol{B}^*, \boldsymbol{B}^t) \le \frac{s_1\sqrt{k}(1-\delta^{(1)}+\delta^{(2)})}{1-\delta^{(1)}}.$$
(42)

*Proof.* The bound on $\|\boldsymbol{F}^t\|_F$ a direct consequence of Lemmas D.1 to D.3 and the fact that the matrix norms are sub-multiplicative. The bound on $\|\boldsymbol{V}^{t+1}\|_F$ is due to the fact that the matrix norms are sub-additive. $\qquad\square$

**Lemma D.5.** *Suppose that the matrix sensing operator $\mathcal{A}^{t,1}$ satisfies Conditions C.2 and C.3 with constants $\delta^{(1)}$ and $\delta^{(2)}$ respectively. Further, suppose that $\max\{\frac{\delta^{(2)}\sqrt{k}}{1-\delta^{(1)}}, \frac{(\delta^{(2)})^2 k}{(1-\delta^{(1)})^2}\} \le \frac{s_k^2 E_0}{36 s_1^2}$. For a sufficiently small step size $\eta_g \le 1/(4s_1^2)$, we have $\boldsymbol{I}_k - \eta_g \boldsymbol{V}^{t+1\top}\boldsymbol{V}^{t+1} \succcurlyeq 0$. Moreover, we can bound*

$$\|\boldsymbol{I}_k - \eta_g \boldsymbol{V}^{t+1\top}\boldsymbol{V}^{t+1}\|_2 \le 1 - \eta_g(E_0 s_k^2 - \frac{2\delta^{(2)}s_1^2\sqrt{k}}{1-\delta^{(1)}}\mathrm{dist}(\boldsymbol{B}^*, \boldsymbol{B}^t)).$$
(43)

*Proof.* We first show that the matrix $\boldsymbol{I}_k - \eta_g \boldsymbol{V}^{t+1\top}\boldsymbol{V}^{t+1}$ is positive semi-definite for a sufficiently small step-size $\eta_g$. Note that following the idea of the seminal work Jain et al. (2013), the update of $\boldsymbol{V}^{t+1} = V(\boldsymbol{B}^t; \mathcal{A}^{t,1})$ can be regarded as a noisy power iteration, as detailed in Lemma D.1. This allows us to compute

$$\|\boldsymbol{V}^{t+1\top}\boldsymbol{V}^{t+1}\|_2 = \|\boldsymbol{V}^*\boldsymbol{B}^{*\top}\boldsymbol{B}^t - \boldsymbol{F}^t\|_2^2 \le 2\|\boldsymbol{V}^*\boldsymbol{B}^{*\top}\boldsymbol{B}^t\|_2^2 + 2\|\boldsymbol{F}^t\|_2^2$$

$$\le 2s_1^2 + 2\left(\frac{\delta^{(2)}s_1\sqrt{k}}{1-\delta^{(1)}}\mathrm{dist}(\boldsymbol{B}^*, \boldsymbol{B}^t)\right)^2 \le 4s_1^2,$$

where we use Lemma D.4 in the first inequality and use $\frac{\delta^{(2)}\sqrt{k}}{1-\delta^{(1)}} \le 1$ in the last. Consequently, for $\eta_g \le \frac{1}{4s_1^2}$, $\boldsymbol{I}_k - \eta_g \boldsymbol{V}^{t+1\top}\boldsymbol{V}^{t+1} \succcurlyeq 0$.

Given the positive semi-definiteness of the matrix $\boldsymbol{I}_k - \eta_g \boldsymbol{V}^{t+1\top}\boldsymbol{V}^{t+1}$, we can bound

$$\|\boldsymbol{I}_k - \eta_g \boldsymbol{V}^{t+1\top}\boldsymbol{V}^{t+1}\|_2 \le 1 - \eta_g s_{\min}(\boldsymbol{V}^{t+1\top}\boldsymbol{V}^{t+1}).$$

By using Lemma D.1 again. We have

$$\boldsymbol{V}^{t+1\top}\boldsymbol{V}^{t+1} = \left(\boldsymbol{V}^*\boldsymbol{B}^{*\top}\boldsymbol{B}^t - \boldsymbol{F}^t\right)^\top \left(\boldsymbol{V}^*\boldsymbol{B}^{*\top}\boldsymbol{B}^t - \boldsymbol{F}^t\right)$$

$$= \boldsymbol{B}^{t\top}\boldsymbol{B}^*\boldsymbol{V}^{*\top}\boldsymbol{V}^*\boldsymbol{B}^{*\top}\boldsymbol{B}^t - \boldsymbol{F}^{t\top}\boldsymbol{V}^*\boldsymbol{B}^{*\top}\boldsymbol{B}^t - \boldsymbol{B}^{t\top}\boldsymbol{B}^*\boldsymbol{V}^{*\top}\boldsymbol{F}^t + \boldsymbol{F}^{t\top}\boldsymbol{F}^t.$$

Note that $\boldsymbol{F}^{t\top}\boldsymbol{F}^t$ is PSD which makes nonnegative contribution to $s_{\min}(\boldsymbol{V}^{t+1\top}\boldsymbol{V}^{t+1})$ and hence

$$s_{\min}(\boldsymbol{V}^{t+1\top}\boldsymbol{V}^{t+1}) \ge s_{\min}(\boldsymbol{B}^{t\top}\boldsymbol{B}^*\boldsymbol{V}^{*\top}\boldsymbol{V}^*\boldsymbol{B}^{*\top}\boldsymbol{B}^t - \boldsymbol{F}^{t\top}\boldsymbol{V}^*\boldsymbol{B}^{*\top}\boldsymbol{B}^t - \boldsymbol{B}^{t\top}\boldsymbol{B}^*\boldsymbol{V}^{*\top}\boldsymbol{F}^t)$$

$$\ge s_{\min}(\boldsymbol{B}^{t\top}\boldsymbol{B}^*\boldsymbol{V}^{*\top}\boldsymbol{V}^*\boldsymbol{B}^{*\top}\boldsymbol{B}^t) - 2s_{\max}(\boldsymbol{F}^{t\top}\boldsymbol{V}^*\boldsymbol{B}^{*\top}\boldsymbol{B}^t)$$

$$\ge s_{\min}^2(\boldsymbol{B}^{t\top}\boldsymbol{B}^*)s_{\min}^2(\boldsymbol{V}^*) - 2\|\boldsymbol{F}^t\|s_{\max}(\boldsymbol{V}^*)$$
(44)

To bound the first term of Eq. equation 44, recall that $s_{\min}^2(\boldsymbol{B^t}^\top \boldsymbol{B^*}) = 1 - (\text{dist}(\boldsymbol{B^*}, \boldsymbol{B^t}))^2$ from Eq. equation 5 and $\text{dist}(\boldsymbol{B^*}, \boldsymbol{B^t}) \leq \text{dist}(\boldsymbol{B^*}, \boldsymbol{B^0})$ from the induction, we have $s_{\min}^2(\boldsymbol{B^t}^\top \boldsymbol{B^*}) \geq E_0$. To bound the last term of Eq. equation 44, we use Lemma D.4 to obtain $\|\boldsymbol{F^t}\|_2 \leq \|\boldsymbol{F^t}\|_F \leq \frac{\delta^{(2)} s_1 \sqrt{k}}{1 - \delta^{(1)}} \text{dist}(\boldsymbol{B^*}, \boldsymbol{B^t})$. Combining the above results, we have

$$\|\boldsymbol{I}_k - \eta_g \boldsymbol{V^{t+1}}^\top \boldsymbol{V^{t+1}}\|_2 \leq 1 - \eta_g (E_0 s_k^2 - \frac{2\delta^{(2)} s_1^2 \sqrt{k}}{1 - \delta^{(1)}} \text{dist}(\boldsymbol{B^*}, \boldsymbol{B^t})). \tag{45}$$

$\square$

**Lemma D.6.** *Recall that $\boldsymbol{V^{t+1}} = V(\boldsymbol{B^t}; \mathcal{A}^{t,1})$ (see the definition of $V(\cdot; \mathcal{A})$ in Eq. equation 15), $\boldsymbol{Q^t} = \boldsymbol{B^t} \boldsymbol{V^{t+1}}^\top - \boldsymbol{B^*} \boldsymbol{V^*}^\top$, $\boldsymbol{G^t} = \frac{1}{n} \sum_{i=1}^n \boldsymbol{G}_i^t$ is the global average of local gradient, and recall that $\text{dist}(\boldsymbol{B^t}, \boldsymbol{B^*})$ is the principal angle distance between the current variable $\boldsymbol{B^t}$ and the ground truth $\boldsymbol{B^*}$. Suppose that the matrix sensing operator $\mathcal{A}^{t,2}$ satisfies Condition C.4 with a constant $\delta^{(3)}$. We have*

$$\|\boldsymbol{G^t} - \boldsymbol{Q^t} \boldsymbol{V^{t+1}}\|_2 \leq \delta^{(3)} s_1^2 k \text{dist}(\boldsymbol{B^*}, \boldsymbol{B^t}). \tag{46}$$

*Proof.* Recall that $\bar{N} = \bar{m}n$. For the simplicity of notations, we use the collection $\{\boldsymbol{A}_l\}$, $l = 1, \ldots, \bar{N}$ to denote $\{\boldsymbol{A}_{ij}\}$, $i \in [n]$, $j \in \mathcal{S}_i^{t,2}$ (there can be a one-to-one mapping between the indices $l$ and $(i, j)$). With this notation, we can compactly write $\boldsymbol{G^t}$ as

$$\boldsymbol{G^t} = \frac{1}{\bar{N}} \sum_{l=1}^{\bar{N}} \langle \boldsymbol{A}_l, \boldsymbol{Q^t} \rangle \boldsymbol{A}_l \boldsymbol{V^{t+1}}. \tag{47}$$

From the definition of $s_{\max}$, for any $\boldsymbol{a} \in \mathbb{R}^d$ with $\|\boldsymbol{a}\|_2 = 1$ and any $\boldsymbol{b} \in \mathbb{R}^k$ with $\|\boldsymbol{b}\|_2 = 1$, we have

$$\|\frac{1}{\bar{N}} \sum_{l=1}^{\bar{N}} \langle \boldsymbol{A}_l, \boldsymbol{Q^t} \rangle \boldsymbol{A}_l \boldsymbol{V^{t+1}} - \boldsymbol{Q^t} \boldsymbol{V^{t+1}}\|_2 = \max_{\|\boldsymbol{a}\|_2 = \|\boldsymbol{b}\|_2 = 1} \boldsymbol{a}^\top \left( \frac{1}{\bar{N}} \sum_{l=1}^{\bar{N}} \langle \boldsymbol{A}_i, \boldsymbol{Q^t} \rangle \boldsymbol{A}_l \boldsymbol{V^{t+1}} - \boldsymbol{Q^t} \boldsymbol{V^{t+1}} \right) \boldsymbol{b}.$$

We obtain the result from Condition C.4

$$|\frac{1}{\bar{N}} \langle \mathcal{A}^{t,2}(\boldsymbol{B^t} \boldsymbol{V^{t+1}}^\top - \boldsymbol{B^*} \boldsymbol{V^*}^\top), \mathcal{A}^{t,2}(\boldsymbol{ab}^\top \boldsymbol{V^{t+1}}^\top) \rangle| \leq \delta^{(3)} s_1^2 k \text{dist}(\boldsymbol{B^t}, \boldsymbol{B^*}). \tag{48}$$

$\square$

**Lemma D.7.** *Recall that $\bar{\boldsymbol{B}}^{t+1} = \boldsymbol{B^{t+1}} \boldsymbol{R^{t+1}}$ is the QR decomposition of $\bar{\boldsymbol{B}}^{t+1}$. Denote $E_0 = 1 - \epsilon_0^2$ and $\sigma = \zeta_g \sigma_g \eta_g / n$. Suppose that Conditions C.2 to C.4 are satisfied with constants $\delta^{(1)}$, $\delta^{(2)}$, and $\delta^{(3)}$. Further, suppose that $\max\{\frac{\delta^{(2)} \sqrt{k}}{1 - \delta^{(1)}}, \frac{(\delta^{(2)})^2 k}{(1 - \delta^{(1)})^2}, \delta^{(3)} k\} \leq \frac{s_k^2 E_0}{36 s_1^2}$ and the level of manually injected noise is sufficiently small $\sigma \leq \frac{E_0}{4 C_\mathcal{N} \kappa^2 \sqrt{k} \log n}$. We have with probability at least $1 - 4 \exp(-k \log n)$*

$$\|(\boldsymbol{R^{t+1}})^{-1}\|_2 \leq \frac{1}{\sqrt{1 - \eta_g s_k^2 E_0 / 2}}. \tag{49}$$

*Proof.* We now focus on bounding $s_{\min}(\boldsymbol{R^{t+1}})$. Recall that $\boldsymbol{G^t} = \partial_{\boldsymbol{B}} \mathcal{F}(\boldsymbol{B^t}, \boldsymbol{V^{t+1}}; \mathcal{A}^{t,2})$ with $\boldsymbol{V^{t+1}} = V(\boldsymbol{B^t}; \mathcal{A}^{t,1})$ (see the definition of $V(\cdot; \mathcal{A})$ in equation 15) and recall that $\bar{\boldsymbol{B}}^{t+1} := \boldsymbol{B^t} - \eta_g \boldsymbol{G^t} + \sigma \boldsymbol{W^t}$. Compute

$$\boldsymbol{R^{t+1}}^\top \boldsymbol{R^{t+1}} = \bar{\boldsymbol{B}}^{t+1\top} \bar{\boldsymbol{B}}^{t+1}$$

$$= \boldsymbol{I}_k + \eta_g^2 \boldsymbol{G^t}^\top \boldsymbol{G^t} + \sigma^2 \boldsymbol{W^t}^\top \boldsymbol{W^t} - \eta_g \boldsymbol{B^t}^\top \boldsymbol{G^t} - \eta_g \boldsymbol{G^t}^\top \boldsymbol{B^t} + \sigma \boldsymbol{B^t}^\top \boldsymbol{W^t} + \sigma \boldsymbol{W^t}^\top \boldsymbol{B^t} - \eta_g \sigma \boldsymbol{G^t}^\top \boldsymbol{W^t} - \eta_g \sigma \boldsymbol{W^t}^\top \boldsymbol{G^t}.$$

Therefore, we have

$$s_{\min}(\boldsymbol{R^{t+1}}^\top \boldsymbol{R^{t+1}}) \geq 1 - 2\eta_g s_{\max}\left( \boldsymbol{B^t}^\top \boldsymbol{G^t} \right) - 2\sigma s_{\max}(\boldsymbol{B^t}^\top \boldsymbol{W^t}) - 2\eta_g \sigma s_{\max}\left( \boldsymbol{G^t}^\top \boldsymbol{W^t} \right). \tag{50}$$

We now bound the last three terms of the R.H.S. of the above inequality.

1. $s_{\max}\left(\boldsymbol{B^t}^\top \boldsymbol{G^t}\right)$: To bound $s_{\max}\left(\boldsymbol{B^t}^\top \boldsymbol{G^t}\right)$, compute that

$$\boldsymbol{B^t}^\top \boldsymbol{G^t} = \boldsymbol{B^t}^\top \boldsymbol{Q^t} \boldsymbol{V}^{t+1} + \boldsymbol{B^t}^\top \left(\boldsymbol{G^t} - \boldsymbol{Q^t}\boldsymbol{V}^{t+1}\right), \tag{51}$$

where we recall the definition of $\boldsymbol{Q^t}$ as $\boldsymbol{Q^t} = \boldsymbol{B^t}\boldsymbol{V}^{t+1}{}^\top - \boldsymbol{B^*}\boldsymbol{V^*}^\top$. Note that the spectral norm of second term in Eq. equation 51 is bounded by Lemma D.6 and we hence focus on the spectral norm of the first term $\boldsymbol{B^t}^\top \boldsymbol{Q^t} \boldsymbol{V}^{t+1}$. Recall the noisy power interpretation of $\boldsymbol{V}^{t+1}$ in Lemma D.1. We can write

$$\boldsymbol{B^t}^\top \boldsymbol{Q^t} \boldsymbol{V}^{t+1} = \boldsymbol{B^t}^\top \left(\boldsymbol{B^t}(\boldsymbol{B^t}^\top \boldsymbol{B^*}\boldsymbol{V^*}^\top - \boldsymbol{F^t}^\top) - \boldsymbol{B^*}\boldsymbol{V^*}^\top\right)\boldsymbol{V}^{t+1} = \boldsymbol{F^t}^\top \boldsymbol{V}^{t+1} = \boldsymbol{F^t}^\top \left(\boldsymbol{V^*}\boldsymbol{B^*}^\top \boldsymbol{B^t} - \boldsymbol{F^t}\right), \tag{52}$$

where we use $\boldsymbol{B^t}^\top \left(\boldsymbol{B^t}\boldsymbol{B^t}^\top - \boldsymbol{I}_d\right)\boldsymbol{B^*}\boldsymbol{V^*}^\top = 0$. Consequently, we can bound $s_{\max}\left(\boldsymbol{B^t}^\top \boldsymbol{G^t}\right)$ as follows

$$s_{\max}\left(\boldsymbol{B^t}^\top \boldsymbol{G^t}\right) \leq s_1 \|\boldsymbol{F^t}\|_2 + \|\boldsymbol{F^t}\|_2^2 + \|\boldsymbol{G^t} - \boldsymbol{Q^t}\boldsymbol{V}^{t+1}\|_2$$

$$\leq \frac{\delta^{(2)} s_1^2 \sqrt{k}}{1 - \delta^{(1)}} + \frac{(\delta^{(2)})^2 s_1^2 k}{(1 - \delta^{(1)})^2} + \delta^{(3)} s_1^2 k \leq \frac{s_k^2 E_0}{12},$$

where we use the assumptions that $\max\{\frac{\delta^{(2)}\sqrt{k}}{1-\delta^{(1)}}, \frac{(\delta^{(2)})^2 k}{(1-\delta^{(1)})^2}, \delta^{(3)} k\} \leq \frac{s_k^2 E_0}{36 s_1^2}$ and $\mathrm{dist}(\boldsymbol{B^*}, \boldsymbol{B^t}) \leq 1$ for the last inequality.

2. $s_{\max}\left(\boldsymbol{B^t}^\top \boldsymbol{W^t}\right)$: Due to the rotational invariance of independent Gaussian random variables, every entry in $\boldsymbol{B^t}^\top \boldsymbol{W^t} \in \mathbb{R}^{k \times k}$ is distributed as $\mathcal{N}(0, 1)$. According to Theorem 4.4.5 in Vershynin (2018), with probability at least $1 - 2\exp(-k \log n)$, we have the bound $\|\boldsymbol{B^t}^\top \boldsymbol{W^t}\| \leq \frac{C_\mathcal{N} \sqrt{k \log n}}{48\sqrt{2}}$ for some universal constant $C_\mathcal{N}$.

3. $s_{\max}\left(\boldsymbol{G^t}^\top \boldsymbol{W^t}\right)$: Let $\boldsymbol{G^t} = U_{\boldsymbol{G}} \boldsymbol{S_G} V_{\boldsymbol{G}}^\top$ be the compact singular value decomposition of $\boldsymbol{G^t}$ such that $U_{\boldsymbol{G}} \in \mathbb{R}^{d \times k}$ and $U_{\boldsymbol{G}}^\top U_{\boldsymbol{G}} = \boldsymbol{I}_k$. We can bound

$$s_{\max}\left(\boldsymbol{G^t}^\top \boldsymbol{W^t}\right) \leq \|\boldsymbol{S_G}\| \|U_{\boldsymbol{G}}^\top \boldsymbol{W^t}\|. \tag{53}$$

Due to the rotational invariance of independent Gaussian random variables, every entry in $\boldsymbol{U_G^t}^\top \boldsymbol{W^t} \in \mathbb{R}^{k \times k}$ is distributed as $\mathcal{N}(0, 1)$ and hence with probability at least $1 - 2\exp(-k \log n)$, we have the bound $\|\boldsymbol{U_G}^\top \boldsymbol{W^t}\| \leq \frac{C_\mathcal{N} \sqrt{k \log n}}{48\sqrt{2}}$ for some universal constant $C_\mathcal{N}$. We now focus on bounding $\|\boldsymbol{S_G}\|_2 = \|\boldsymbol{G^t}\|_2$. Note that

$$\boldsymbol{G^t} = \boldsymbol{Q^t}\boldsymbol{V}^{t+1} + \left(\boldsymbol{G^t} - \boldsymbol{Q^t}\boldsymbol{V}^{t+1}\right), \tag{54}$$

where the spectral norm of second term in Eq. equation 54 is bounded by Lemma D.6. Recall the noisy power interpretation of $\boldsymbol{V}^{t+1}$ in Lemma D.1. We can write

$$\boldsymbol{Q^t}\boldsymbol{V}^{t+1} = \left(\boldsymbol{B^t}(\boldsymbol{B^t}^\top \boldsymbol{B^*}\boldsymbol{V^*}^\top - \boldsymbol{F^t}^\top) - \boldsymbol{B^*}\boldsymbol{V^*}^\top\right)\boldsymbol{V}^{t+1}$$

$$= \left((\boldsymbol{B^t}\boldsymbol{B^t}^\top - \boldsymbol{I}_d)\boldsymbol{B^*}\boldsymbol{V^*}^\top - \boldsymbol{B^t}\boldsymbol{F^t}^\top\right)\left(\boldsymbol{V^*}\boldsymbol{B^*}^\top \boldsymbol{B^t} - \boldsymbol{F^t}\right),$$

and therefore we can bound

$$\|\boldsymbol{Q^t}\boldsymbol{V}^{t+1}\|_2 \leq (s_1 + \|\boldsymbol{F^t}\|_2)^2 \leq \left(s_1 + \frac{\delta^{(2)} s_1 \sqrt{k}}{1 - \delta^{(1)}}\mathrm{dist}(\boldsymbol{B^*}, \boldsymbol{B^t})\right)^2 \leq 4s_1^2, \tag{55}$$

since we assume that $\frac{\delta^{(2)}\sqrt{k}}{1-\delta^{(1)}} \leq 1$.

Combining the above three points, we have with probability at least $1 - 4\exp(-k \log n)$

$$s_{\min}(\boldsymbol{R}^{t+1}{}^\top \boldsymbol{R}^{t+1}) \geq 1 - \eta_g s_k^2 E_0/6 - \sigma \cdot C_\mathcal{N} \sqrt{k \log n}/6 - \eta_g \sigma \cdot C_\mathcal{N} \sqrt{k \log n} \cdot s_1^2/6. \tag{56}$$

Hence, if we choose $\sigma$ such that (recall that $\eta_g \leq 1/4s_1^2$)

$$\sigma \cdot C_\mathcal{N} \sqrt{k \log n} \leq \eta_g s_k^2 E_0 \text{ and } \eta_g \sigma \cdot C_\mathcal{N} \sqrt{k \log n} \cdot s_1^2 \leq \eta_g s_k^2 E_0 \Rightarrow \sigma \leq \frac{E_0}{4C_\mathcal{N} \kappa^2 \sqrt{k \log n}}, \quad (57)$$

we have

$$s_{\min}(\boldsymbol{R}^{t+1^\top} \boldsymbol{R}^{t+1}) \geq 1 - \eta_g s_k^2 E_0/2 \Rightarrow s_{\max}((\boldsymbol{R}^{t+1})^{-1}) \leq \frac{1}{\sqrt{1 - \eta_g s_k^2 E_0/2}}. \quad (58)$$

$\square$

# E    ESTABLISH LEMMAS C.2 TO C.4 FOR THE LRL CASE

## E.1    PROOF OF LEMMA C.2

For the simplicity of the notations, we omit the superscript of $\mathcal{A}_i^{t,1}$ and $\mathcal{A}^{t,1}$. Moreover, recall that $\boldsymbol{v}_i = \boldsymbol{V}_{i,:} \in \mathbb{R}^k$ denotes the $i$th row of the matrix $\boldsymbol{V}$.

While we can directly use an $\epsilon$-net argument to establish the desired property on the set of matrices $\boldsymbol{V} \in \mathbb{R}^{n \times k}, \|\boldsymbol{V}\|_F = 1$, it leads to a suboptimal bound since the size of the $\epsilon$-net is $\mathcal{O}((\frac{2}{\epsilon} + 1)^{nk})$. In the following, we show that by exploiting the special structure of the operator $\mathcal{A}$, i.e. $\boldsymbol{V}$ is row-wise separable in $\mathcal{A}(\boldsymbol{B}^t \boldsymbol{V}^\top)$, we can reduce the size of the $\epsilon$-net to $\mathcal{O}((\frac{2}{\epsilon} + 1)^k)$: Compute that

$$\langle \frac{1}{\sqrt{N}} \mathcal{A}(\boldsymbol{B}^t \boldsymbol{V}^\top), \frac{1}{\sqrt{N}} \mathcal{A}(\boldsymbol{B}^t \boldsymbol{V}^\top) \rangle = \frac{1}{n} \sum_{i=1}^n \frac{1}{\bar{m}} \langle \mathcal{A}_i(\boldsymbol{B}^t \boldsymbol{v}_i \boldsymbol{e}^{(i)\top}), \mathcal{A}_i(\boldsymbol{B}^t \boldsymbol{v}_i \boldsymbol{e}^{(i)\top}) \rangle.$$

If for any $\boldsymbol{v} \in \mathbb{R}^k, \|\boldsymbol{v}\|_2 = 1$, we have $\frac{1}{\bar{m}} \langle \mathcal{A}_i(\boldsymbol{B}^t \boldsymbol{v} \boldsymbol{e}^{(i)\top}), \mathcal{A}_i(\boldsymbol{B}^t \boldsymbol{v} \boldsymbol{e}^{(i)\top}) \rangle = n (1 \pm \mathcal{O}(\delta^{(1)}))$, we can show

$$\langle \frac{1}{\sqrt{N}} \mathcal{A}(\boldsymbol{B}^t \boldsymbol{V}^\top), \frac{1}{\sqrt{N}} \mathcal{A}(\boldsymbol{B}^t \boldsymbol{V}^\top) \rangle = \frac{1}{n} \sum_{i=1}^n \|\boldsymbol{v}_i\|_2^2 \frac{1}{\bar{m}} \langle \mathcal{A}_i(\boldsymbol{B}^t \frac{\boldsymbol{v}_i \boldsymbol{e}^{(i)\top}}{\|\boldsymbol{v}_i\|_2}), \mathcal{A}_i(\boldsymbol{B}^t \frac{\boldsymbol{v}_i \boldsymbol{e}^{(i)\top}}{\|\boldsymbol{v}_i\|_2}) \rangle$$

$$= \sum_{i=1}^n \|\boldsymbol{v}_i\|_2^2 \left(1 \pm \mathcal{O}(\delta^{(1)})\right) = 1 \pm \mathcal{O}(\delta^{(1)}), \forall \boldsymbol{V} \in \mathbb{R}^{n \times k}, \|\boldsymbol{V}\|_F = 1,$$

which is the desired result (note that $\langle \boldsymbol{B}^t \boldsymbol{V}^\top, \boldsymbol{B}^t \boldsymbol{V}^\top \rangle = \|\boldsymbol{B}^t \boldsymbol{V}^\top\|_F = 1$). We now establish

$$\frac{1}{\bar{m}} \langle \mathcal{A}_i(\boldsymbol{B}^t \boldsymbol{v} \boldsymbol{e}^{(i)\top}), \mathcal{A}_i(\boldsymbol{B}^t \boldsymbol{v} \boldsymbol{e}^{(i)\top}) \rangle = \frac{1}{\bar{m}} \sum_{j=1}^{\bar{m}} n((\boldsymbol{x}_{ij}^\top \boldsymbol{B}^t)^\top \boldsymbol{v})^2 = n \left(1 \pm \mathcal{O}(\delta^{(1)})\right)$$

holds for any $\boldsymbol{v} \in \mathbb{R}^k, \|\boldsymbol{v}\|_2 = 1$. Let $\mathcal{S}^{k-1}$ be the sphere in the $k$-dimensional Euclidean space and let $\mathcal{N}_k$ be the $1/4$-net of cardinality $9^k$ (see Corollary 4.2.13 in Vershynin (2018)). Note that $\frac{1}{\bar{m}} \sum_{j=1}^{\bar{m}} n((\boldsymbol{x}_{ij}^\top \boldsymbol{B}^t)^\top \boldsymbol{v})^2 - n = n \boldsymbol{v}^\top \left(\boldsymbol{B}^{t\top}(\frac{1}{\bar{m}} \sum_{j=1}^{\bar{m}} \boldsymbol{x}_{ij} \boldsymbol{x}_{ij}^\top) \boldsymbol{B}^t - \boldsymbol{I}_k\right) \boldsymbol{v}$ and we have

$$\sup_{\boldsymbol{v} \in \mathcal{S}^{k-1}} \boldsymbol{v}^\top \left(\boldsymbol{B}^{t\top}(\frac{1}{\bar{m}} \sum_{j=1}^{\bar{m}} \boldsymbol{x}_{ij} \boldsymbol{x}_{ij}^\top) \boldsymbol{B}^t - \boldsymbol{I}_k\right) \boldsymbol{v} \leq 2 \sup_{\boldsymbol{v} \in \mathcal{N}_k} \boldsymbol{v}^\top \left(\boldsymbol{B}^{t\top}(\frac{1}{\bar{m}} \sum_{j=1}^{\bar{m}} \boldsymbol{x}_{ij} \boldsymbol{x}_{ij}^\top) \boldsymbol{B}^t - \boldsymbol{I}_k\right) \boldsymbol{v},$$

$$(59)$$

where we use Lemma 4.4.1 in Vershynin (2018). In the following, we prove

$$\sup_{\boldsymbol{v} \in \mathcal{N}_k} \boldsymbol{v}^\top \left(\boldsymbol{B}^{t\top}(\frac{1}{\bar{m}} \sum_{j=1}^{\bar{m}} \boldsymbol{x}_{ij} \boldsymbol{x}_{ij}^\top) \boldsymbol{B}^t - \boldsymbol{I}_k\right) \boldsymbol{v} \leq \delta^{(1)}/2$$

so that we have $\frac{1}{\bar{m}} \sum_{j=1}^{\bar{m}} n((\boldsymbol{x}_{ij}^\top \boldsymbol{B}^t)^\top \boldsymbol{v})^2 = n \left(1 + \mathcal{O}(\delta^{(1)})\right)$.

For any fixed index $i$, denote $Z_{ij} = (\boldsymbol{x}_{ij}^\top \boldsymbol{B}^t \boldsymbol{v})^2 = n(\boldsymbol{x}_{ij}^\top \boldsymbol{B}^t \boldsymbol{v})^2$ and note that $\mathbb{E}_{\boldsymbol{x}_{ij}}[Z_{ij}] = n$. Since $\boldsymbol{x}_{ij}$'s are independent subgaussian variables, $Z_{ij}$'s are independent subexponential variables. Recall that $\|\cdot\|_{\psi_2}$ and $\|\cdot\|_{\psi_1}$ denote the subgaussian norm and subexponential norm respectively and

we have $\|XY\|_{\psi_1} \leq \|X\|_{\psi_2}\|Y\|_{\psi_2}$ for subgaussian random variables $X$ and $Y$ (see Lemma 2.2.7 in Vershynin (2018)). Therefore, we can bound $\|Z_{ij}\|_{\psi_1} \leq n\|\boldsymbol{B}^t\boldsymbol{v}\|_2^2 = n$. Using the centering property (see Exercise 2.7.10 of Vershynin (2018)) and Bernstein's inequality (see Theorem 2.8.1 of Vershynin (2018)) of the zero mean subexponential variables, we can bound, for every $\tau \geq 0$

$$\Pr\{|\frac{1}{\bar{m}}\sum_{j=1}^{\bar{m}} Z_{ij} - n| \geq n\tau\} \leq 2\exp\left(-c\min\{\frac{\bar{m}^2 n^2 \tau^2}{\sum_{j=1}^{\bar{m}} \|Z_{ij}\|_{\psi_1}^2}, \frac{\bar{m}n\tau}{\max_{j\in[\bar{m}]} \|Z_{ij}\|_{\psi_1}}\}\right)$$

$$= 2\exp\left(-c\min\{\bar{m}\tau^2, \bar{m}\tau\}\right) \overset{\tau\leq 1}{=} 2\exp\left(-c\bar{m}\tau^2\right)$$

where $c > 0$ is an absolute constant. Using the union bound over $\mathcal{N}_k$ and set $\tau = \sqrt{k\log n}/\sqrt{\bar{m}}$. We obtain $\delta^{(1)} \leq \sqrt{k\log n}/\sqrt{\bar{m}}$ with probability at least $1 - \exp(-c'k\log n)$ for some constant $c'$. Similarly, we can show that $\frac{1}{\bar{m}}\sum_{j=1}^{\bar{m}} n((\boldsymbol{x}_{ij}^\top\boldsymbol{B}^t)^\top\boldsymbol{v})^2 = n\left(1 - \mathcal{O}(\delta^{(1)})\right)$. We therefore have the result.

### E.2 Proof of Lemma C.3

Recall that $\boldsymbol{v}_i \in \mathbb{R}^k$ denotes the $i$th row of the matrix $\boldsymbol{V}$ and denote $\boldsymbol{W} = (\boldsymbol{B}^t\boldsymbol{B}^{t\top} - \boldsymbol{I}_d)\boldsymbol{B}^*$.

Following a similar argument as Lemma C.2, we simply need to focus on showing for any $\boldsymbol{v}_1, \boldsymbol{v}_2 \in \mathbb{R}^k, \|\boldsymbol{v}_1\| = \|\boldsymbol{v}_2\| = 1$, we have $\frac{1}{\bar{m}}\langle\mathcal{A}_i(\boldsymbol{W}\boldsymbol{v}_1\boldsymbol{e}^{(i)\top}), \mathcal{A}_i(\boldsymbol{B}^t\boldsymbol{v}_2\boldsymbol{e}^{(i)\top})\rangle = \pm\mathcal{O}(n\delta^{(2)})$ (Note that $\langle\boldsymbol{W}\boldsymbol{v}_1\boldsymbol{e}^{(i)\top}, \boldsymbol{B}^t\boldsymbol{v}_2\boldsymbol{e}^{(i)\top}\rangle = 0$). Let $\mathcal{S}^{k-1}$ be the sphere in the $k$-dimensional Euclidean space and let $\mathcal{N}_k$ be the $1/4$-net of cardinality $9^k$ (see Corollary 4.2.13 in Vershynin (2018)). Note that

$$\frac{1}{\bar{m}}\langle\mathcal{A}_i(\boldsymbol{W}\boldsymbol{B}^*\boldsymbol{v}_1\boldsymbol{e}^{(i)\top}), \mathcal{A}_i(\boldsymbol{B}^t\boldsymbol{v}_2\boldsymbol{e}^{(i)\top})\rangle = n\boldsymbol{v}_1^\top\left(\boldsymbol{W}^\top(\frac{1}{\bar{m}}\sum_{j=1}^{\bar{m}}\boldsymbol{x}_{ij}\boldsymbol{x}_{ij}^\top)\boldsymbol{B}^t\right)\boldsymbol{v}_2. \quad (60)$$

Moreover, according to Exercise 4.4.3 in Vershynin (2018), we have

$$\sup_{v_1,v_2\in\mathcal{S}^{k-1}} \boldsymbol{v}_1^\top\left(\boldsymbol{W}^\top(\frac{1}{\bar{m}}\sum_{j=1}^{\bar{m}}\boldsymbol{x}_{ij}\boldsymbol{x}_{ij}^\top)\boldsymbol{B}^t\right)\boldsymbol{v}_2 \leq 2\sup_{v_1,v_2\in\mathcal{N}_k} \boldsymbol{v}_1^\top\left(\boldsymbol{W}^\top(\frac{1}{\bar{m}}\sum_{j=1}^{\bar{m}}\boldsymbol{x}_{ij}\boldsymbol{x}_{ij}^\top)\boldsymbol{B}^t\right)\boldsymbol{v}_2.$$

In the following, we prove that the quantity on the RHS is bounded by $\delta^{(2)}/2$ so that we have the desired result.

For any fixed index $i$, denote $Z_{ij} = n(\boldsymbol{x}_{ij}^\top\boldsymbol{W}\boldsymbol{v}_1)(\boldsymbol{x}_{ij}^\top\boldsymbol{B}^t\boldsymbol{v}_2)$ and note that $\mathbb{E}_{\boldsymbol{x}_{ij}}[Z_{ij}] = 0$. Since $\boldsymbol{x}_{ij}$'s are independent subgaussian variables, $Z_{ij}$'s are independent subexponential variables. We can bound $\|Z_{ij}\|_{\psi_1} \leq n\|\boldsymbol{W}\boldsymbol{v}_1\|_2\|\boldsymbol{B}^t\boldsymbol{v}_2\|_2 = n\mathrm{dist}(\boldsymbol{B}^t, \boldsymbol{B}^*)$. Using the centering property (see Exercise 2.7.10 of Vershynin (2018)) and Bernstein's inequality (see Theorem 2.8.1 of Vershynin (2018)) of the zero mean subexponential variables, we can bound, for every $t \geq 0$

$$\Pr\{|\frac{1}{\bar{m}}\sum_{j=1}^{\bar{m}} Z_{ij}| \geq n\tau\} \leq 2\exp\left(-c\min\{\frac{\bar{m}^2 n^2 \tau^2}{\sum_{j=1}^{\bar{m}} \|Z_{ij}\|_{\psi_1}^2}, \frac{\bar{m}n\tau}{\max_{j\in[\bar{m}]} \|Z_{ij}\|_{\psi_1}}\}\right)$$

$$= 2\exp\left(-c\min\{\frac{\bar{m}\tau^2}{(\mathrm{dist}(\boldsymbol{B}^t, \boldsymbol{B}^*))^2}, \frac{\bar{m}\tau}{\mathrm{dist}(\boldsymbol{B}^t, \boldsymbol{B}^*)}\}\right)$$

where $c > 0$ is an absolute constant. Use the union bound over $\mathcal{N}_k^2$, set $\delta^{(2)} \leq \sqrt{k\log n}/\sqrt{\bar{m}}$ and set $\tau = \mathrm{dist}(\boldsymbol{B}^t, \boldsymbol{B}^*)\delta^{(2)}$. We show that Condition C.3 is satisfied with parameter $\delta^{(2)}$ with probability at least $1 - \exp(-c'k\log n)$ for some constant $c'$.

### E.3 Proof of Lemma C.4

Denote $\boldsymbol{Q}^t = \boldsymbol{B}^t\boldsymbol{V}^{t+1\top} - \boldsymbol{B}^*\boldsymbol{V}^{*\top}$ and recall that $\boldsymbol{V}_{i,:} \in \mathbb{R}^k$ denotes the $i$th row of the matrix $\boldsymbol{V}$. For simplicity, denote $\boldsymbol{W} = \frac{1}{N}\sum_{i=1,j=1}^{n,\bar{m}}\langle\boldsymbol{A}_{ij}, \boldsymbol{Q}^t\rangle\boldsymbol{A}_{ij}\boldsymbol{V}^{t+1}$ and note that $\mathbb{E}_{\boldsymbol{x}_{ij}}[\boldsymbol{W}] = \boldsymbol{Q}^t\boldsymbol{V}^{t+1}$.

We use an $\epsilon$-net argument to establish the desired property on the set of vectors $\boldsymbol{a} \in \mathbb{R}^d, \boldsymbol{b} \in \mathbb{R}^k, \|\boldsymbol{a}\| = \|\boldsymbol{b}\| = 1$. Let $\mathcal{S}^{k-1}$ and $\mathcal{S}^{d-1}$ be the spheres in the $k$-dimensional and $d$-dimensional

Euclidean spaces and let $\mathcal{N}_k$ and $\mathcal{N}_d$ be the $1/4$-net of cardinality $9^k$ and $9^d$ respectively (see Corollary 4.2.13 in Vershynin (2018)). Note that

$$\langle \mathcal{A}(\boldsymbol{Q}^t), \mathcal{A}(\boldsymbol{a}\boldsymbol{b}^\top \boldsymbol{V}^{t+1^\top}) \rangle = \boldsymbol{a}^\top \boldsymbol{W} \boldsymbol{b}. \tag{61}$$

Moreover, according to Exercise 4.4.3 in Vershynin (2018), we have

$$\sup_{\boldsymbol{a} \in \mathcal{S}^{d-1}, \boldsymbol{b} \in \mathcal{S}^{k-1}} \boldsymbol{a}^\top (\boldsymbol{W} - \boldsymbol{Q}^t \boldsymbol{V}^{t+1}) \boldsymbol{b} \leq 2 \sup_{\boldsymbol{a} \in \mathcal{N}_d, \boldsymbol{b} \in \mathcal{N}_k} \boldsymbol{a}^\top (\boldsymbol{W} - \boldsymbol{Q}^t \boldsymbol{V}^{t+1}) \boldsymbol{b}.$$

In the following, we prove that the quantity on the RHS is bounded by $\delta^{(3)} s_1^2 k \mathrm{dist}(\boldsymbol{B}^t, \boldsymbol{B}^*)/2$ so that we have the desired result. We first bound $\|\boldsymbol{Q}^t \boldsymbol{e}^{(i)}\|_2$ and $\|\boldsymbol{V}_{i,:}^{t+1}\|_2$ as they will be used in our concentration argument.

**Bound $\|\boldsymbol{V}_{i,:}^{t+1}\|_2$.** Using Lemma D.1, we can write $\boldsymbol{V}^{t+1} = \boldsymbol{V}^* \boldsymbol{B}^{*\top} \boldsymbol{B}^t - \boldsymbol{F}^t$ and therefore $\boldsymbol{V}_{i,:}^{t+1} = \boldsymbol{B}^{t^\top} \boldsymbol{B}^* \boldsymbol{V}_{i,:}^* - \boldsymbol{F}_{i,:}^t$. Using Assumption 5.2, we have $\|\boldsymbol{V}_{i,:}^*\|_2 \leq \mu\sqrt{k} s_k/\sqrt{n}$. Further, we can compute that

$$\boldsymbol{F}_{i,:}^t = \left( \frac{1}{\bar{m}} \boldsymbol{B}^{t^\top} \boldsymbol{x}_{ij} \boldsymbol{x}_{ij}^\top \boldsymbol{B}^t \right)^{-1} \left( \frac{1}{\bar{m}} \boldsymbol{B}^{t^\top} \boldsymbol{x}_{ij} \boldsymbol{x}_{ij}^\top (\boldsymbol{B}^t \boldsymbol{B}^{t^\top} - \boldsymbol{I}_d) \boldsymbol{B}^* \right) \boldsymbol{V}_{i,:}^*. \tag{62}$$

Using the variational formulation of the spectral norm and Conditions C.2 and C.3 (note that $\delta^{(1)} = \delta^{(2)} = \sqrt{k \log n}/\sqrt{\bar{m}}$), we have $\|\boldsymbol{F}_{i,:}^t\|_2 \leq \frac{\delta^{(2)} \mathrm{dist}(\boldsymbol{B}^t, \boldsymbol{B}^*)}{1 - \delta^{(1)}} \cdot \frac{\mu\sqrt{k} s_k}{\sqrt{n}}$. Therefore we obtain

$$\|\boldsymbol{V}_{i,:}^{t+1}\|_2 \leq 2\mu\sqrt{k} s_k/\sqrt{n}. \tag{63}$$

The high probability upper bound of $\boldsymbol{w}_i^t$ in Lemma C.1 can be derived from the above inequality by noting that $\boldsymbol{w}_i^t = \sqrt{n} \boldsymbol{V}_{i,:}^t$.

**Bound $\|\boldsymbol{Q}^t \boldsymbol{e}^{(i)}\|_2$.** Recall the definition of $\boldsymbol{Q}^t = \boldsymbol{B}^t \boldsymbol{V}^{t+1^\top} - \boldsymbol{B}^* \boldsymbol{V}^{*\top}$. Using Lemma D.1, we obtain

$$\boldsymbol{Q}^t = \boldsymbol{B}^t (\boldsymbol{V}^* \boldsymbol{B}^{*\top} \boldsymbol{B}^t - \boldsymbol{F}^t)^\top - \boldsymbol{B}^* \boldsymbol{V}^{*\top} = (\boldsymbol{B}^t \boldsymbol{B}^{t^\top} - \boldsymbol{I}_d) \boldsymbol{B}^* \boldsymbol{V}^* - \boldsymbol{B}^t \boldsymbol{F}^{t^\top}. \tag{64}$$

We can bound $\|\boldsymbol{Q}^t \boldsymbol{e}^{(i)}\|_2 \leq \|(\boldsymbol{B}^t \boldsymbol{B}^{t^\top} - \boldsymbol{I}_d) \boldsymbol{B}^*\|_2 \|\boldsymbol{v}_i^*\|_2 + \|\boldsymbol{F}_{i,:}^t\|_2 \leq 2\mathrm{dist}(\boldsymbol{B}^t, \boldsymbol{B}^*) \mu\sqrt{k} s_k/\sqrt{n}$.

Denote $Z_{ij} = n(\boldsymbol{x}_{ij}^\top \boldsymbol{Q}^t \boldsymbol{e}^{(i)})(\boldsymbol{x}_{ij}^\top \boldsymbol{a}\boldsymbol{b}^\top \boldsymbol{V}_{i,:}^{t+1})$ and note that $\frac{1}{n} \sum_{i=1}^n \mathbb{E}_{\boldsymbol{x}_{ij}}[Z_{ij}] = n\boldsymbol{b}^\top \boldsymbol{V}^{t+1^\top} \boldsymbol{Q}^t \boldsymbol{a}$. Since $\boldsymbol{x}_{ij}$'s are independent subgaussian variables, $Z_{ij}$'s are independent subexponential variables. We can bound $\|Z_{ij}\|_{\psi_1} \leq n\|\boldsymbol{Q}^t \boldsymbol{e}^{(i)}\|_2 \|\boldsymbol{a}\boldsymbol{b}^\top \boldsymbol{V}_{i,:}^{t+1}\|_2 \leq 4k\mu^2 s_k^2 \mathrm{dist}(\boldsymbol{B}^t, \boldsymbol{B}^*)$. Using the centering property (see Exercise 2.7.10 of Vershynin (2018)) and Bernstein's inequality (see Theorem 2.8.1 of Vershynin (2018)) of the zero mean subexponential variables, we can bound, for every $t \geq 0$

$$\Pr\{ |\frac{1}{n\bar{m}} \sum_{i=1,j=1}^{n,\bar{m}} Z_{ij} - \boldsymbol{a}^\top (\boldsymbol{Q}^t \boldsymbol{V}^{t+1}) \boldsymbol{b}| \geq \tau \}$$

$$\leq 2\exp\left( -c \min\{ \frac{\bar{m}^2 n^2 \tau^2}{\sum_{i=1,j=1}^{n,\bar{m}} \|Z_{ij}\|_{\psi_1}^2}, \frac{\bar{m} n \tau}{\max_{i \in [n], j \in [\bar{m}]} \|Z_{ij}\|_{\psi_1}} \} \right)$$

$$= 2\exp\left( -c \min\{ \frac{\bar{m} n \tau^2}{(4k\mu^2 s_k^2 \mathrm{dist}(\boldsymbol{B}^t, \boldsymbol{B}^*))^2}, \frac{\bar{m} n \tau}{4k\mu^2 s_k^2 \mathrm{dist}(\boldsymbol{B}^t, \boldsymbol{B}^*)} \} \right)$$

Set $\tau = \delta^{(3)} s_1^2 k \mathrm{dist}(\boldsymbol{B}^t, \boldsymbol{B}^*)$. We have that when $\delta^{(3)} = \frac{4(\sqrt{d} + \sqrt{k \log n})}{\sqrt{\bar{m}} n \kappa^2}$,

$$\Pr\{ |\frac{1}{n\bar{m}} \sum_{i=1,j=1}^{n,\bar{m}} Z_{ij} - \boldsymbol{a}^\top (\boldsymbol{Q}^t \boldsymbol{V}^{t+1}) \boldsymbol{b}| \geq \tau \} \leq \exp\left( -c(d + k \log n) \right). \tag{65}$$

Use the union bound over $\mathcal{N}_k \times \mathcal{N}_d$, we have that with probability at least $1 - \exp(-c'' k \log n)$ Condition C.4 holds with $\delta^{(3)} = \frac{4(\sqrt{d} + \sqrt{k \log n})}{\sqrt{\bar{m}} n \kappa^2}$.

## F   ANALYSIS OF THE INITIALIZATION PROCEDURE

In this section, we present the privacy and utility guarantees of the INITIALIZATION procedure (Algorithm 3), when the local data points $x_{ij}$ are Gaussian variables. Recall that to establish the utility guarantee for CENTAUR in the LRL setting, a column orthonormal initialization $B^0 \in \mathbb{R}^{d \times k}$ such that the initial error $\text{dist}(B^0, B^*) \le \epsilon_0 = 0.2$ is required. While such a requirement can be ensured by the private power method (PPM, presented in Algorithm 5), the utility guarantee only holds *with a constant probability*, e.g. 0.99.

A key contribution of our work is to propose a *cross-validation* type scheme to boost the success probability of PPM to $1 - O(n^{-k})$ with a small extra cost of $O(k \log n)$. Note that boosting the success probability has a small cost of utility and hence we need a higher target accuracy $\epsilon_i = 0.01$[5] for PPM compared to $\epsilon_0 = 0.2$. The most important novelty of our selection scheme is that it only takes as input the results of $O(k \log n)$ independent PPM runs, which hence can be treated as *post-processing* and is free of privacy leakage.

### F.1   UTILITY AND PRIVACY GUARANTEES OF THE PPM PROCEDURE

In this section, we present the guarantees for the PPM procedure, under the additional assumption that $x_{ij}$ are Gaussian variables. One can prove that the choice of $\zeta_0$ described in Lemma F.1 is a high probability upper bound of $\|Y_i^l\|_F$ (see Lemma G.1 in the appendix). Therefore, with the same high probability, the clipping operation in the Gaussian mechanism will not take effect.

Conditioned on the above event (clipping takes no effect), to establish the utility guarantee of Algorithm 5, we view PPM Hardt & Price (2014) as a specific instance of the perturbed power method presented in Algorithm 6. Suppose that the level of perturbation is sufficiently small, we can exploit the analysis from (Hardt & Price, 2014) to prove the following lemma.

**Lemma F.1** (Utility Guarantee of a Single PPM Trial). *Consider the LRL setting. Suppose that Assumptions 5.2 and 5.1 hold and $x_{ij}$ are Gaussian variables. Let $\epsilon_i = 0.01$ be the target accuracy. For PPM presented in Algorithm 5, set $\zeta_0 = c_0' \mu^2 k^{1.5} s_k^2 d^{0.5} \log n \cdot (\sqrt{\log n} + \sqrt{k})$, $L = c_L'(s_k^2 + \sum_{j=1}^k s_j^2)/s_k^2 \cdot \log(kd/\epsilon_i)$. Suppose that $n$ is sufficiently large so that there exists $\bar{m}_0$ such that*

$$\frac{\bar{m}n}{\log^6 \bar{m}n} \ge c_1' \frac{d \cdot \log^6 d \cdot k^3 \cdot \mu^2 \cdot \sum_{i=1}^k s_i^2}{s_k^2}. \tag{66}$$

*Choose a noise multiplier $\sigma_0 = ns_k^2/c_2'\zeta_0 k\sqrt{d \log L}$. We have with probability at least 0.99, we have $\text{dist}(B^0, B^*) \le \epsilon_i$.*

**Remark F.1.** *If we focus on the dependence on the problem dimension $d$ and the number clients $n$, treat other quantities, e.g. the rank $k$, as constant, and ignore the logarithmic terms, we have $\bar{m} \ge \Theta(d/n)$, which is the same as the requirement on $\bar{m}$ in Lemma C.5.*

The following RDP guarantee of PPM can be established using the Gaussian mechanism of RDP and the RDP composition lemma.

**Lemma F.2** (Privacy Guarantee). *Consider PPM presented in Algorithm 5. Set inputs $L$, the number of communication rounds, $\sigma_0$, the noise multiplier according to Lemma F.1. We have that PPM is an $(\alpha, \epsilon_{init}')$-RDP mechanism with*

$$\epsilon_{init}' = \frac{\alpha L \cdot (c_2'\zeta_0 k\sqrt{d \log L})^2}{n^2 s_k^4} = \tilde{O}\left(\frac{\alpha \kappa^2 k^7 \mu^4 d^2}{n^2}\right), \tag{67}$$

*where $\tilde{O}$ hides the constants and the log terms and we treat $(s_k^2 + \sum_{j=1}^k s_j^2)/s_k^2 = O(k)$.*

### F.2   BOOST THE SUCCESS PROBABILITY WITH CROSS-VALIDATION

The following lemma shows that if the output of PPM has utility at least $\epsilon_i$, e.g. $\epsilon_i = 0.01$, with probability $p$, e.g. $p = 0.99$, then any candidate that passes the test (68) has utility no less than $\epsilon_0$, e.g. $\epsilon_0 = 0.2$, with a high probability $(1 - \delta)$. The proof is provided in Appendix H.

---

[5]The choice of $\epsilon_i$ should satisfy equation 69. $\epsilon_i = 0.01$ is one valid example.

---

**Algorithm 5** PPM: Private Power Method (Adapted from Hardt & Price (2014))

---

1: **procedure** PPM($L$, $\sigma_0$, $\zeta_0$, $\bar{m}_0$)
2:     Choose $\boldsymbol{X}^0 \in \mathbb{R}^{d \times k}$ to be a random column orthonormal matrix;
3:     **for** $l = 1$ to $L$ **do**
4:         **for** $i = 1$ to $n$ **do**
5:             Sample without replacement a subset $\mathbb{S}_i^{l,0}$ from $\mathbb{S}_i$ with cardinality $\bar{m}_0$.
6:             Denote $\boldsymbol{M}_i^l = \frac{1}{\bar{m}} \sum_{j \in \mathbb{S}_i^{l,0}} y_{ij}^2 \boldsymbol{x}_{ij} \boldsymbol{x}_{ij}^\top$.
7:             Compute $\boldsymbol{Y}_i^l := \boldsymbol{M}_i^l \boldsymbol{X}^{l-1}$.
8:         Let $\boldsymbol{X}^l = \mathcal{QR}(\boldsymbol{Y}^l)$, where $\boldsymbol{Y}^l = \mathrm{GM}_{\zeta_0, \sigma_0}(\{\boldsymbol{Y}_i^l\}_{i=1}^n)$.
9:         // $\mathcal{QR}(\cdot)$ denotes the QR decomposition and only returns the $Q$ matrix.

---

**Algorithm 6** NPM: Noisy Power Method (Adapted from Hardt & Price (2014))

---

1: **procedure** NPM($\boldsymbol{A}$, $L$)
2:     // $\boldsymbol{A}$ is the target matrix
3:     Choose $\boldsymbol{X}^0 \in \mathbb{R}^{d \times k}$ to be a random column orthonormal matrix;
4:     **for** $l = 1$ to $L$ **do**
5:         Let $\boldsymbol{X}^l = \mathcal{QR}(\boldsymbol{Y}^l)$, where $\boldsymbol{Y}^l = \boldsymbol{A}\boldsymbol{X}^{l-1} + \boldsymbol{G}^l$.
6:         // $\boldsymbol{G}^l$ is some perturbation matrix.
7:         // $\mathcal{QR}(\cdot)$ denotes the QR decomposition and only returns the $Q$ matrix.

---

**Lemma F.3.** *Use $\epsilon_0$ to denote the accuracy required by* CENTAUR *in the LRL setting and use $\epsilon_i$ to denote the accuracy of a single* PPM *trial. Recall that $p = 0.99$ is the probability of success of* PPM. *Use $\boldsymbol{B}_{0,c}$ to denote the output of* PPM *in the $c^{th}$ trial and set $T_0 = 8p \log 1/\delta$ in the procedure* INITIALIZATION *(Algorithm 3). We have with probability at least $1 - \delta$ that there exists one element $\hat{c}$ in $\{1, \ldots, T_0\}$ such that for at least half of $c \in \{1, \ldots, T_0\}$,*

$$s_i(\boldsymbol{B}_{0,c}^\top \boldsymbol{B}_{0,\hat{c}}) \geq 1 - 2\epsilon_i^2, \forall i \in [k]. \tag{68}$$

*Moreover, $\boldsymbol{B}_{0,\hat{c}}$ must satisfy $\mathrm{dist}(\boldsymbol{B}_{0,\hat{c}}, \boldsymbol{B}^*) \leq \epsilon_0$ with a sufficiently small $\epsilon_i \in [0,1]$ such that*

$$\sqrt{1 - \epsilon_0^2} + 1 - \sqrt{1 - \epsilon_i^2} + \epsilon_i < 1 - 2\epsilon_i^2. \tag{69}$$

*One valid example is $\epsilon_i = 0.01$ and $\epsilon_0 = 0.2$, which is the one chosen in our previous discussions.*

**Corollary F.1.** *Consider the* INITIALIZATION *procedure presented in Algorithm 3. By setting $T_0 = c_T' k \log n$, $\epsilon_i = 0.01$, and setting $L_0$, $\sigma_0$ and $\zeta_0$ according to Lemma F.1, we have with probability at least $1 - n^{-k}$, the output $\boldsymbol{B}_{\hat{c}}$ satisfies $\mathrm{dist}(\boldsymbol{B}_{\hat{c}}, \boldsymbol{B}^*) \leq \epsilon_0 = 0.2$. Moreover, the* INITIALIZATION *procedure is an $(\alpha, \epsilon_{init})$-RDP mechanism with*

$$\epsilon_{init} = \epsilon_{init}' \cdot T_0 = \tilde{O}\left(\frac{\alpha \kappa^2 k^8 \mu^4 d^2}{n^2}\right). \tag{70}$$

The idea of boosting the success probability is inspired by Algorithm 5 of (Liang et al., 2014). The major improvement of our approach is that given the outputs of $O(k \log n)$ PPM trials, it no long requires access to the dataset and hence can be treated as postprocessing, while (Liang et al., 2014) requires an extra data-dependent SVD operation which violates the purpose of DP protection in the first place.

## G    GUARANTEES FOR THE PPM PROCEDURE

In this section, we present the analysis to establish the guarantees for the PPM procedure. We first show that the choice of $\zeta_0$ described in Lemma F.1 is a high probability upper bound of $\|\boldsymbol{Y}_i^l\|_F$. Therefore, with the same high probability, the clipping operation in the Gaussian mechanism will not take effect.

**Lemma G.1.** *Consider the LRL setting. Under the assumptions 5.2 and 5.1, we have with a probability at least $1 - 3\bar{m}n^{-100}$,*

$$\|\boldsymbol{Y}_i^l\|_F \leq \zeta_0 \dot{=} c_0 \mu^2 k^{1.5} s_k^2 (\log n)(\sqrt{\log n} + \sqrt{d})(\sqrt{\log n} + \sqrt{k})$$

where $\boldsymbol{Y}_i^l$ is computed in line 7 of Algorithm 5 and $c_0$ is some universal constant.

*Proof.* The detailed expression of $\boldsymbol{Y}_i^l$ in line 7 of Algorithm 5 can be calculated as follows:

$$\boldsymbol{Y}_i^l = \boldsymbol{M}_i^l \boldsymbol{X}^{l-1} = \frac{1}{\bar{m}} \sum_{j \in \mathbb{S}_i^{l,0}} y_{ij}^2 \boldsymbol{x}_{ij} \boldsymbol{x}_{ij}^\top \boldsymbol{X}^{l-1}. \tag{71}$$

Using the triangle inequality of the matrix norm, $\zeta$ is a high probability upper bound of $\|\boldsymbol{Y}_i^l\|_F$ if the following inequality holds with a high probability,

$$\|y_{ij}^2 \boldsymbol{x}_{ij} \boldsymbol{x}_{ij}^\top \boldsymbol{X}^{l-1}\|_F = |y_{ij}|^2 \|\boldsymbol{x}_{ij}\| \|\boldsymbol{x}_{ij}^\top \boldsymbol{X}^{l-1}\| \le \zeta_0. \tag{72}$$

To bound $|y_{ij}|^2$, recall that in Assumption 5.1, we assume that $\boldsymbol{x}_{ij}$ is a sub-Gaussian random vector with $\|\boldsymbol{x}_{ij}\|_{\psi_2} = 1$. Using the definition of a sub-Gaussian random vector, we have

$$\mathbb{P}\{|y_{ij}| \ge \tau\} \le 2 \exp(-c_s \tau^2 / \|\boldsymbol{w}_i^*\|^2) \le \exp(-100 \log n), \tag{73}$$

with the choice $\tau = \mu \sqrt{k} s_k \cdot \sqrt{(100 \log n + \log 2)/c_s}$ since $\|\boldsymbol{w}_i^*\|_2 \le \mu \sqrt{k} s_k$. Here $c_s$ is some constant and we recall that $s_k$ is a shorthand for $s_k(\boldsymbol{W}^*/\sqrt{n})$.

To bound $\|\boldsymbol{x}_{ij}\|$, recall that $\boldsymbol{x}_{ij}$ is a sub-Gaussian random vector with $\|\boldsymbol{x}_{ij}\|_{\psi_2} = 1$ and therefore with probability at least $1 - \delta$,

$$\|\boldsymbol{x}_{ij}\| \le 4\sqrt{d} + 2\sqrt{\log \frac{1}{\delta}}. \tag{74}$$

Therefore by taking $\delta = \exp(-100 \log n)$, we have that $4\sqrt{d} + 2\sqrt{\log \frac{1}{\delta}} = 4\sqrt{d} + 2\sqrt{100 \log n} = \zeta_x$.

To bound $\|\boldsymbol{x}_{ij}^\top \boldsymbol{X}^{l-1}\|$, note that due to the rotational invariance of the Gaussian random vector $\boldsymbol{x}_{ij}$ (recall that $\boldsymbol{X}^{l-1}$ is an column orthonormal matrix), the $\ell_2$ norm $\|\boldsymbol{x}_{ij}^\top \boldsymbol{X}^{l-1}\|$ is distributed like the $\ell_2$ norm of a Gaussian random vector drawn from $\mathcal{N}(0, \boldsymbol{I}_k)$. Therefore, w.p. at least $1 - n^{-100}$, $\|\boldsymbol{x}_{ij}^\top \boldsymbol{X}^{l-1}\| \le c(\sqrt{k} + \sqrt{\log n}) \doteq \zeta$.

Using the union bound and the fact that $\zeta_x \zeta_y^2 \zeta \le \zeta_0$ leads to the conclusion. $\square$

Conditioned on the above event (clipping takes no effect), to establish the utility guarantee of Algorithm 5, we view `PPM` as a specific instance of the noisy power method (`NPM`) presented in Algorithm 6 Hardt & Price (2014), where the target matrix is $\boldsymbol{A} = (2\Gamma + \text{trace}(\Gamma)\boldsymbol{I}_d)$ with $\Gamma = \boldsymbol{B}^* \boldsymbol{V}^{*\top} \boldsymbol{V}^* \boldsymbol{B}^{*\top}$ and the perturbation matrix $\boldsymbol{G}^l = \boldsymbol{P}_1^l + \boldsymbol{P}_2^l$ is the sum of the noise matrix added by the Gaussian mechanism, $\boldsymbol{P}_2^l = \frac{\sigma_0 \zeta_0}{n} \boldsymbol{W}^l$, and the error matrix $\boldsymbol{P}_1^l = (\boldsymbol{M}^l - \boldsymbol{A})\boldsymbol{X}^{l-1}$. One can easily check that with these choices, we recover line 8 of Algorithm 5

$$\boldsymbol{Y}^l = \boldsymbol{A}\boldsymbol{X}^{l-1} + \boldsymbol{G}^l = \boldsymbol{M}^l \boldsymbol{X}^{l-1} + \frac{\sigma_0 \zeta_0}{n} \boldsymbol{W}^l = \frac{1}{n} \sum_{i=1}^n \boldsymbol{M}_i^l \boldsymbol{X}^{l-1} + \frac{\sigma_0 \zeta_0}{n} \boldsymbol{W}^l. \tag{75}$$

Suppose that the level of perturbation is sufficiently small, we can exploit the following analysis of `NPM` from (Hardt & Price, 2014).

**Theorem G.1** (Adapted from Corollary 1.1 of Hardt & Price (2014))**.** *Consider the noisy power method (NPM) presented in Algorithm 6. Let $\boldsymbol{U} \in \mathbb{R}^{d \times k}$ represent the top-k eigenvectors of the input matrix $\boldsymbol{A} \in \mathbb{R}^{d \times d}$. Suppose that the perturbation matrix $\boldsymbol{G}^l$ satisfies for all $l \in \{1, \dots, L\}$*

$$5\|\boldsymbol{G}^l\| \le \epsilon(s_k(\boldsymbol{A}) - s_{k+1}(\boldsymbol{A})), 5\|\boldsymbol{U}^\top \boldsymbol{G}^l\| \le (s_k(\boldsymbol{A}) - s_{k+1}(\boldsymbol{A})) \frac{C}{\tau \sqrt{kd}} \tag{76}$$

*for some fixed parameter $\tau$ and $\epsilon < 1/2$. Then with all but $1/\tau + e^{-\Omega(d)}$ probability, there exists an $L = \mathcal{O}(\frac{s_k(\boldsymbol{A})}{s_k(\boldsymbol{A}) - s_{k+1}(\boldsymbol{A})} \log(kd\tau/\epsilon))$ so that after $L$ steps we have that $\|(\boldsymbol{I} - \boldsymbol{X}_L \boldsymbol{X}_L^\top)\boldsymbol{U}\| \le \epsilon$. Here $C > 0$ is a constant defined in Lemma I.3.*

To prove that the perturbation matrix $G^l$ satisfies the conditions required by the above theorem, we bound $M^l - A$ and $\frac{\sigma_0 \zeta_0}{n} W^l$ individually, both with high probabilities.

*Proof of Lemma F.1.* Recall that $A = 2\Gamma + \text{trace}(\Gamma) I_d$ and note that $rank(\Gamma) = k$ and that its singular values are $s_1^2, \ldots, s_k^2$. Therefore $s_i(A) = 2s_i^2 + \sum_{j=1}^k s_j^2$ for $i \le k$ and $s_i(A) = \sum_{j=1}^k s_j^2$ for $i > k$.

In this proof, we will show that for both matrices $P_1^l = (M^l - A)X^{l-1}$ and $P_2^l = \frac{\sigma_0 \zeta_0}{n} W^l$ the following inequalities hold for all $l \in \{1, \ldots, L\}$, which is a sufficient condition for Lemma G.1 to hold

$$10\|P^l\| \le \epsilon(s_k(A) - s_{k+1}(A)), 10\|U^\top P^l\| \le (s_k(A) - s_{k+1}(A))\frac{C}{\tau\sqrt{kd}}. \tag{77}$$

**Control terms related to $P_1^l$.** 1. We first bound $\|P_1^l\|$. By the independence between $M^l$ and $X^{l-1}$, we have that

$$\mathbb{E}\left[(M^l - A)X^{l-1}\right] = \mathbb{E}\left[M^l - A\right] \cdot \mathbb{E}\left[X^{l-1}\right] = 0.$$

To bound the norm term $\mathbb{E}\left[\|(M^l - A)X^{l-1}\|\right]$, we begin by controlling the norms of $Z_{ij}^l \cdot X^{l-1}$, where $Z_{ij}^l = y_{ij}^2 x_{ij} x_{ij}^\top$. By the proof for Lemma G.1, with a probability at least $1 - \delta$, we have that $\|Z_{ij}^l \cdot X^{l-1}\| \le O\left((\sqrt{d} + \log 1/\delta) \cdot (\sqrt{k} + \log 1/\delta) \cdot \|w_i^*\|^2\right) \le O\left((\sqrt{d} + \log 1/\delta) \cdot (\sqrt{k} + \log 1/\delta)\mu^2 k s_k^2\right)$ since $\|w_i^*\| \le \mu\sqrt{k}s_k$. We then compute an upper bound on the matrix variance term

$$\|\mathbb{E}\left[(Z_{ij}^l X^{l-1})^\top Z_{ij}^l X^{l-1}\right]\| \le \|\mathbb{E}\left[y_{ij}^4 \|x_{ij}\|^2 (x_{ij}^\top X^{l-1})^\top x_{ij}^\top X^{l-1}\right]\| \tag{78}$$

$$= \|(X^{l-1})^\top \mathbb{E}\left[y_{ij}^4 \|x_{ij}\|^2 x_{ij} x_{ij}^\top\right] X^{l-1}\| \tag{79}$$

Due to isotropy of the Gaussian, by (Tripuraneni et al., 2021, Lemma 5), we have that

$$\mathbb{E}\left[y_{ij}^4 \|x_{ij}\|^2 x_{ij} x_{ij}^\top\right] = \|w_i^*\|^4 \cdot \left((2d + 75)e_1 e_1^\top + (3d + 15)I_d\right) \tag{80}$$

By plugging equation 80 into equation 79, we prove the following inequality.

$$\|\mathbb{E}\left[(Z_{ij}^l X^{l-1})^\top Z_{ij}^l X^{l-1}\right]\| \le \|w_i^*\|^4 \cdot \left((2d + 75)(X^{l-1})^\top e_1 e_1^\top \cdot X^{l-1} + (3d + 15)I_k\right) \tag{81}$$

$$\le O(d\|w_i^*\|^4) \le O(d\mu^2 k s_k^2 \sum_{i=1}^k s_i^2) \tag{82}$$

By combining both the norm bound and the matrix variance bound and using the modified Bernstein matrix inequality (Tripuraneni et al., 2021, Lemma 31), we have

$$\|(M^l - A)X^{l-1}\| \le \log^3 \bar{m}n \cdot \log^3 d \cdot \mathcal{O}\left(\sqrt{\frac{d\mu^2 k s_k^2 \sum_{i=1}^k s_i^2}{\bar{m}n}} + \frac{\sqrt{kd}\mu^2 k s_k^2}{\bar{m}n}\right). \tag{83}$$

2. We then proceed to bound the term $\|U^\top P_1^l\| = \|U^\top(M^l - A)X^{l-1}\|$. By using proof of Lemma G.1, we first bound the norms of $\|U^\top Z_{ij}^l X^{l-1}\| \le O((\sqrt{k} + \log 1/\delta)(\sqrt{k} + \log 1/\delta)\mu^2 k s_k^2)$, where $Z_{ij}^l = y_{ij}^2 x_{ij} x_{ij}^\top$. We then compute an upper bound on the matrix variance term.

$$\|\mathbb{E}[(U^\top Z_{ij}^l X^{l-1})^\top U^\top Z_{ij}^l X^{l-1}]\| \le \|\mathbb{E}\left[y_{ij}^4(U^\top x_{ij} x_{ij}^\top X^{l-1})^\top U^\top x_{ij} x_{ij}^\top X^{l-1}\right]\| \tag{84}$$

$$= \|(X^{l-1})^\top \mathbb{E}\left[y_{ij}^4 \|U^\top x_{ij}\|^2 x_{ij} x_{ij}^\top\right] X^{l-1}\| \tag{85}$$

$$= \|(X^{l-1})^\top \mathbb{E}\left[(w_i^{*\top} B^{*\top} x_{ij})^4 \|U^\top x_{ij}\|^2 x_{ij} x_{ij}^\top\right] X^{l-1}\| \tag{86}$$

$$= \|(X^{l-1})^\top \mathbb{E}\left[(w_i^{*\top} U^\top x_{ij})^4 \|U^\top x_{ij}\|^2 x_{ij} x_{ij}^\top\right] X^{l-1}\|, \tag{87}$$

where the last equality is because by definition, we have $\boldsymbol{B}^* = \boldsymbol{U}$. We now perform variable transformation and denote $\boldsymbol{v}_{ij} = \boldsymbol{V}^\top \boldsymbol{x}_{ij}$ where $\boldsymbol{V} = [\boldsymbol{U}, \boldsymbol{U}']$ is an orthogonal matrix that is extended based on $\boldsymbol{U}$. Therefore, we equivalently write the above equation 87 to the following inequality.

$$\|\mathbb{E}\left[(\boldsymbol{U}^\top \boldsymbol{Z}_{ij}^l \boldsymbol{X}^{l-1})^\top \boldsymbol{U}^\top \boldsymbol{Z}_{ij}^l \boldsymbol{X}^{l-1}\right]\| \tag{88}$$

$$\leq \|(\boldsymbol{V}^\top \boldsymbol{X}^{l-1})^\top \mathbb{E}\left[[\boldsymbol{w}_i^{*\top}(\boldsymbol{v}_{ij}[1], \cdots, \boldsymbol{v}_{ij}[k])^\top]^4 (\sum_{a=1}^k \boldsymbol{v}_{ij}[a]^2) \boldsymbol{v}_{ij} \boldsymbol{v}_{ij}^\top\right] \boldsymbol{V}^\top \boldsymbol{X}^{l-1}\|, \tag{89}$$

where $\boldsymbol{v}_{ij}[a]$ means the $a$-th coordinate of the $k$-dimensional vector $\boldsymbol{v}_{ij}$, which is also distributed as Gaussian. Due to the isotropy of the Gaussian it suffices to compute the expectation assuming $\boldsymbol{w}_i^* \propto \|\boldsymbol{w}_i\| \boldsymbol{e}_1$ where $\boldsymbol{e}_1 = (1, 0, \cdots, 0)^\top$. Then following the proof for (Tripuraneni et al., 2021, Lemma 5), by combinatorics, we have the following equation.

$$\mathbb{E}[(\boldsymbol{w}_i^{*\top}(\boldsymbol{v}_{ij}[1], \cdots, \boldsymbol{v}_{ij}[k])^\top)^4 (\sum_{a=1}^k \boldsymbol{v}_{ij}[a])^2 \boldsymbol{v}_{ij} \boldsymbol{v}_{ij}^\top] \tag{90}$$

$$= \|\boldsymbol{w}_i^*\|^4 \mathbb{E}\left[(\boldsymbol{v}_{ij}[1])^4 (\sum_{a=1}^k \boldsymbol{v}_{ij}[a]^2) \boldsymbol{v}_{ij} \boldsymbol{v}_{ij}^\top\right] = O(\|\boldsymbol{w}_i^*\|^4 k) \tag{91}$$

Therefore, by plugging the above equation into equation 89, we prove the following inequality.

$$\|\mathbb{E}\left[(\boldsymbol{U}^\top \boldsymbol{Z}_{ij}^l \boldsymbol{X}^{l-1})^\top \boldsymbol{U}^\top \boldsymbol{Z}_{ij}^l \boldsymbol{X}^{l-1}\right]\| \leq O(k\|\boldsymbol{w}_i^*\|^4) \leq O(k\mu^2 k s_k^2 \sum_{i=1}^k s_i^2) \tag{92}$$

By combining both the norm bound and the above matrix variance bound and using the matrix Bernstein inequality (Tripuraneni et al., 2021, Lemma 31), we have

$$\|\boldsymbol{U}^\top (\boldsymbol{M}^l - \boldsymbol{A}) \boldsymbol{X}^{l-1}\| \leq \log^3 \bar{m}n \cdot \log^3 d \cdot \mathcal{O}\left(\sqrt{\frac{k^2 \mu^2 s_k^2 \sum_{i=1}^k s_i^2}{\bar{m}n}} + \frac{\mu^2 k^2 s_k^2}{\bar{m}n}\right). \tag{93}$$

Therefore, to ensure that equation 77 holds, it suffices to use $\bar{m}n$ sufficiently large such that

$$\log^3 \bar{m}n \cdot \log^3 d \cdot \mathcal{O}\left(\sqrt{\frac{d\mu^2 k s_k^2 \sum_{i=1}^k s_i^2}{\bar{m}n}} + \frac{\sqrt{kd}\mu^2 k s_k^2}{\bar{m}n}\right) \leq \epsilon(s_k(\boldsymbol{A}) - s_{k+1}(\boldsymbol{A})) \tag{94}$$

$$\log^3 \bar{m}n \cdot \log^3 d \cdot \mathcal{O}\left(\sqrt{\frac{k^2 \mu^2 s_k^2 \sum_{i=1}^k s_i^2}{\bar{m}n}} + \frac{\mu^2 k^2 s_k^2}{\bar{m}n}\right) \leq (s_k(\boldsymbol{A}) - s_{k+1}(\boldsymbol{A}))\frac{C}{\tau\sqrt{kd}} \tag{95}$$

For simplicity, assume that $\sqrt{\frac{d\mu^2 k s_k^2 \sum_{i=1}^k s_i^2}{\bar{m}n}} \geq \frac{\sqrt{kd}\mu^2 k s_k^2}{\bar{m}n}$ and $\epsilon_a \geq 1/\tau\sqrt{kd}$. The above inequalities can be simplified as follows, with $c_1$ being some constant.

$$\frac{\bar{m}n}{\log^6 \bar{m}n} \geq c_1 \frac{d \cdot \log^6 d \cdot k^3 \cdot \mu^2 \cdot \sum_{i=1}^k s_i^2}{s_k^2}. \tag{96}$$

**Control terms related to $\boldsymbol{P}_2^l$.** Recall that $\boldsymbol{P}_2^l \sim \mathcal{N}(0, \sigma^2)^{d \times k}$, with $\sigma = \sigma_0 \zeta_0/n$. Using Lemma I.2, we have with probability $1 - 2e^{-x}$:

- $\max_{l \in [L]} \|\boldsymbol{P}_2^l\| \leq C_\mathcal{N} \sigma(\sqrt{d} + \sqrt{p} + \sqrt{\ln(2L + x)})$;

- $\max_{l \in [L]} \|\boldsymbol{U}^\top \boldsymbol{P}_2^l\| \leq C_\mathcal{N} \sigma(2\sqrt{p} + \sqrt{\ln(2L + x)})$.

Consequently, we can obtain bound the terms related to $\boldsymbol{P}_2^l$ by setting $\sigma_0$ sufficiently small such that the following inequalities hold with $x = 100 \log n$

$$10 C_\mathcal{N} \sigma(\sqrt{d} + \sqrt{k} + \sqrt{\ln(2L + x)}) \leq \epsilon_a s_k^2, \quad 10 C_\mathcal{N} \sigma(2\sqrt{k} + \sqrt{\ln(2L + x)}) \leq s_k^2/(\tau\sqrt{dk}).$$

To simplify the above inequalities, suppose that $\epsilon_a$ is a constant and neglect the $\log \log$ terms. We obtain

$$\frac{c_2 \sigma_0 \zeta_0}{n} \leq \frac{s_k^2}{\tau k \sqrt{d} \sqrt{\log L}}. \tag{97}$$

for some constant $c_2$.

Having established equation 77 for both $\boldsymbol{P}_1^l$ and $\boldsymbol{P}_2^l$, we can then use Lemma G.1 to obtain the target result.

$\square$

## H    PROOF OF LEMMA F.3

*Proof.* In the following, we first show that 1. the index $\hat{c}$ exists with a high probability and we then show that 2. any candidate $\boldsymbol{B}_{\hat{c}}$ that passes the test equation 68 has utility no less than $b$, i.e. $\text{dist}(\boldsymbol{B}_{\hat{c}}, \boldsymbol{B}^*) \leq b$.

**Existence of $\hat{c}$**    Suppose that both $\boldsymbol{B}_{c_1}$ and $\boldsymbol{B}_{c_2}$ are successful, i.e. $\text{dist}(\boldsymbol{B}_{c_i}, \boldsymbol{B}^*) \leq a$ for $i = 1, 2$ or equivalently $s_{\min}(\boldsymbol{B}_{c_i}^\top \boldsymbol{B}^*) \geq \sqrt{1 - a^2}$ for $i = 1, 2$. Recall that $\boldsymbol{B}^* \boldsymbol{B}^{*\top} + \boldsymbol{B}_\perp^* \boldsymbol{B}_\perp^{*\top} = \boldsymbol{I}_d$. Compute that

$$
\begin{aligned}
&s_{\min}(\boldsymbol{B}_{c_1}^\top \boldsymbol{B}_{c_2}) \\
&= s_{\min}\left( \boldsymbol{B}_{c_1}^\top (\boldsymbol{B}^* \boldsymbol{B}^{*\top} + \boldsymbol{B}_\perp^* \boldsymbol{B}_\perp^{*\top}) \boldsymbol{B}_{c_2} \right) \\
&\geq s_{\min}(\boldsymbol{B}_{c_1}^\top \boldsymbol{B}^* \boldsymbol{B}^{*\top} \boldsymbol{B}_{c_2}) - s_{\max}(\boldsymbol{B}_{c_1}^\top \boldsymbol{B}_\perp^* \boldsymbol{B}_\perp^{*\top} \boldsymbol{B}_{c_2}) \\
&\geq s_{\min}(\boldsymbol{B}_{c_1}^\top \boldsymbol{B}^*) s_{\min}(\boldsymbol{B}^{*\top} \boldsymbol{B}_{c_2}) - s_{\max}(\boldsymbol{B}_{c_1}^\top \boldsymbol{B}_\perp^*) s_{\max}(\boldsymbol{B}_\perp^{*\top} \boldsymbol{B}_{c_2}) \geq 1 - 2a^2. \tag{98}
\end{aligned}
$$

Define the binomial random variable $X_c = \mathbf{1}_{\boldsymbol{B}_c \text{ is successful}}$. We have that $\mathbb{E}[X_c] = \mathbb{P}\{\boldsymbol{B}_c \text{ is successful}\} \geq p$.

Using the concentration of the binomial random variable, we have

$$\mathbb{P}\{\sum_{c=1}^{T_0} X_c \leq T_0 \cdot \mathbb{E}[X_c] - t\} \leq \exp(-t^2/(2T_0 p)). \tag{99}$$

Therefore, with the choice $t = T_0/2$ and $T_0 \geq 8p \log 1/\delta$, we have with probability at least $1 - \delta$, at least half of the outputs of $T_0$ independent NPM runs are successful. Consequently, there exists at least $T_0/2$ pairs of $\boldsymbol{B}_{c_1}$ and $\boldsymbol{B}_{c_2}$ such that Eq. equation 98 holds, which shows the existence of $\hat{c}$.

**Utility of $\boldsymbol{B}_{\hat{c}}$**    We now show that the candidate $\boldsymbol{B}_{\hat{c}}$ that passes the test Eq. equation 68 must satisfy $\text{dist}(\boldsymbol{B}_{\hat{c}}, \boldsymbol{B}^*) \leq b$. We prove via contradiction. Suppose that there exists a candidate $\boldsymbol{B}$ that passes the test, but with $\text{dist}(\boldsymbol{B}, \boldsymbol{B}^*) > b$. This means that there exists $\hat{x} \in \mathbb{R}^k$ and $\hat{y} \in \mathbb{R}^k$ with $\|\hat{x}\|_2 = \|\hat{y}\|_2 = 1$ achieving the minimum singular value of $\boldsymbol{B}^\top \boldsymbol{B}^*$, such that $\langle \boldsymbol{B}\hat{x}, \boldsymbol{B}^*\hat{y} \rangle \leq \sqrt{1 - b^2}$.

Let $\boldsymbol{B}_i$ be a successful candidate, i.e. $\text{dist}(\boldsymbol{B}_i, \boldsymbol{B}^*) \leq a$ (note that with a high probability they are in the majority according to the discussion above). We have

$$s_{\min}(\boldsymbol{B}^\top \boldsymbol{B}_i) = s_{\min}(\boldsymbol{B}^\top \left( \boldsymbol{B}^* \boldsymbol{B}^{*\top} + \boldsymbol{B}_\perp^* \boldsymbol{B}_\perp^{*\top} \right) \boldsymbol{B}_i) \leq s_{\min}(\boldsymbol{B}^\top \boldsymbol{B}^* \boldsymbol{B}^{*\top} \boldsymbol{B}_i) + s_{\max}(\boldsymbol{B}^\top \boldsymbol{B}_\perp^* \boldsymbol{B}_\perp^{*\top} \boldsymbol{B}_i).$$

For the second term, we have $s_{\max}(\boldsymbol{B}^\top \boldsymbol{B}_\perp^* \boldsymbol{B}_\perp^{*\top} \boldsymbol{B}_i) \leq s_{\max}(\boldsymbol{B}^\top \boldsymbol{B}_\perp^*) \cdot s_{\max}(\boldsymbol{B}_\perp^{*\top} \boldsymbol{B}_i) \leq 1 \cdot a \leq a$. To bound the first term, recall the variational formulation of the minimum singular value $s_{\min}(A) = \min_{\|x\|=\|y\|=1} x^\top A y$ and hence

$$s_{\min}(\boldsymbol{B}^\top \boldsymbol{B}^* \boldsymbol{B}^{*\top} \boldsymbol{B}_i) \leq \hat{x}^\top \boldsymbol{B}^\top \boldsymbol{B}^* \boldsymbol{B}^{*\top} \boldsymbol{B}_i \hat{y} = \hat{x}^\top \boldsymbol{B}^\top \boldsymbol{B}^* \hat{y} + \hat{x}^\top \boldsymbol{B}^\top \boldsymbol{B}^* (\hat{y} - \boldsymbol{B}^{*\top} \boldsymbol{B}_i \hat{y})$$

$$\leq \sqrt{1 - b^2} + \hat{x}^\top \boldsymbol{B}^\top \boldsymbol{B}^* (\hat{y} - \boldsymbol{B}^{*\top} \boldsymbol{B}_i \hat{y}) \leq \sqrt{1 - b^2} + \|\boldsymbol{I}_k - \boldsymbol{B}^{*\top} \boldsymbol{B}_i\|_2 \leq \sqrt{1 - b^2} + 1 - \sqrt{1 - a^2}.$$

where we recall the definitions of $\hat{x}$ and $\hat{y}$ in the above paragraph. Combining the above bounds, we obtain $s_{\min}(\boldsymbol{B}^\top \boldsymbol{B}_i) \leq \sqrt{1 - b^2} + 1 - \sqrt{1 - a^2} + a$ which is strictly smaller than $1 - 2a^2$ for a sufficiently small $a$ and a sufficiently large $b$, e.g. $a = 0.01$ and $b = 0.2$. To put it in other words, we obtain that $\boldsymbol{B}$ will fail test (68). This leads to a contradiction and hence we have proved that any candidate that passes the test (68) must satisfy $\text{dist}(\boldsymbol{B}_{\hat{c}}, \boldsymbol{B}^*) \leq b$. $\square$

## I   PRELIMINARY ON MATRIX CONCENTRATION INEQUALITIES

**Lemma I.1** (Theorem 4.4.5 in Vershynin (2018)). *Let $G_1, \ldots, G_L \sim \mathcal{N}(0, \sigma^2)^{d \times n}$. There exists a constant $C_{\mathcal{N}}$ such that with probability at least $1 - e^{-x}$,*

$$\max_{l \in [L]} \|G_l\| \leq C_{\mathcal{N}} \sigma (\sqrt{n} + \sqrt{d} + \sqrt{\ln(2L + x)}). \tag{100}$$

**Lemma I.2** (Lemma A.2 of Hardt & Price (2014)). *Let $U \in \mathbb{R}^{d \times p}$ be a matrix with orthonormal columns. Let $G_1, \ldots, G_L \sim \mathcal{N}(0, \sigma^2)^{d \times p}$ with $0 \leq p \leq d$. There exists a constant $C_{\mathcal{N}}$ such that with probability at least $1 - e^{-x}$,*

$$\max_{l \in [L]} \|U^\top G_l\| \leq C_{\mathcal{N}} \sigma (2\sqrt{p} + \sqrt{\ln(2L + x)}). \tag{101}$$

**Lemma I.3** (Minimum Singular Value of a Square Gaussian Matrix (Theorem 1.2 of Rudelson & Vershynin (2008))). *Let $A \in \mathbb{R}^{k \times k}$ be a Gaussian random matrix, i.e. $A_{ij} \sim \mathcal{N}(0, 1)$. Then, for every $\epsilon > 0$, we have*

$$\Pr\{s_{\min}(A) \leq \epsilon k^{-1/2}\} \leq C\epsilon + c^k, \tag{102}$$

*where $C > 0$ and $c \in (0, 1)$ are absolute constants.*

## J   AN ELABORATED DISCUSSION ON THE UTILITY-PRIVACY TRADEOFF IN COROLLARY 5.1

Consider the result stated in Corollary 5.1.

- Suppose that the left hand side $\tilde{c}_t \kappa k^{1.5} \mu^2 d/n$ is fixed, then for a target accuracy $\epsilon_a$, we *cannot* establish the theoretical guarantee that CENTAUR achieves the accuracy $\epsilon_a$ within a DP budget of $\epsilon_{dp} \leq \frac{\tilde{c}_t \kappa k^{1.5} \mu^2 d}{n \epsilon_a}$ (this is natural since a smaller DP budget $\epsilon_{dp}$ requires a larger noise multiplier $\sigma_g$ which jeopardizes the convergence analysis of CENTAUR). However, we need to emphasize that, we are not ruling out the possibility that such a DP budget $\epsilon_{dp}$ is achieved, since the privacy guarantee that we are establishing is just an upper bound. Hence we are *not* establishing a lower bound.

- Now suppose that all factors other than the number of clients $n$ is fixed. Corollary 5.1 implies that for an $n$ that is sufficiently large, i.e. $n \geq \frac{\tilde{c}_t \kappa k^{1.5} \mu^2 d}{\epsilon_a \cdot \epsilon_{dp}}$, we can establish the guarantee that the output of CENTAUR achieves an $\epsilon_a$ utility within an $\epsilon_{dp}$ budget. This interpretation also allows us to understand the benefit of having a better dependence on $d$: A better dependence on $d$ means that a smaller $n$ is sufficient to achieve the same utility-privacy guarantee.

## K   DISCUSSION ON THE REQUIRED ASSUMPTIONS

In this section, we show that the requirements we made in Assumption 5.1 to 5.3 are similar to the assumptions made in (Collins et al., 2021) and (Jain et al., 2021).

**Discussion on Assumption 5.1** We note that our Assumption 5.1 is the same as Assumption 1 in (Collins et al., 2021), and is similar to point (i) of Assumption 4.1 in (Jain et al., 2021) where $x_{ij}$ is assumed to be exactly Gaussian.

**Discussion on Assumption 5.2** We note that our Assumption 5.2 is the same as the definition of the incoherence parameter $\mu$ in (Jain et al., 2021) (the parameter $\lambda_k$ therein is equivalent to $\sigma_k^2$ in our paper), and is similar to Assumption 3 in (Collins et al., 2021) where the incoherence parameter $\mu$ as well as $\sigma_k$ is assumed to be 1.

**Discussion on Assumption 5.3** We focus on the dependence on the parameters $d$ and $n$ while treating log terms and the other parameters like the rank $k$ and the incoherence parameter $\mu$ as constants. In this case, Assumption 5.3 can be simplified as $m = \Omega(d/n)$. We note that, under this setting our Assumption 5.3 is the same as the requirement (12) in (Collins et al., 2021) and Lemma 4.6 in (Jain et al., 2021). The equivalence to the requirement (12) in (Collins et al., 2021) is straight forward.

To see the equivalence to Lemma 4.6 in (Jain et al., 2021), note that in order for the convergence analysis of the main procedure to hold, the initializer $\boldsymbol{U}^{init}$ in (Jain et al., 2021) should satisfy $\|\boldsymbol{U}_{\perp}^{\top}\boldsymbol{U}^{init}\|_{F} = O(1)$. To achieve this, the R.H.S. of Lemma 4.6 in (Jain et al., 2021) should be bounded by a constant, which means that $m = \Omega(d/n)$.

