# OpenReview forum: "Share Your Representation Only: Guaranteed Improvement of the Privacy-Utility Tradeoff in Federated Learning"
_ICLR.cc/2023/Conference — ICLR 2023 poster_

### Official Review · Reviewer_4btx · 2022-10-27

**Confidence:** 3
**Correctness:** 4
**Technical Novelty And Significance:** 3
**Empirical Novelty And Significance:** 3
**Recommendation:** 8

**Clarity, Quality, Novelty And Reproducibility:**

The ideas and results are clearly presented.

The algorithmic ideas and experiment results are promising and exceed previous works.

It seems that the code is not provided via link nor in the supplementary materials.

**Strength And Weaknesses:**

Strengths:
- The idea of separately training representation and classification heads is intuitively nice.
- The experiment results look promising. The proposed algorithm outperforms the DP-FedAvg and existing personalized FL baselines, and is the only method that outperforms local training.
- Theoretical results demonstrate significant improvement for linear networks.


Weakness:
- The experiment results would be more convincing if more experiments on done on larger datasets like ImageNet.
- This is just a curious question: how would the size of the classification head and representation network affect the performance, and what is a good way to select these parameters in practice? Would it depend on the training task, e.g. for simple classification it suffices to use a small head, but for more complicated tasks larger head is needed?


=== Update ===

After reading other reviewers' comments, it seems that there is a mismatch between the error metric used in this work (matrix L2 norm) and Jain et.al (Frobenius norm). Therefore, the theoretical improvement is unclear. I'm adjusting my score to 6 accordingly.

=== Update ===
The authors have addressed my concern about the mismatch in the error metric. I also believe the authors have adequately addressed most concerns of other reviewers. Thus I will change back to my original score of 8.

**Summary Of The Paper:**

This work proposes studies federated learning under differential privacy with heterogeneous users, where the goal is to learn simultaneously learn good global and local models. The authors propose to only share weights of the representation networks and locally train the user-specific heads for personalization. Theoretical analysis is provided for linear networks, and experiment results verify the effectiveness of the approach.

**Summary Of The Review:**

This work proposes a framework for DP federated learning that separately trains the representation network and local heads. The performance surpasses previous works. The theoretical results for linear networks intuitively prove the effectiveness of the idea. I only have some minor issues stated in the weakness part.

---

> ### Author Response · Authors · 2022-11-17
> **Response to review**
>
> We thank the reviewer for the positive feedback.
>
> ===== update =====\
> Dear Reviewer 4btx,
>
> We thank you for updating your review. In your update, you mentioned that "there is a mismatch between the error metric used in this work (matrix L2 norm) and Jain et.al (Frobenius norm)", and you adjusted your score to 6. However, we would like to point out that Reviewer Usxb, who had the same concern, has accepted our response, and raised the score accordingly (so the concern was addressed). We really hope that you can consider this in further updating your score. Please find below our response to the concern:
>
>
> [About the matrix norm that appears in the conclusions] Since the matrix under consideration, i.e. $B_\perp^\top B^* \in \mathbb{R}^{d \times k}$, is of low rank, we can bound its Frobenius norm by its spectral norm with an extra $\sqrt{k}$ factor, which does not change the dependence on $d$ ($k$ is regarded as a small constant in the literature of matrix factorization). Moreover, with this extra $\sqrt{k}$ factor, Theorem 5.1 in our paper and Lemma 4.4 in (Jain et al. 2021) have the same quadratic dependence on $k$, while the dependency on $d$ is substantially reduced, from $d^{1.5}$ to $d$ (where $NSR =0$ since we focus on the noiseless setting).

---

> > ### Comment · Reviewer_4btx · 2022-11-21
> > **Thank you for your response.**
> >
> > You have resolved my concerns. I will update my score accordingly.

---

### Official Review · Reviewer_CCKF · 2022-10-28

**Confidence:** 4
**Correctness:** 2
**Technical Novelty And Significance:** 2
**Empirical Novelty And Significance:** 2
**Recommendation:** 6

**Clarity, Quality, Novelty And Reproducibility:**

Summary (see previous section for details):
- Clarity: well-written, but misleading presentation.
- Quality: some minor technical issues. Expanding the discussion of utility bound and/or empirical results can help.
- Novelty: limited.

**Strength And Weaknesses:**

## Strengths
- The presentation is crisp.
- The $O(\sqrt d)$ improvement upon the method of Jain et al. is interesting (though the comparison may not be fair).
- The empirical results are promising.

## Weaknesses

My main concern is that the vast majority of ideas, which are presented as novel, are not new at all, they have been studied both in the federated learning literature and under differential privacy.

The idea of learning shared representations differentially privately while learning a personalized model for each user/client is exactly the setting of private model personalization (Jain et al. and others). In fact, they describe a general alternating minimization algorithm (their Algorithm 1), and explicitly mention DP-SGD as one possible way of solving the server-side problem, which is what Algorithm 1-2 in this paper does. Granted, their analysis is carried only for sufficient statistics perturbation and not gradient descent, so the utility bound in Theorem 5.1 is new in this setting. But the framework and ideas presented in the paper are well-established, for example arguments made in Sections 1-3 are well known.

The contribution of the paper is much more narrow than what the paper claims:
1) Proving a utility bound (which includes a \sqrt d improvement upon the SOTA) in the linear case, *in the low-privacy regime* (more discussion would help here, to understand how restrictive this condition is, for example some estimate of how large epsilon has to be in realistic scenarios).
2) Good empirical results on CIFAR and EMNIST.

I believe the presentation is misleading, and should be significantly updated to reflect the real contributions. All discussion up to and including Section 3 should be presented as a review of existing ideas.

On the technical side: the statement of results lacks rigor at times.
- The privacy guarantee should be stated more carefully. The rigorous guarantee should use the joint-(R)DP notion.
- Corollary 5.1 should be clearly stated -- how should one interpret statements such as "we have the following trade-off"? It is unclear, as stated, what conditions should hold, what parameters are fixed, what parameters need to be known for running the algorithm, etc.
- As pointed out by other reviewers, the assumptions for the main result are not the same as Jain et al., this must be further discussed.

#### Experiments:
The experimental results are promising. I have a few questions/suggestions:
- Did the authors compare to DP-FedAvg-fb of Yu et al.? (i.e. fine-tuning the last layer instead of the entire model)? This appears closer to the setting under consideration.
- It is always worrisome to see that all competing methods are worse than even the non-FL baseline. Does one really expect all these other methods to fail in practice? Surely there is a regime in which isolated training would fail (say in the extreme case of one example per client). Can the authors give a more nuanced discussion?


==== Post-discussion update ====
The authors acknowledged that the idea of learning shared DP representation while learning a personalized classifier per client, is not new, and promised to rephrase their contributions to reflect this, and to better place the work in the context of existing work.
Other improvements have been made, including discussions about the technical assumptions, and empirical support for the claims made in remark 3.1.

**Summary Of The Paper:**

The paper considers the (federated) private model personalization problem, and provides a utility guarantee in the bilinear case.
The technical contribution is to prove a utility bound when using DPSGD to learn the shared representation, which improves upon the best known bound (that uses sufficient statistics perturbation instead) by a factor of $\sqrt d$.

**Summary Of The Review:**

Ideas are not new. The improvement of utility bound is interesting (but the low privacy condition needs to be a clear caveat) and the empirical results are promising (but need a perhaps more nuanced discussion).

---

> ### Author Response · Authors · 2022-11-17
> **Response to review (4/4)**
>
> > As pointed out by other reviewers, the assumptions for the main result are not the same as Jain et al., this must be further discussed.
>
> We are considering the noiseless case where the noise level parameter $\sigma_F$ in (Jain et al., 2021) is set to $0$. We will clarify this in our revision. About the matrix norm that appears in the conclusions, since the matrix under consideration, i.e. $B_\perp^\top B^* \in \mathbb{R}^{d \times k}$, is of low rank, we can bound its Frobenius norm by its spectral norm with an extra $\sqrt{k}$ factor, which does not change the dependence on $d$ ($k$ is regarded as a small constant in the literature of matrix factorization). Moreover, with this extra $\sqrt{k}$ factor, Theorem 5.1 in our paper and Lemma 4.4 in (Jain et al. 2021) have the same quadratic dependence on $k$, while the dependency on $d$ is substantially reduced, from $d^{1.5}$ to $d$ (where $NSR =0$ since we focus on the noiseless setting).
>
>
> > Did the authors compare to DP-FedAvg-fb of Yu et al.? (i.e. fine-tuning the last layer instead of the entire model)? This appears closer to the setting under consideration.
>
> Yes, DP-FedAvg-ft (second column of Table 1) is exactly the method described by the reviewer.
>
>
> > It is always worrisome to see that all competing methods are worse than even the non-FL baseline. Does one really expect all these other methods to fail in practice? Surely there is a regime in which isolated training would fail (say in the extreme case of one example per client). Can the authors give a more nuanced discussion?
>
> The reviewer is absolutely correct that there exist extreme scenarios where private FL methods perform better than the non-FL baseline. Specifically, for each method, assuming that the number of records per-client is fixed, then there is a transition point (for the tuple of  the number of clients $n$ and differential privacy budget $\epsilon_{dp}$) after which the private FL algorithm performs better than the non-FL baseline.
> *  Suppose the privacy budget $\epsilon_{dp}$ is fixed. Then as the number of clients $n$ tends to infinity, the standard deviation of injected noise in each update scales with $O(1/n) \rightarrow 0$. Under this increasingly small noise, the private FL method would have a performance similar to the standard (non-DP) FL method, which is stronger than non-FL baseline.
> * Suppose the number of clients $n$ is fixed. Then as the differential privacy budget $\epsilon_{dp}$ tends to infinity, the standard deviation of the injected noise in each update scales as $O(1/\epsilon_{dp}) \rightarrow 0$. Under this increasingly small noise, the private FL method would have a performance similar to the standard (non-DP) FL method, which is stronger than non-FL baseline.
>
> However, note that for a realistic dataset, to maintain a fixed per-client record number, the total number of clients $n$ is limited since we only have finite data points as a whole. This precludes the possibility of growing $n$ to infinity. Also, in any case of real-world interest, $\epsilon_{dp}$ cannot be infinite. \
> In our experiments, for a fixed DP budget $\epsilon_{dp}$ (Table 1), we observe that this transition point for the number of clients $n$ is smaller for CENTAUR than for other FL methods. That is, we require less stringent assumptions on the number of clients to beat the non-FL baseline. For other FL methods, this transition point is too large to be reached, and hence they perform worse than the non-FL baseline\
> On the other hand, suppose that the number of clients $n$ is fixed. We observe in Figure 1 that the transition point for the DP budget $\epsilon_{dp}$ is smaller for CENTAUR than other FL methods. That is, for a reasonable set of differential privacy budget values $\epsilon_{dp}$, CENTAUR has a higher accuracy than all competing private FL methods, and outperforms the non-FL baseline most of the time ($\epsilon >= 0.5$).

---

> > ### Comment · Reviewer_CCKF · 2022-11-29
> > **Re: Response to review (4/4)**
> >
> > Thank you for the discussion.
> >
> > > Yes, DP-FedAvg-ft (second column of Table 1) is exactly the method described by the reviewer.
> >
> > Thank you for confirming. Please note that they describe two fine-tuning methods: the **FT** method fine-tunes the entire model, while the **FB** method (for freeze-base) fine-tunes the last layer. To avoid any confusion, it may be useful to re-label the method as DP-FedAvg-**fb**.
> >
> > ### Regarding the experimental setting
> > I appreciate the argument that you are constrained by the size of the data sets you chose (in terms of number of clients n). But there are other design parameters that one can adjust: for example, increasing $\epsilon_{DP}$ can simulate increasing $n$ (as apparent from Corollary 5.1). Also decreasing the number of examples per class per client: it seems to be around 20-25 in the current setting. It would be useful to evaluate in a sparser regime.
> >
> > The higher-level point is that evaluation should be more nuanced. I hope the authors can at least comment that the conclusions hold for this specific setting, and may be different under different sparsity and noise levels.

---

> ### Author Response · Authors · 2022-11-17
> **Response to review (3/4)**
>
>
>
> > All discussion up to and including Section 3 should be presented as a review of existing ideas.
>
> We acknowledge that the representation learning formulation Eq. (3) has been considered in the prior works: Jain et al. (2021) considered the representation learning problem with DP restriction, but only in the linear representation setting; Collins et al. (2021) considered this problem in the general federated learning setting, but without DP guarantees. These works are discussed in the related work section as well as the problem formulation section.\
> However, we believe our discussions in Remark 3.1 are new with regard to the following two points: 1.how gradient clipping and noise injection affect the standard FL training in the data heterogeneous setting and 2.how the representation learning formulation is particularly suitable for FL training in the data heterogeneous setting. To see this:
> * While Jain et al. (2021) provides the DP guarantee, they did not discuss how data heterogeneity affects federated training;
> * Collins et al. (2021) did not consider the DP setting at all.
> In fact, our research is motivated by this new consideration that representation learning formulation can alleviate the negative effect of data heterogeneity in differentially private federated learning. Therefore it should be emphasized on its own.
>
>
>
>
> > The privacy guarantee should be stated more carefully. The rigorous guarantee should use the joint-(R)DP notion.
>
> We acknowledge that our current differential privacy guarantee holds under the very special threat model of billboard model, where only the shared representations b_{T_g} in Algorithm 1 are released as output, while the personalized classifier of each user in Algorithm 2 is never released. Under this threat model, we prove the usual (R\’enyi) differential privacy guarantee for releasing the shared representations b_{T_g} via standard composition theorems of components in the algorithm.
>
> That said, under a different threat model assumption where all the personalized models w_i^{T_l} of individual users are also released as outputs of the Algorithm 2, then our bound needs to be restated equivalently as a joint-(R)DP guarantee.
>
> We have added more clarifications in the revised paper about: 1) what are released in Algorithm 1 and 2; and 2) in the case of a different threat model where all the personalized models of individual users are released, our differential privacy bound is equivalent to the notion of joint-(R)DP guarantee widely studied in the literature.
>
>
> > Corollary 5.1 should be clearly stated -- how should one interpret statements such as "we have the following trade-off"? It is unclear, as stated, what conditions should hold, what parameters are fixed, what parameters need to be known for running the algorithm, etc.
>
> We apologize for the confusion caused by Corollary 5.1. We are **not** establishing a lower bound on the DP budget $\epsilon_{dp}$. Instead, Corollary 5.1 states that CENTUAR outputs a solution that **provably** achieves an $\epsilon_a$ utility within an $\epsilon_{dp}$ budget, under the condition that the tuple $(\epsilon_a, \epsilon_{dp})$ satisfies the inequality ${\tilde c_t \kappa k^{1.5} \mu^2d}/{n}  \leq \epsilon_a \cdot \epsilon_{dp}$. \
> To elaborate a bit more:
> 1. Suppose that the left hand side ${\tilde c_t \kappa k^{1.5} \mu^2d}/{n}$ is fixed, then for a target accuracy $\epsilon_{a}$, we **cannot** establish the theoretical guarantee that CENTAUR achieves the accuracy $\epsilon_a$ within a DP budget of $\epsilon_{dp} \leq \frac{\tilde c_t \kappa k^{1.5} \mu^2d}{n \epsilon_{a}}$  (this is natural since a smaller DP budget $\epsilon_{dp}$ requires a larger noise multiplier $\sigma_g$ which jeopardizes the convergence analysis of CENTAUR). However, we need to emphasize that, we are not ruling out the possibility that such a DP budget $\epsilon_{dp}$ is achieved, since the privacy guarantee that we are establishing is just an upper bound. Hence we are **not** establishing a lower bound.
> 2. Now suppose that all factors other than the number of clients $n$ is fixed. Corollary 5.1 implies that for an $n$ that is sufficiently large, i.e. $n \geq \frac{\tilde c_t \kappa k^{1.5} \mu^2d}{\epsilon_a \cdot \epsilon_{dp}}$, we can establish the guarantee that the output of CENTAUR achieves an $\epsilon_a$ utility within an $\epsilon_{dp}$ budget. This interpretation also allows us to understand the benefit of having a better dependence on $d$: A better dependence on $d$ means that a smaller $n$ is sufficient to achieve the same utility-privacy guarantee.
>
> Now to demonstrate what the utility-privacy tradeoff suggests in practice, we conduct an empirical study which is presented in Figure 1 (page 2) of our paper. It shows that under a fixed privacy budget, CENTAUR achieves a uniformly better accuracy, on the EMNIST dataset.

---

> > ### Comment · Reviewer_CCKF · 2022-11-29
> > **Re: Response to review (3/4)**
> >
> > Regarding the novelty of the setting: please see my earlier [comment](https://openreview.net/forum?id=oJpVVGXu9i&noteId=xdbAQrrea9A)
> >
> > > These works are discussed in the related work section as well as the problem formulation section.
> >
> > I don't see these works discussed in the problem formulation (Section 3). When the model is decomposed into shared and local parameters, it should be mentioned that this follow prior works including Jain et al., Collins et al. (and the same idea is also present in Singhal et al. (2021)).
> >
> > The revised statement of Corollary 5.1 makes sense.

---

> ### Author Response · Authors · 2022-11-17
> **Response to review (2/4)**
>
> > more discussion would help here, to understand how restrictive this condition is, for example some estimate of how large epsilon has to be in realistic scenarios
>
> This is a great question. From a theoretical perspective, in our paper, we require the privacy budget $\epsilon_{dp}$ to satisfy $\epsilon_{dp} \geq c_1 d/n$ where we hide the dependence on other parameters like the rank $k$ and the incoherence parameter $\mu$ in $c_1$. In contrast, in Jain et al. (2021) (please see Assumption 4.1 therein), the acceptable privacy budget $\epsilon_{dp}$ should satisfy $\epsilon_{dp} \geq c_2 d^{1.5}/n$, where again $c_2$ hides the dependence on other parameters. Consequently, we have a weaker restriction on the acceptable privacy budget, compared to the previous SOTA.
> From a practical perspective, we believe this restriction very mild, since we usually expect the number of clients that participate in the training procedure to be very large in federated learning.

---

> > ### Comment · Reviewer_CCKF · 2022-11-29
> > **Re: Response to review (2/4)**
> >
> > Thank you for the discussion. This point has been addressed.

---

> ### Author Response · Authors · 2022-11-17
> **Response to review (1/4)**
>
> We thank the reviewer for carefully going over our paper and making helpful suggestions in the review. We have incorporated the suggestions in our revision and invite the reviewer to take a look. In the following, we address the major comments from the reviewer one by one.
>
> > My main concern is that the vast majority of ideas, which are presented as novel, are not new at all, they have been studied both in the federated learning literature and under differential privacy.\
> The idea of learning shared representations differentially privately while learning a personalized model for each user/client is exactly the setting of private model personalization (Jain et al. and others). In fact, they describe a general alternating minimization algorithm (their Algorithm 1), and explicitly mention DP-SGD as one possible way of solving the server-side problem, which is what Algorithm 1-2 in this paper does.
>
> We state the additional **algorithmic** novelty of our work (besides the contribution in our theoretical analysis) by comparing with the previous works Jain et al. (2021) and Collins et al. (2021):
>
> 1. [Comparing with Jain et al. (2021), main procedure] While the main procedure of Jain et al. (2021) is a general alternating minimization algorithm, it requires the server and the clients to solve their own subproblems. While the minimization of the subproblem on the client end can be efficiently conducted, solving the minimizing subproblem on the server side is expense even with the DP-SGD as a subrountine: To carry out line 6 of Algorithm 1 in Jain et al. (2021) with DP-SGD, every iteration of DP-SGD requires a round of communication between the server and the participating clients (since we are in the federated learning setting, the data are privately held by the clients and hence the gradient can only be computed on the client side). Moreover, if we solve the subproblem on the server side with DP-SGD, the resulting algorithm is a **double loop** method, and tuning the number of steps in the inner loop introduces extra difficulties.\
> In contrast, CENTAUR is a **single loop** method, and requires a single communication round per iteration.
> 2. [Comparing with Jain et al. (2021), initialization procedure] The initialization procedure is crucial for solving the linear representation learning problem as the main procedure heavily relies on the fact that the initial point $U^{0}$, i.e. the output from the initialization procedure, is already constantly close to the optimal solution (e.g. $\mathrm{dist}(U^0, U^*)\leq 0.1$) for both Jain et al. (2021) and our paper. To achieve this, Jain et al. (2021) directly uses the result from the DP-SVD method. A key algorithmic drawback of this approach is the requirement of **transmitting a $d \times d$ sized matrix**, which is prohibitive if $d$ is large.\
> In contrast, CENTAUR only requires **transmitting matrices of size $d \times k$** for a **logarithmic** amount of iterations.
> 3. [Comparing with Collins et al. (2021), main procedure] We acknowledge that the design of the Server procedure resembles prior work Collins et al. (2021) as it is built on Gaussian mechanism. However, this is a common phenomenon in the literature of differential privacy, e.g. DP-SGD is adapted from SGD by adding the Gaussian mechanism and Jain et al. (2021) is adapted from the alternating minimization method from the literature of matrix factorization. In fact, this is the advantage of using Gaussian mechanism: If we know the problem of interest can be efficiently solved by a non-DP method, Gaussian mechanism allows us to enhance the non-DP method with DP guarantee, but with a minimum change of the method.
> 4. [Comparing with Collins et al. (2021), initialization procedure] Since Collins et al. (2021) considers the non-DP setting, its initialization procedure, which is in essence an SVD step, has **no DP** guarantee. A simple idea would be to replace the SVD with Private Power Method (PPM), Hardt & Price (2014). However, PPM itself only has a constant success probability and is hence insufficient for the purpose of initializing CENTUAR in the linear representation learning case. \
> In contrast, the Initialization procedure of CENTAUR (Algorithm 3) that allows us to establish the $\sqrt{d}$ improvement over Jain et al. (2021) is of great novelty: As articulated in the contribution section (please see the last sentence of the first paragraph on page 2), a novel cross validation scheme is designed to boost the success probability of Private Power Method (PPM). Specifically, given the outputs of a logarithmic amount of i.i.d. PPM trials, the proposed procedure, as post-processing, boosts the success probability of PPM without introducing extra privacy loss.

---

> > ### Comment · Reviewer_CCKF · 2022-11-29
> > **Regarding contributions**
> >
> > Indeed, there are differences between the algorithm analyzed in Jain et al. and the algorithm analyzed in this paper. The point that I tried to make is that **the idea of sharing representations only (the very title of the current paper) while learning a personalized model per client is not new**.
> >
> > The presentation makes it appear as though this is a new idea. Quoting the contributions paragraph in the introduction (emphasis mine):
> > > we propose a novel framework named CENTAUR that is based on *learning information that is agreed on among most parties in a differentially private manner, while also allowing each client to personalize its learning*...
> >
> > And
> >
> > > Therefore, instead of solving the standard classification problem, *we view the neural network model as a composition of a representation extractor and a small classifier head, and optimize these two components in different manners*.
> >
> > Both of these ideas are present in prior works (Jain et al., Collins et al., Singhal et al. [1]). But they are presented as contributions.
> >
> > Notice that two reviewers were under the impression that this is a new idea. Quoting reviewer 4btx
> > > This work proposes a framework for DP federated learning that separately trains the representation network and local heads.
> >
> > and quoting reviewer Usxb
> >
> > > The authors propose to use a sharing-representation-only method to solve the FL+DP problem in deep learning
> >
> > This point needs to be made crystal clear. As an actionable feedback: I invite the authors to rewrite the contributions section to clearly and unambiguously state the real contribution of the paper. Section 3 (problem formulation) should also reference these prior works when introducing the decomposition into representation+classifier.
> >
> > This is my main concern and I am willing to raise my score if this is properly addressed.
> >
> > [1] Singhal et al., Federated Reconstruction: Partially Local Federated Learning. NeurIPS 2021

---

> > > ### Author Response · Authors · 2022-12-02
> > > **Response to the comments on the contributions (2/2)**
> > >
> > > To clarify the concerns of the reviewer, we will rewrite the first paragraph (the text above Figure 1) of the contribution as follows.
> > >
> > > > In this work, we identify an important bottleneck for achieving high utility in FL under a tight privacy budget: There exists a magnified conflict between learning the representation function and classification head, *when* we clip gradients to bound their sensitivity (which is required for achieving DP). This conflict causes slow convergence of the representation function and disproportional scaling of the local gradients, and consequently leads to the inevitable utility drop in DP FL. To address this issue, we observe that in many FL classification scenarios, participants have minimal disagreement on data representations (Bengio et al., 2013; Chen et al., 2020; Collins et al., 2021), but possibly have very different classifier heads (e.g., the last layer of the neural network). Therefore, instead of solving the standard classification problem, **we borrow ideas from the literature of model personalization** and view the neural network model as a composition of a representation extractor and a small classifier head, and optimize these two components in different manners. In the proposed scheme, CENTAUR, we train a single differentially private global representation extractor while allowing each participant to have a different personalized classifier head. **Such a decomposition has been considered in previous arts like Collins et al. and Singhal et al, but only in a non-DP setting, and also in Jain et al, but only for a linear embedding case.**
> > >
> > >
> > >
> > > We agree with the reviewer that the relation between the current submission and the previous works should be clearly stated. In section 3 (problem formulation), we will include the reference to these prior works when introducing the decomposition of the representation and classifier head, as requested by the reviewer.
> > >
> > > However, we also would like to point out that in our submission, we have pointed out the relation between our work and Collins et al., 2021 (the fact that our work is built on it). Please see the first sentence in the last paragraph of section 1.1.
> > > > “Our work builds on the FedRep algorithm (Collins et al., 2021) … “.
> > >
> > > We have also stated that Jain et al., (2021) has proposed a framework similar to this work. Please see the second to the last paragraph of section 1.1.
> > > > “The closest work to our approach in the literature is Jain et al. (2021), who also propose differentially privately learning shared low-dimensional linear representation with individualized classification head.”
> > >
> > > Please let us know if we have addressed the concerns from the reviewer. We are happy to make further adjustments.

---

> > > > ### Comment · Reviewer_CCKF · 2022-12-06
> > > > **Re: Response to the comments on the contributions**
> > > >
> > > > Thank you for the updated statement of contributions. I believe this is a more accurate account of the true contribution of the paper and should resolve the ambiguity.
> > > >
> > > > These promised changes have addressed my main concern, and I have raised my score accordingly.

---

> > > > > ### Author Response · Authors · 2022-12-06
> > > > > **Thank you for raising the score!**
> > > > >
> > > > > We thank the reviewer for providing the constructive feedback and we are glad that our revised statement of contribution removed the ambiguity. We thank the reviewer again for raising the score.

---

> > > ### Author Response · Authors · 2022-12-02
> > > **Response to the comments on the contributions (1/2)**
> > >
> > > We thank the reviewer for providing detailed and actionable feedback.
> > >
> > > We would like to clarify the misunderstanding caused by our wordings in the paper. We do not want to claim that "the idea of sharing representations only" is the contribution of this paper. The technique itself is indeed very fundamental, and can have many applications including training personalized models, as presented in the prior work. It is also based on the basics of fine-tuning in machine learning. We do not claim any credit for the technique itself, and will update the text to avoid making such an impression.
> > >
> > > The prior work, Jain et al. paper takes such a technique as a "given" standard way to train personalized models, and asks the question of how to perform model personalization under differential privacy guarantees (as the title, abstract, and problem statement of the paper suggest). The paper states "we formulate a model for reasoning rigorously about the loss to privacy incurred by sharing information for model personalization". It assumes a given type of "shared information" as "useful feature representation or starting set of parameters for optimization", and designs a DP algorithm for it, and provides an analysis. This is a very important problem; it overlaps with a sub-problem that we solve (which we analyze differently) in our paper, but it is not the same problem that we tackle in our paper.
> > >
> > > Given this context, the novelty of our work is twofold.
> > >
> > > First, in this paper, we show that differentially private federated learning based on DP-FedAvg (which is widely suggested in the literature and implemented in DP-FL libraries eg Opacus for FL) on *heterogenous* data can fail at outperforming standalone training. This is a major issue, and if the accuracy cost of the algorithm is simply associated with randomness in DP mechanisms, without an in depth investigation, it will be wrongly interpreted as an inherent cost of DP for federated learning. We investigate this issue, and identify one root cause of this problem, which is in the magnified conflict between learning the representation function and classification function, *when* we clip gradients to bound their sensitivity (which is required for achieving DP). We show that the significant disagreement between local classifiers can result in large gradient values in the upper classification part of the model (please see our response “Empirical evidence for Remark 3.1” to reviewer Usxb) which after norm bounding (clipping) can force the updated gradient values on the representation parts to become small and disproportional among clients. Specifically,
> > >
> > > * [Small Gradient for Representation] This results in a low signal to noise ratio for the representation parts after we add the noise (which is the 2nd step for achieving DP). Subsequently, the updates for representation are slowed down which result in inefficient extraction of useful representation features from local data. Due to this slow convergence, under a fixed DP budget (which restricts the number of iterations), local models could not fully extract useful representation features from local data, which results in their low accuracy.
> > >
> > > * [Disproportional Gradient Among Clients] For FL under data heterogeneity, the local gradients (especially in the upper classification part) at the global optimal solution are large, but are in an equilibrium state, i.e. they cancel each other after aggregation. The scaling of these large gradients, after norm clipping, becomes disproportional and breaks this equilibrium and hence the global optimal solution in the non-DP setting is **no longer optimal** in the DP setting.
> > >
> > > So, sharing representations is not only a technique for personalization. In our paper, we suggest it as a *solution* to the learning bottleneck of DP-SGD in federated learning with heterogeneous data.
> > >
> > > Second, we also prove theoretically (Section 5) that our scheme is a better DP model personalization method than the prior work, Jain et al. (we achieve a $\sqrt{d}$ improvement in the utility-privacy tradeoff). Jain et al rely on solving the least square problem exactly at server side, by performing SVD on perturbed noisy representation matrix. Hence, the generalization guarantee of their algorithm intrinsically has an expensive $d^{1.5}$ dependency on the data dimension $d$. By contrast, we perform noisy gradient descent at server side and improve upon this error dependency by a factor of $\sqrt{d}$. Their algorithm is also limited to the linear representation learning problem, unlike our algorithm which enables training multiple layers of shared representations.

---

> ### Author Response · Authors · 2022-11-29
> **Please let us know if your concerns are address.**
>
> Dear reviewer, please let us know if we have addressed your concerns. We are happy to answer new questions as well. Looking forward to your reply.

---

### Official Review · Reviewer_Usxb · 2022-10-30

**Confidence:** 3
**Correctness:** 3
**Technical Novelty And Significance:** 3
**Empirical Novelty And Significance:** 2
**Recommendation:** 6

**Clarity, Quality, Novelty And Reproducibility:**

The paper is easy to read and the overall quality is good. It is not super novel in terms of the main algorithm (Figure 2 by Collins et al. (2021) looks almost the same as the Figure 2 in this paper), but the authors' insight is clear, and the analysis of utility and privacy guarantee is kind of solid. I believe the results are reproducible but the code is not provided so I cannot try it by myself.

**Strength And Weaknesses:**

Major strengths:
1. This idea is intuitive and the empirical performance is very good.
2. The theoretical statements are very detailed.

Major weaknesses:
1. The LRL problem assumes that the true model is $y_{ij}=(w_{i}^*)^T (B^*)^T x_{ij}$ where there is no noise in $y_{ij}$. This is different from Jain et al. (2021). I know that you can set the $\sigma_F=0$ in Jain et al. (2021) Lemma 4.4, but this is not mentioned in the current paper, so the comparison may not be fair. Another problem is that the rate in Jain et al. (2021) Lemma 4.4 is for $\\|\cdot\\|_F$ while the rate in Theorem 5.1 in this paper is for $\\|\cdot\\|_2$. As we know $\\|\cdot\\|_2 \leq \\|\cdot\\|_F$ for matrices, this comparison is not fair.
2. Corollary 5.1 is not solid. It claims a lower bound but we only have an upper bound for the utility in Theorem 5.1. This can be revised and needs to be corrected.
3. In the experiments, the authors said 'we do not perform any data augmentation, as we observe that naive data augmentation
for DP training leads to worse accuracy, as also reported in De et al. (2022).' But I think De et al. (2022) actually recommends doing augmentation in the correct way '... leveraging the benefits of data augmentation by averaging
per-example gradients across multiple augmentations of the same image before the clipping operation
(Hoffer et al., 2019) ...' This needs further explanations.

Minor weaknesses:
The writing can be improved. There are many cases that one notation is used before it is introduced. For example,
1. The first paragraph of Page 7 refers to Algorithm 3 which has $s_i$ but it is introduced in Section 5.1.
2. The 'AGGREGATION' method set on Page 7 needs further explanation on how to transform $\\{G_i^t\\}_{i\in\mathbb{C}^t}$ to $G^t$.
3. On Page 7, the authors said 'line 5 in Algorithm 4 uses Gaussian mechanism' twice, but this is not shown in the algorithm.

On page 16, there is a missing remark. Similar in Remark F.1.

On page 18, Equation (29) is based on Remark C.1 but I do not see how. By the way, I guess $\mathrm{dist}(B^*, B^t)\leq 1$ is from a property of $\mathrm{dist}$, but please make it clear somewhere.

The proof for Lemma D.4 and Lemma F.1 is missing.



Some typos:
1. Page 5, Algorithm 1, Line 4, I guess $p_c$ should be $p_g$.
2. Page 6, Remark 4.1, $S_\alpha(p_c, \sigma_g)=\ldots$ needs to be corrected.
3. Page 10, references, there are duplicates for 'Durmus Alp Emre Acar, et al. (2020, 2021)'.
4. Appendix A.4, the sensitivity is $2\zeta_g/n$.
5. Appendix C, the equation above (MSP) should be $E_{x_{ij}}[\langle \frac{1}{\sqrt{N}}A(X),\frac{1}{\sqrt{N}}A(X) \rangle ]$. Similar problem in Page 23 last equation.
6. Appendix Page 18, the first equation, the second line, the last part $\eta_g\sigma_g \zeta_g W^t$ should be $\eta_g\sigma_g \zeta_g W^t/n$
7. Appendix Eq (42), I guess $F_t$ is $F^t$.
8. On page 24, the first line in Section E.2, I think $I_k$ is $I_d$.
9. On page 26, the second last line, Hardt & Price (2014) does not have Algorithm 6.

----------------------------
Update: The major weaknesses listed above are all solved by the authors. However, I have new concerns (listed below) about this paper, which made me change my score to 5 since those concerns are not easily addressable in the camera-ready version of this paper if it is accepted.
1. The writing of this paper is a bit misleading since the proposed framework, CENTAUR, is very similar to existing methods where the only difference is in the implementation of the special setting, Linear Representation Learning. Therefore, the introduction of this paper needs rewriting.
2. The question raised by reviewer CCKF about the comparison to DP-FedAvg-fb is unanswered. The authors have compared their method to DP-FedAvg-*ft*, which uses local fine-tuning of the full model. However, the *fb* variant fine-tunes the last layer only, which follows almost the same idea as the newly proposed method in this paper, therefore it is more valuable to compare with the *fb* variant.
3. Although the empirical experiment shows good results in the given settings with $\varepsilon\in \{0.25, 0.5, 1, 2, 4\}$ for DP, there could be more settings (e.g., more choices of $\varepsilon$) for a more detailed comparison between CENTAUR and other methods. The current results may be surprising to the experts in federated learning since other FL methods are mostly worse than the non-FL baseline even when $\varepsilon=4$. Adding experiments with $\varepsilon > 4$ or even $\varepsilon \rightarrow \infty$, which deactivates the effect from DP, may help address this issue.

---
Update-2: The new responses from the authors include the code (which solves the name issue about 'DP-FedAvg-fb' and makes the results reproducible) and some experimental support for Remark 3.1.

I would change my recommendation score to 6 since this paper points out a reasonable and promising direction of federated learning with differential privacy with theoretical and experimental evidence. I choose not to give an 8 because the theoretical analysis only covers the noiseless setting, which is much narrower than Jain et al. (2021), and the major theoretical contribution, 'the cross-validation based initialization scheme', is not used in the deep learning experiments.


**Summary Of The Paper:**

The authors propose to solve the statistical heterogeneity problem in federated learning with differential privacy by separating the model into two parts, one part for each client is personalized and different while the other part on the server is shared for all the clients. The communication between the client and the server is the non-private gradient for updating the part on the server, and privacy is guaranteed by using the Gaussian mechanism on the server. The authors proved rates for their method on a linear setting for the utility-privacy tradeoff which improves the SOTA by $\sqrt{d}$. The empirical performance is also very good.

**Summary Of The Review:**

The authors propose to use a sharing-representation-only method to solve the FL+DP problem in deep learning, and they provide theoretical guarantees along with good empirical results. I would raise my recommendation score if the problems above are corrected and explained.

\-----

After reading other reviewers' comments, I agree that this paper is misleading regarding its contributions. On Page 2, the authors said, 'In our scheme, we train a single differentially private global representation extractor while allowing each participant to have a different personalized classifier head.' However, such a scheme is already described in the existing literature, not a novel one. Remark 3.1 could be novel but needs further theoretical and experimental verification.

---

> ### Author Response · Authors · 2022-11-17
> **Response to review (2/2)**
>
> > Corollary 5.1 is not solid. It claims a lower bound but we only have an upper bound for the utility in Theorem 5.1. This can be revised and needs to be corrected.
>
> We apologize for the confusion caused by Corollary 5.1. We are **not** establishing a lower bound on the DP budget $\epsilon_{dp}$. Instead, Corollary 5.1 states that CENTUAR outputs a solution that **provably** achieves an $\epsilon_a$ utility within an $\epsilon_{dp}$ budget, under the condition that the tuple $(\epsilon_a, \epsilon_{dp})$ satisfies the inequality ${\tilde c_t \kappa k^{1.5} \mu^2d}/{n}  \leq \epsilon_a \cdot \epsilon_{dp}$. \
> To elaborate a bit more:
> 1. Suppose that the left hand side ${\tilde c_t \kappa k^{1.5} \mu^2d}/{n}$ is fixed, then for a target accuracy $\epsilon_{a}$, we **cannot** establish the theoretical guarantee that CENTAUR achieves the accuracy $\epsilon_a$ within a DP budget of $\epsilon_{dp} \leq \frac{\tilde c_t \kappa k^{1.5} \mu^2d}{n \epsilon_{a}}$  (this is natural since a smaller DP budget $\epsilon_{dp}$ requires a larger noise multiplier $\sigma_g$ which jeopardizes the convergence analysis of CENTAUR). However, we need to emphasize that, we are not ruling out the possibility that such a DP budget $\epsilon_{dp}$ is achieved, since the privacy guarantee that we are establishing is just an upper bound. Hence we are **not** establishing a lower bound.
> 2. Now suppose that all factors other than the number of clients $n$ is fixed. Corollary 5.1 implies that for an $n$ that is sufficiently large, i.e. $n \geq \frac{\tilde c_t \kappa k^{1.5} \mu^2d}{\epsilon_a \cdot \epsilon_{dp}}$, we can establish the guarantee that the output of CENTAUR achieves an $\epsilon_a$ utility within an $\epsilon_{dp}$ budget. This interpretation also allows us to understand the benefit of having a better dependence on $d$: A better dependence on $d$ means that a smaller $n$ is sufficient to achieve the same utility-privacy guarantee.
>
> In order to demonstrate what the utility-privacy tradeoff suggests in practice, we conducted an empirical study which is presented in Figure 1 (page 2) of our paper. It shows that under a fixed privacy budget, CENTAUR achieves a uniformly better accuracy, on the EMNIST dataset.
>
>
> > About whether data augmentation should be used.
>
> Yes, (De et al., 2022) recommends using a new type of data augmentation, which is different from how it is usually done in the literature of deep learning. In our experiments, we tried both types, i.e. the classic data augmentation and the one suggested by (De et al., 2022), but none of them worked.  We conjecture that it is because their experiments focus on the centrailized learning setting, while we are working in a date-heterogeneous federated learning setting; but this is definitely a potential future direction worth looking into. The statement 'we do not perform any data augmentation, as we observe that naive data augmentation for DP training leads to worse accuracy, as also reported in De et al. (2022).' is referring to the comment of De et al. (2022) on the classic data augmentation technique. We have clarified this in our revision. \
> However, we would like to emphasize that since all baselines in our experiments are implemented without data-augmentation, our comparison remains fair in this regard.
>
> > On page 18, Equation (29) is based on Remark C.1 but I do not see how
>
> In Remark C.1, we have
> $$\frac{\delta^{(2)} \sqrt{k}}{1-\delta^{(1)}} \leq \frac{s_k^2 (1 - \epsilon_0^2)}{36s_1^2} \Leftrightarrow \frac{2\delta^{(2)} \sqrt{k} s_1^2}{1-\delta^{(1)}} \leq \frac{s_k^2 (1 - \epsilon_0^2)}{18} \leq \frac{s_k^2 (1 - \epsilon_0^2)}{4}  ,$$
> which implies Eq.~(29) since $E_0 = 1 - \epsilon_0^2$.
>
> > The proof for Lemma D.4 and Lemma F.1 is missing.
>
> Lemma D.4 is a direct consequence of Lemmas D.1 to D.3. We have added its proof for completeness. Proof of Lemma F.1 can be found in section G, below Theorem G.1.
>
> > On page 26, the second last line, Hardt & Price (2014) does not have Algorithm 6.
>
> By Algorithm 6, we mean the one presented in our appendix, not the one in Hardt & Price (2014) (which does not exist).

---

> > ### Comment · Reviewer_Usxb · 2022-11-18
> > **Thanks for the clarification!**
> >
> > I would raise my recommendation score to 6 but not higher since I see that the novelty in this paper is a bit marginal. I understand that there is an improvement in terms of the rate and the algorithm. Note that Remark 3.1 is quite insightful but lacks theoretical or experimental support. It could help make this paper stronger if the authors elaborate on Remark 3.1 with more details.
> >
> > ---
> >
> > Update: I changed my score to 5 because of the issues I added in the 'Strength And Weaknesses' section above, which I learned from others' reviews.

---

> > > ### Author Response · Authors · 2022-11-28
> > > **Empirical evidence for Remark 3.1 (2/2)**
> > >
> > >
> > >
> > > Note that DP-FedAvg \[rep\] stands for the local gradient norm of DP-FedAvg on the representation part only.
> > >
> > >
> > > |           | DP-FedAvg (homo) | DP-FedAvg \[rep\] (homo) | CENTAUR (homo) | DP-FedAvg (hetero) | DP-FedAvg \[rep\] (hetero) | CENTAUR (hetero) |
> > > | --------- | ---------------- | ------------------------ | -------------- | ------------------ | -------------------------- | ---------------- |
> > > | iteration | mean (std)       | mean (std)               | mean (std)     | mean (std)         | mean (std)                 | mean (std)       |
> > > | 9         | 0.168 (0.04)     | 0.147 (0.032)            | 0.197 (0.016)  | 0.232 (0.055)      | 0.189 (0.028)              | 0.199 (0.025)    |
> > > | 19        | 0.292 (0.046)    | 0.194 (0.013)            | 0.144 (0.026)  | 0.39 (0.058)       | 0.196 (0.013)              | 0.136 (0.035)    |
> > > | 29        | 0.294 (0.05)     | 0.177 (0.015)            | 0.132 (0.032)  | 0.424 (0.052)      | 0.178 (0.016)              | 0.122 (0.04)     |
> > > | 39        | 0.255 (0.046)    | 0.171 (0.017)            | 0.126 (0.034)  | 0.386 (0.053)      | 0.173 (0.018)              | 0.116 (0.041)    |
> > > | 49        | 0.229 (0.041)    | 0.166 (0.021)            | 0.122 (0.038)  | 0.35 (0.052)       | 0.173 (0.018)              | 0.112 (0.043)    |
> > > | 59        | 0.213 (0.038)    | 0.161 (0.024)            | 0.116 (0.039)  | 0.318 (0.05)       | 0.176 (0.018)              | 0.11 (0.047)     |
> > > | 69        | 0.203 (0.038)    | 0.159 (0.025)            | 0.115 (0.043)  | 0.295 (0.047)      | 0.178 (0.017)              | 0.106 (0.047)    |
> > > | 79        | 0.198 (0.038)    | 0.157 (0.026)            | 0.111 (0.044)  | 0.279 (0.048)      | 0.179 (0.018)              | 0.105 (0.05)     |
> > > | 89        | 0.193 (0.038)    | 0.156 (0.028)            | 0.106 (0.044)  | 0.266 (0.046)      | 0.179 (0.019)              | 0.102 (0.051)    |
> > > | 99        | 0.189 (0.038)    | 0.154 (0.028)            | 0.105 (0.048)  | 0.257 (0.045)      | 0.178 (0.021)              | 0.099 (0.051)    |

---

> > > ### Author Response · Authors · 2022-11-28
> > > **Empirical evidence for Remark 3.1 (1/2)**
> > >
> > > We thank the reviewer for taking interest in Remark 3.1. The observation stated therein,  “statistical heterogeneity makes DP guarantees harder to establish”, is the primary motivation for us to study the federated representation learning problem. In fact, that is an important part of the novelty of the paper: To avoid a phenomenon that is (at least partially) the cause of the utility drop in DP federated learning.
> > >
> > > In the following, we provide additional empirical observations that corroborate the statements in Remark 3.1. Specifically, we consider the image classification problem on the EMNIST dataset of digits. We split the training set in the same manner as we did in our submission and set $n=1000$ and $S=5$. We focus on the comparison between the training phase of DP-FedAvg and CENTAUR in order to articulate the phenomenon we predicted in Remark 3.1.
> > >
> > > As a side remark, we note that DP-FedAvg-ft (DP-FedAvg-fb in Yu et al, 2020) is the same as DP-FedAvg during the representation training phase. Their difference lies in that before the testing phase, the classifier head is optimized using the local training dataset, while keeping the representation function fixed.
> > >
> > > In the following table, we report the norm of the local update, which can be regarded as a proxy of the norm of the local gradient, **before the gradient clipping step** of the Gaussian mechanism. We have the following observations:
> > >
> > > 1. The norm of the local gradient in DP-FedAvg is substantially larger than that of CENTAUR in the data heterogeneous setting ($S=5$). This agrees with Remark 3.1 which states that directly solving the standard FL minimization problem leads to large local gradients.
> > > 2. Since the norm of the local gradient consists of two parts: the one of the classifier and the one of the representation, we further report the norm of the local gradient of only the representation part. There are two important observations: i. The norm of the local gradient of the classifier part is substantial. In fact, the gradient norm of the classifier part is observed to surpass the one of the representation part, which agrees with our conjecture that due to the data heterogeneity, the local classifiers can be significantly diversified and this leads to large local gradients. ii. Even if we consider the norm of only the representation part, the gradient norm of DP-FedAvg is still consistently larger than that of CENTAUR. This is evidence that CENTAUR trains a better representation than DP-FedAvg, which also explains the improvement of CENTAUR over DP-FedAvg-ft we observed in our experiments. As a reminder, DP-FedAvg-ft (DP-FedAvg-fb in Yu et al, 2020) fixes the representation (to be the same as that learned by DP-FedAvg) and fine tunes the classifier only before the testing phase.
> > > 3. As a sanity check, we observe that the gradient norm first increases and then decreases for both methods in both heterogeneous and homogeneous settings, which is a common phenomenon in DNN training.
> > > 4. Finally, we also compute the gradient norm in the homogeneous setting, i.e. where all clients have the same data distribution. We observe that the local gradient norm is smaller or comparable to that under the heterogeneous setting.

---

> ### Author Response · Authors · 2022-11-17
> **Response to review (1/2)**
>
> We thank the reviewer for carefully going over our paper and making detailed and valuable suggestions in the review. We have incorporated the suggestions in our revision and welcome further comments from the reviewer. In the following, we address the major comments from the reviewer one by one.
>
> > The problem formulation of the LRL setting is different from that of (Jain et al., 2021).
>
> Since the matrix under consideration, i.e. $B_\perp^\top B^* \in \mathbb{R}^{d \times k}$, is of low rank, we can bound its Frobenius norm by its spectral norm with an extra $\sqrt{k}$ factor, which does not change the dependence on $d$ ($k$ is regarded as a small constant in the literature of matrix factorization). Moreover, with this extra $\sqrt{k}$ factor, Theorem 5.1 in our paper and Lemma 4.4 in (Jain et al. 2021) have the same quadratic dependence on $k$, while the dependency on $d$ is substantially reduced, from $d^{1.5}$ to $d$ (where $NSR =0$ since we focus on the noiseless setting).

---

> > ### Comment · Reviewer_Usxb · 2022-11-18
> > **Thanks for the response**
> >
> > It solved my question perfectly.

---

> ### Author Response · Authors · 2022-11-28
> **Response to Additional concerns (3/3)**
>
> > Adding experiments with $\epsilon > 4$ or even $\epsilon \rightarrow \infty$, which deactivates the effect from DP, may help address this issue.
>
> We agree that the setting $\epsilon \rightarrow \infty$ will illuminate this effect. In fact, we have an elaborated answer in response (4/4) to reviewer CCKF. For the convenience of the reviewer, we repost the question and the answer in the following. We will add the experiment for the large $\epsilon_{dp} setting, as required by the reviewer, in the camera-ready version.
>
> > **Question from  reviewer CCKF** It is always worrisome to see that all competing methods are worse than even the non-FL baseline. Does one really expect all these other methods to fail in practice? Surely there is a regime in which isolated training would fail (say in the extreme case of one example per client). Can the authors give a more nuanced discussion?
>
> **Answer** The reviewer is absolutely correct that there exist extreme scenarios where private FL methods perform better than the non-FL baseline. Specifically, for each method, assuming that the number of records per-client is fixed, then there is a transition point (for the tuple of  the number of clients $n$ and differential privacy budget $\epsilon_{dp}$) after which the private FL algorithm performs better than the non-FL baseline.
> *  Suppose the privacy budget $\epsilon_{dp}$ is fixed. Then as the number of clients $n$ tends to infinity, the standard deviation of injected noise in each update scales with $O(1/n) \rightarrow 0$. Under this increasingly small noise, the private FL method would have a performance similar to the standard (non-DP) FL method, which is stronger than non-FL baseline.
> * Suppose the number of clients $n$ is fixed. Then as the differential privacy budget $\epsilon_{dp}$ tends to infinity, the standard deviation of the injected noise in each update scales as $O(1/\epsilon_{dp}) \rightarrow 0$. Under this increasingly small noise, the private FL method would have a performance similar to the standard (non-DP) FL method, which is stronger than non-FL baseline.
>
> However, note that for a realistic dataset, to maintain a fixed per-client record number, the total number of clients $n$ is limited since we only have finite data points as a whole. This precludes the possibility of growing $n$ to infinity. Also, in any case of real-world interest, $\epsilon_{dp}$ cannot be infinite. \
> In our experiments, for a fixed DP budget $\epsilon_{dp}$ (Table 1), we observe that this transition point for the number of clients $n$ is smaller for CENTAUR than for other FL methods. That is, we require less stringent assumptions on the number of clients to beat the non-FL baseline. For other FL methods, this transition point is too large to be reached, and hence they perform worse than the non-FL baseline\
> On the other hand, suppose that the number of clients $n$ is fixed. We observe in Figure 1 that the transition point for the DP budget $\epsilon_{dp}$ is smaller for CENTAUR than other FL methods. That is, for a reasonable set of differential privacy budget values $\epsilon_{dp}$, CENTAUR has a higher accuracy than all competing private FL methods, and outperforms the non-FL baseline most of the time ($\epsilon >= 0.5$).

---

> > ### Comment · Reviewer_Usxb · 2022-11-28
> > **About the experiments**
> >
> > Thanks for reposting the question and the answer here.
> >
> > You said, 'we observe that this transition point for the number of clients is smaller for CENTAUR than for other FL methods.'. However, from Figure 1, I cannot see the transition point for other FL methods, and it seems like even $\varepsilon\rightarrow\infty$, they will still not be better than 'Stand-alone-no-FL'.
> >
> > You also said, 'We confirm that the DP-FedAvg-ft (second column of Table 1) is exactly the method described by the reviewer, i.e. keeping all but the last layer fixed and fine tune the last layer.' This does not convince me since the existing version of the paper has an insufficient description of all the other methods compared with CENTAUR (I understand the page limit of the main text, but there is no section about the experiment of 'DP-FedAvg-ft' in the appendix either). The code is not provided either. Therefore, I am worried about the actual implementation since I cannot reproduce the results or verify whether the result is from 'DP-FedAvg-ft' or 'DP-FedAvg-fb'.
> >
> > Therefore, although I am excited to see the good experiment results of CENTAUR, I choose not to regard it as a solid strength of this paper.
> >
> > ---
> >
> > Update: In Section B, is the sentence 'For baselines that require local fine tuning, we perform 15 local epochs to fine tune the local head
> > with a fixed step size of 0.01.' a description of 'DP-FedAvg-fb'? If it is, I would agree that the authors used *fb* instead of *ft* and my concerns are mostly solved.

---

> > > ### Author Response · Authors · 2022-11-28
> > > **Response to the comments on the experiments**
> > >
> > > Yes, we confirm that DP-FedAvg-ft listed in our paper is exactly the method DP-FedAvg-fb from Yu et al. 2020. We hope this resolve the concerns of the reviewer. We really hope that you can consider this in further updating your score.

---

> > > ### Author Response · Authors · 2022-11-28
> > > **Code for reproducing our experimental results**
> > >
> > > To further remove the concerns raised by the reviewer, we upload the code for reproducing our experimental results in this anonymous [link](https://anonymous.4open.science/r/Torch-Privacy-Federated-Learning-ICLR2023-rebuttal). Please run the scripts under the folder script/user-level-DP.
> > >
> > > We are happy to answer further questions from the reviewer as well.

---

> > > > ### Comment · Reviewer_Usxb · 2022-11-29
> > > > **Thanks for sharing the code**
> > > >
> > > > The code is written in a very clear way. I confirm that in this code, DP-FedAvg-ft is indeed only fine-tuning the head instead of the whole model.

---

> ### Author Response · Authors · 2022-11-28
> **Response to Additional concerns (2/3)**
>
> > The question raised by reviewer CCKF about the comparison to DP-FedAvg-fb is unanswered. The authors have compared their method to DP-FedAvg-ft, which uses local fine-tuning of the full model. However, the fb variant fine-tunes the last layer only, which follows almost the same idea as the newly proposed method in this paper, therefore it is more valuable to compare with the fb variant.
>
> We emphasize that in our response (4/4) to reviewer CCKF, we have answered the question: We confirm that the DP-FedAvg-ft (second column of Table 1) is exactly the method described by the reviewer, i.e. keeping all but the last layer fixed and fine tune the last layer. We apologize for the mismatch of the naming, i.e. the same algorithm is named DP-FedAvg-fb in Yu et al.. We will fix this in our camera-ready version.
>
> > Although the empirical experiment shows good results in the given settings with $\epsilon \in 0.25, 0.5, 1, 2, 4$ for DP, there could be more settings (e.g., more choices of
> $\epsilon$) for a more detailed comparison between CENTAUR and other methods.
>
> We would like to thank the reviewer for acknowledging our empirical improvement. We argue that for a DP method, the most important regime is when the DP budget $\epsilon_{dp}$ is small. This is because the difference between two distributions grows exponentially with $\epsilon_{dp}$, e.g. for  $\epsilon_{dp} = 4$, the difference can be as large as $e^4 \simeq 54.6$ and this level of DP guarantee is too loose to be meaningful. Therefore, we did not compare with other baselines in a larger $\epsilon_{dp}$ setting. However, we are happy to add results for larger epsilons in camera ready version (this is not a difficult implementation).\
> We did not further consider the smaller $\epsilon_{dp}$ setting since when $\epsilon_{dp}$ is very small, all methods will be worse than the simple stand-alone-no-fl baseline.
>
> > The current results may be surprising to the experts in federated learning since other FL methods are mostly worse than the non-FL baseline even when $\epsilon = 4$.
>
> We agree that in the non-DP FL setting, the FL solvers should outperform the non-FL baseline. However, in the presence of DP constraint, the problem becomes much harder and in our experiments, we do observe the existing FL solvers fail to beat the non-FL baseline. Please see an elaborated discussion in the answer to the next question.

---

> ### Author Response · Authors · 2022-11-28
> **Response to Additional concerns (1/3)**
>
> We would like to emphasize that the additional concerns raised by the reviewer can all be addressed.
>
> > The writing of this paper is a bit misleading since the proposed framework, CENTAUR, is very similar to existing methods where the only difference is in the implementation of the special setting, Linear Representation Learning. Therefore, the introduction of this paper needs rewriting.
>
> First, we would like to point out that in the contribution section of our introduction, we **did not** emphasize the design of the Server/Client procedure as our major contribution. Instead, we highlighted our novel theoretical improvement, the cross-validation based initialization scheme, and the improved empirical result. The relation between our work and the previous works Collins et al. (2021) and Jain et al. (2021) are all clearly discussed in the related work section.\
> However, we still would like to (re)state the algorithmic improvement of our work over these two previous works, to alleviate the reviewer’s concerns.
>
>
> [Comparing with Collins et al. (2021)]  We acknowledge that the design of our Server and Client procedures resemble the prior work Collins et al. (2021), except the additional usage of the Gaussian mechanism. However, this is a common phenomenon in the literature of differential privacy, e.g. DP-SGD is adapted from SGD by adding the Gaussian mechanism. In fact, a major portion of the DP literature is devoted to enhancing the non-DP method with DP guarantee, particularly for the ones that are already known to solve the problem of interest efficiently.\
> Moreover, we emphasize that simply adding the Gaussian mechanism to Collins et al. (2021) does not yield an algorithm with valid DP guarantee since their initialization scheme is data-dependent and is hence non-DP. This is in stark contrast to the case of adapting DP-SGD from SGD where random (and hence data-independent) initialization is acceptable.
>
>
> [Comparing with Jain et al. (2021)]
> Firstly, we would like to invite the reviewer to check our response (1/4) to reviewer CCKF for a detailed algorithmic comparison. Please note that the first point [Comparing with Jain et al. (2021), main procedure] is not specific to the linear representation setting. For the convenience of the reviewer, we repost this point [Comparing with Jain et al. (2021), main procedure] as follows.
>
> **Answer to reviewer CCKF** The main procedure of Jain et al. (2021) is a general alternating minimization algorithm, and hence it requires the server and the clients to solve their own subproblems. While the minimization of the subproblem on the client end can be efficiently conducted, solving the minimizing subproblem on the server side is expensive even with the DP-SGD as a subroutine: To carry out line 6 of Algorithm 1 in Jain et al. (2021) with DP-SGD, every iteration of DP-SGD requires a round of communication between the server and the participating clients (since we are in the federated learning setting, the data are privately held by the clients and hence the gradient can only be computed on the client side). Moreover, if we solve the subproblem on the server side with DP-SGD, the resulting algorithm is a **double loop** method, and tuning the number of steps in the inner loop introduces extra difficulties. \
> In contrast, CENTAUR is a **single loop** method, and requires a single communication round per iteration.
>
> Secondly, although [Jain et al. (2021)] mentioned that their server-side subproblem could be solved with DP-SGD, they do not provide any specific design, theoretical analysis, nor experimental evaluation of this potential new variant of their algorithm. On the contrary, we provide a novel sqrt(d) theoretical improvement and thorough experimental evaluations for CENTAUR on real-world FL tasks under data heterogeneity.

---

> > ### Comment · Reviewer_Usxb · 2022-11-28
> > **About the contribution**
> >
> > Thanks for explaining and addressing this issue about possible miscommunication of the contribution.
> >
> > > First, we would like to point out that in the contribution section of our introduction, we did not emphasize the design of the Server/Client procedure as our major contribution. Instead, we highlighted our novel theoretical improvement, the cross-validation based initialization scheme, and the improved empirical result.
> >
> > In your introduction section, the contribution part has two paragraphs, and what you wanted to highlight, as said above, was only the last two sentences of the first paragraph, and the other parts of that paragraph gave me the impression that the whole procedure was also your contribution. It seems inappropriate to have 1/3 of the contribution part describing the existing framework without citing the paper you have mentioned in the related work. If we use DP-SGD as an example, they did not claim that SGD was part of their contribution.
> >
> > I appreciate your comments about the comparison to the methods from Collins et al. (2021) and Jain et al. (2021), and I hope that they will also be added to the main text.

---

> > > ### Author Response · Authors · 2022-11-28
> > > **Response to the comments on the contribution**
> > >
> > > > It seems inappropriate to have 1/3 of the contribution part describing the existing framework without citing the paper you have mentioned in the related work.
> > >
> > > The purpose of describing the framework is to briefly introduce the methodology that we take in this paper. We will clarify the statement of our contribution to avoid the confusion that the Server/Client procedures are completely newly designed in this paper.
> > >
> > > > I appreciate your comments about the comparison to the methods from Collins et al. (2021) and Jain et al. (2021), and I hope that they will also be added to the main text.
> > >
> > > We will add these comparisons in our related work section.

---

> ### Author Response · Authors · 2022-11-29
> **Response to Update-2**
>
> We thank the reviewer for the careful discussion and for acknowledging our contributions. We will work on the extension of our theoretical guarantee to the more general noisy setting.

---

### Official Review · Reviewer_hVPm · 2022-11-02

**Confidence:** 2
**Correctness:** 3
**Technical Novelty And Significance:** 3
**Empirical Novelty And Significance:** 3
**Recommendation:** 8

**Clarity, Quality, Novelty And Reproducibility:**

The paper makes precise statements and provides adequately detailed proofs. The experiments are likewise well documented.

**Strength And Weaknesses:**

Strengths:

- The paper provides a substantial rate improvement over existing work for the problem of LRL
- The paper provides sound empirical reasoning for their method (i.e. that data representations are less prone to distribution shift)
- The paper supports its results empirically

Weakness:
- The $O(d/n)$ rate is stated is for the matrix 2-norm but Jain et al. 2021 provide convergence guarantees for the Frobenious norm, making the comparison to existing work less clear.
- The proposed algorithm is not incredibly novel compared to prior work.
- The presentation of the analysis for the theoretical results is a bit verbose. Theorem 5.1 and 5.2 are not easily interpreted on their own and it seems more useful instead to provide a more direct statement of Corollary 5.1.
- Given the rate improvement, an investigation into what DP lower bounds suggest is necessary would be nice.

**Summary Of The Paper:**

The paper presents a new algorithm for DP federated learning (with trusted central server) under model personalization assumptions. The algorithm (CENTAUR) works by having the clients only send a subset of their parameters to the central server to be trained privately. The final (classification) layer is trained on each client non-privately. The authors provide rigorous analysis for the specific problem of linear representation learning (LRL) and further support the efficacy of their algorithm empirically with experiments.

**Summary Of The Review:**

The paper makes progress for the problem model personalization under user level differential privacy and federated learning constraints.

---

> ### Author Response · Authors · 2022-11-17
> **Response to review (3/3)**
>
> > The presentation of the analysis for the theoretical results is a bit verbose. Theorem 5.1 and 5.2 are not easily interpreted on their own and it seems more useful instead to provide a more direct statement of Corollary 5.1.
>
> Theorem 5.1 and 5.2 allows us to explicitly see the dependence of the utility and privacy guarantees of CENTAUR on the noise multiplier $\sigma_g$, i.e. the trade-off parameter. The dependence is less straight-forward if only Corollary 5.1 is presented.
>
> > Given the rate improvement, an investigation into what DP lower bounds suggest is necessary would be nice.
>
> We apologize for the confusion caused by Corollary 5.1. We are **not** establishing a lower bound on the DP budget $\epsilon_{dp}$. Instead, Corollary 5.1 states that CENTUAR outputs a solution that **provably** achieves an $\epsilon_a$ utility within an $\epsilon_{dp}$ budget, under the condition that the tuple $(\epsilon_a, \epsilon_{dp})$ satisfies the inequality ${\tilde c_t \kappa k^{1.5} \mu^2d}/{n}  \leq \epsilon_a \cdot \epsilon_{dp}$. \
> To elaborate a bit more:
> 1. Suppose that the left hand side ${\tilde c_t \kappa k^{1.5} \mu^2d}/{n}$ is fixed, then for a target accuracy $\epsilon_{a}$, we **cannot** establish the theoretical guarantee that CENTAUR achieves the accuracy $\epsilon_a$ within a DP budget of $\epsilon_{dp} \leq \frac{\tilde c_t \kappa k^{1.5} \mu^2d}{n \epsilon_{a}}$  (this is natural since a smaller DP budget $\epsilon_{dp}$ requires a larger noise multiplier $\sigma_g$ which jeopardizes the convergence analysis of CENTAUR). However, we need to emphasize that, we are not ruling out the possibility that such a DP budget $\epsilon_{dp}$ is achieved, since the privacy guarantee that we are establishing is just an upper bound. Hence we are **not** establishing a lower bound.
> 2. Now suppose that all factors other than the number of clients $n$ is fixed. Corollary 5.1 implies that for an $n$ that is sufficiently large, i.e. $n \geq \frac{\tilde c_t \kappa k^{1.5} \mu^2d}{\epsilon_a \cdot \epsilon_{dp}}$, we can establish the guarantee that the output of CENTAUR achieves an $\epsilon_a$ utility within an $\epsilon_{dp}$ budget. This interpretation also allows us to understand the benefit of having a better dependence on $d$: A better dependence on $d$ means that a smaller $n$ is sufficient to achieve the same utility-privacy guarantee.
>
> In order to demonstrate what the utility-privacy tradeoff suggests in practice, we conducted an empirical study which is presented in Figure 1 (page 2) of our paper. It shows that under a fixed privacy budget, CENTAUR achieves a uniformly better accuracy, on the EMNIST dataset.

---

> > ### Comment · Reviewer_hVPm · 2022-12-06
> > **Updating Score**
> >
> > I have been following the discussion with the other authors (in addition to the response you provided), and I believe any significant concerns have been addressed. As such I am raising my score.

---

> > > ### Author Response · Authors · 2022-12-06
> > > **Thank you for updating the score!**
> > >
> > > We thank the reviewer for carefully going over the discussions and raising the score. We are glad that we are able to make our contribution clear.

---

> ### Author Response · Authors · 2022-11-18
> **Response to review (2/3)**
>
> > The proposed algorithm is not incredibly novel compared to prior work.
>
> We acknowledge that the design of our Server and Client procedures resemble the prior work Collins et al. (2021), except the additional usage of the Gaussian mechanism. However, this is a common phenomenon in the literature of differential privacy, e.g. DP-SGD is adapted from SGD by adding the Gaussian mechanism. In fact, a major portion of the DP literature is devoted to enhancing the non-DP method with DP guarantee, particularly for the ones that are already known to solve the problem of interest efficiently.\
> Moreover, we emphasize that simply adding the Gaussian mechanism to Collins et al. (2021) does not yield an algorithm with valid DP guarantee since their initialization scheme is data-dependent and is hence non-DP. This is in stark contrast to the case of adapting DP-SGD from SGD where random (and hence data-independent) initialization is acceptable. This is to be elaborated as follows.
> * [Novelty of our initialization scheme] Let us first recall that the initialization procedure is crucial for solving the linear representation learning problem under consideration: The utility analyses of the main procedures of all three works, i.e. Collins et al. (2021), Jain et al. (2021), and this submission, heavily rely on the fact that the initial point $U^{0}$, i.e. the output from the initialization procedure, is already within a constant distance to the optimal solution, e.g. $\mathrm{dist}(U^0, U^*)\leq 0.1$. To achieve this goal, it is shown in Tripuraneni et al. (2021) that by setting $U^0$ to be the top-k singular vectors of a second-order moment matrix, the above requirement can be satisfied. However, if the SVD step is carried out exactly, like Collins et al. (2021), the initialization procedure has **no DP** guarantee. Instead, to ensure that the entire procedure enjoys DP guarantee, the initialization procedure should compute the singular value decomposition (SVD) of the aforementioned moment matrix in a **differentially private** manner.\
> There are two existing options to equip SVD with extra DP guarantee, one is to substitute the exact SVD step with DP-SVD (see [Dwork
>  et al. (2014)](https://dl.acm.org/doi/10.1145/2591796.2591883)), as did Jain et al. (2021), and the other is to use the Private Power Method (PPM), Hardt & Price (2014). We now compare our initialization scheme with these two approaches.
>
> [Comparing with Jain et al. (2021), i.e. DP-SVD]
> * Algorithmically, to perform the aforementioned SVD, Jain et al. (2021) directly uses the result from the DP-SVD method. A key drawback of this approach is the requirement of **transmitting a $d \times d$ sized matrix**, which is prohibitive if $d$ is large. In contrast, CENTAUR only requires **transmitting matrices of size $d \times k$** for a **logarithmic** amount of iterations.
> * In terms of the theoretical guarantee, Lemma 4.6 of Jain et al. (2021) states that for $\mathrm{dist}(U^0, U^*)$ to be bounded by a constant, the DP budget $\epsilon_{dp}$ should satisfy $\epsilon_{dp} \geq c_1 d^{1.5}/n$, where we hide the dependence on other parameters like the rank $k$ and the incoherence parameter $\mu$ in $c_1$ (and set $NSR =0$ since we focus on the noiseless setting). In contrast, Lemma F.2 of our work states that to achieve the same goal, the DP budget $\epsilon_{dp}$ should satisfy $\epsilon_{dp} \geq c_2 d/n$ (simply use the RDP to DP conversion), which has a $\sqrt{d}$ improvement over the result of Jain et al. (2021).
>
> [Comparing with PPM]
>
> While PPM avoids the algorithmic and theoretical drawbacks of DP-SVD mentioned above, it only has a constant success probability, e.g. 0.99. This is insufficient for the purpose of initializing CENTAUR in the linear representation learning case, since the failure event **cannot** be neglected even with a large $n$ (the number of clients). \
> The initialization procedure of CENTAUR boosts the success probability of PPM to $1 - O(n^{-k})$, which quickly approaches $1$ as $n$ increases (this is the setting of interest since we usually assume there are a large amount of clients participating the training procedure in federated learning). Moreover, we achieve this result without compromising the DP guarantee of PPM: Concretely, since the proposed cross validation scheme requires **no further access** to the dataset, given the outputs of a logarithmic amount of i.i.d. PPM trials, it can be regarded as a post-processing procedure, which allows us to establish the DP-guarantee and further the $\sqrt{d}$ improvement over Jain et al. (2021). Consequently, the initialization procedure of CENTAUR is of great novelty.

---

> ### Author Response · Authors · 2022-11-18
> **Response to review (1/3)**
>
> First of all, we would like to thank the reviewer for the positive feedback and detailed comments. We have revised our submission according to these comments and invite the reviewer to take a look. In the following, we address the concerns raised by the reviewer one by one.
>
>
> > The O(d/n) rate is stated is for the matrix 2-norm but Jain et al. 2021 provide convergence guarantees for the Frobenious norm, making the comparison to existing work less clear.
>
> Since the matrix under consideration, i.e. $B_\perp^\top B^* \in \mathbb{R}^{d \times k}$, is of low rank, we can bound its Frobenius norm by its spectral norm with an extra $\sqrt{k}$ factor, which does not change the dependence on $d$ ($k$ is regarded as a small constant in the literature of matrix factorization). Moreover, with this extra $\sqrt{k}$ factor, Theorem 5.1 in our paper and Lemma 4.4 in (Jain et al. 2021) have the same quadratic dependence on $k$, while the dependency on $d$ is substantially reduced, from $d^{1.5}$ to $d$ (where $NSR =0$ since we focus on the noiseless setting). We have added clarifications in the paper after Eq. (5).

---

### Decision · Program_Chairs · 2023-01-20

**Decision:**

Accept: poster

**Justification For Why Not Higher Score:**

The contributions are above the bar of acceptance but the level of impact is not high enough to merit an oral or spotlight presentation.

**Justification For Why Not Lower Score:**

The paper is above the bar for acceptance. It makes progress on an important problem and formally shows a clear, quantitative improvement over prior work.

**Metareview: Summary, Strengths And Weaknesses:**

The paper studies privacy-preserving federated learning (FL) with a trusted central server. The authors give a new differentially private (DP) FL algorithm under model personalization assumptions. Their algorithm follows a general paradigm, where a shared representation is globally learned under DP guarantee, then each client locally train a personalized model using their own data and the learned shared representation.

The authors analyze their algorithm for the specific problem of linear representation learning (LRL) and prove formal privacy and accuracy guarantees. They show $O(\sqrt{d})$ improvement in the accuracy guarantee over the prior work by [Jain et al. 2021]. This is a substantial quantitative improvement for the LRL problem. The authors complement their formal results with empirical evaluation that further supports their findings.

The authors have also made several improvements to their paper during the discussion phase including clarifying their contributions, discussing their technical assumptions, and elaborating on some of the important claims in the paper.

The general consensus is that the paper makes progress on an important problem. The authors formally prove a quantitative improvement over prior work and support their results with empirical evaluation.


**Note From Pc:**

if the above contains the word "oral" or "spotlight" please see: "oral" presentation means -> notable-top-5% and "spotlight" means -> notable-top-25%. As stated in our emails, we are disassociating presentation type from AC recommendations